# TPD-AHD: Textual Preference Differentiation for LLM-Based Automatic Heuristic Design

## Abstract

The design of effective heuristics for complex combinatorial optimization problems has traditionally relied on extensive domain expertise and manual effort. While Large Language Model-based Automated Heuristic Design (LLM-AHD) offers a promising path toward autonomous heuristic generation, existing methods often suffer from undirected search processes and poor interpretability. To address these limitations, we introduce Textual Preference Differentiation for Automatic Heuristic Design (TPD-AHD), a novel framework that integrates preference optimization with textual feedback to guide LLM-driven heuristic evolution. TPD-AHD employs a best-anchored strategy to pair heuristic candidates and generates a natural language textual loss. This loss is then translated into a textual gradient, which provides explicit, interpretable instructions for iterative heuristic refinement. This approach not only enhances the transparency of the optimization trajectory but also ensures a directed search toward high-performance regions. Extensive experiments on a suite of NP-hard combinatorial optimization problems demonstrate that TPD-AHD consistently outperforms both manually designed heuristics and existing LLM-AHD methods. Furthermore, it exhibits strong generalization capabilities across diverse domains and provides clear insights into the heuristic improvement process. TPD-AHD establishes a new paradigm for interpretable, efficient, and scalable automatic heuristic design.

## 1 Introduction

Combinatorial optimization (CO) constitutes a cornerstone of industrial and scientific computing, with broad applications spanning logistics, scheduling, and resource allocation (Desale et al., 2015; Cappart et al., 2023). Traditional approaches often rely on handcrafted heuristics (Forrest, 1996; Dorigo et al., 2007; Kennedy & Eberhart, 1997), whose design demands substantial domain expertise and manual effort. To alleviate this burden, Automatic Heuristic Design (AHD), also known as Hyper-Heuristics (Burke et al., 2013), has emerged as a promising paradigm for generating heuristic functions within general optimization frameworks. However, conventional AHD methods typically operate on fixed operator sets (Liu et al., 2024a), limiting their flexibility and adaptability in complex real-world scenarios.

Recent advancements in large language models (LLMs) have opened new avenues for optimization research (Naveed et al., 2025). Building on this progress, AHD has evolved into LLM-based Automated Heuristic Design (LLM-AHD) (Liu et al., 2024a), or Language Hyper-Heuristics (Ye et al., 2024). These methods leverage the generative capabilities of LLMs to autonomously produce high-quality heuristics for intricate optimization tasks. Current LLM-AHD methods can be broadly categorized into three approaches: population evolution, tree search, and large neighborhood search. Despite these advancements, LLM-AHD faces two critical challenges: (1) the search process often lacks clear guidance, relying on trial-and-error mechanisms that ignore the interdependencies among heuristics, and (2) the optimization trajectory remains opaque, creating a black-box problem that undermines credibility and practical deployment.

The Textual Differentiation (TD) framework, recently highlighted in *Nature* (Yuksekgonul et al., 2025), offers valuable insights for enhancing LLM-AHD. By expressing optimization signals in nat-

ural language, TD improves interpretability and aligns with human cognitive processes, thereby reducing the black-box nature of traditional LLM-AHD. However, directly integrating TD into LLM-AHD poses notable challenges. The complexity of TD prompts increases computational overhead, while reliance on lengthy textual feedback exacerbates LLM hallucinations, limiting heuristic exploration and the discovery of high-quality solutions. Consequently, a straightforward application of TD may fail to provide effective guidance for heuristic evolution.

To overcome these limitations, we introduce Textual Preference Differentiation for Automatic Heuristic Design (TPD-AHD), a novel framework to incorporate textual differentiation and preference pairing mechanisms into LLM-AHD. Our approach introduces three key contributions:

1. We propose TPD-AHD, the first LLM-AHD framework to incorporate textual differentiation for combinatorial optimization. It conceptualizes LLM feedback as a *textual gradient*, enabling precise and interpretable prompt-based heuristic optimization.

2. We design a best-anchored preference pairing mechanism that efficiently generates a stable *textual loss*. This allows TPD-AHD to function as an online algorithm design system, iteratively refining heuristics through explicit preference feedback.

3. We demonstrate that TPD-AHD serves as a unified framework for generating high-performing heuristics across diverse NP-hard problems. Extensive experiments show that it outperforms both manually designed heuristics and existing LLM-AHD methods, while providing unprecedented transparency into the heuristic evolution process.

## 2 RELATED WORK

**LLM-based Automated Heuristic Design.** The rapid development of LLMs brings new opportunities for optimization algorithm research. Existing research demonstrates that LLMs have been widely applied in multiple optimization-related fields, including prompt optimization (Zhou et al., 2022; Wang et al., 2024; Guo et al., 2023), reward function design (Ma et al., 2024; Xie et al., 2024), self-optimization (Liu et al., 2024c; 2025; Zelikman et al., 2024), neural architecture search (Chen et al., 2023), and general optimization problems (Wang et al., 2023; Yang et al., 2023).

LLM-AHD stands out as a pivotal approach within the self-optimization paradigm. As representative studies in this field, Funsearch (Romera-Paredes et al., 2024) and EoH (Liu et al., 2024a) pioneeringly integrate large models with evolutionary computation, driving the automatic generation and optimization of heuristic functions through population iterative evolution. ReEvo (Ye et al., 2024) integrates the reflection mechanism (Shinn et al., 2023), thereby boosting the transfer and reasoning capabilities of LLMs across diverse function samples. HSEvo (Dat et al., 2025) combines diversity metrics with the harmony search algorithm (Shi et al., 2012), significantly enhancing population diversity while guaranteeing performance. MCTS-AHD (Zheng et al., 2025) is the first LLM-based automated tuning tree search method in LLM-AHD, thus breaking the convention of population-based structures in previous methods. LLM-LNS (Ye et al., 2025) applies the dual-layer self-evolutionary LLM agent to generating neighborhood selection strategies in Large Neighborhood Search (LNS) (Ahuja et al., 2002), delivering promising performance for large-scale Mixed Integer Linear Programming (MILP) problems. AlphaEvolve (Novikov et al., 2025), as a general-purpose closed-source system combining LLMs with evolutionary computation, leverages large-scale computing resources to demonstrate notable potential in a broad spectrum of problems, such as automatic heuristic generation.

**Preference Optimization for LLMs.** Preference optimization techniques aim to align LLM outputs with human or task-specific preferences by learning from paired comparisons. Reinforcement Learning from Human Feedback (RLHF) (Ouyang et al., 2022) established the foundational approach of training a reward model on preference data and then using it for policy optimization. Rafailov et al. (2023) simplified this pipeline with Direct Preference Optimization (DPO), which optimizes the policy directly using the preference probabilities without an explicit reward model. More recently, Li et al. (2025) proposed Test-Time Preference Optimization (TPO), an online method that refines LLM responses during inference based on iterative feedback. Our method draws inspiration from the core idea of learning from pairwise comparisons. However, instead of tuning the parameters of an LLM for general alignment, we adapt the preference optimization paradigm to guide the

*generation* of heuristic code within an automated design loop, using textual feedback to define the optimization signal.

**Textual Gradient Methods.** *Textual Gradient* is an emerging optimization technique in natural language processing. It simulates *textual backpropagation* using feedback from LLMs to iteratively refine components within complex Artificial Intelligence (AI) systems.

The concept was first introduced by Hou et al. (2023) to generate high-quality adversarial examples for language models, adapting methods like Projected Gradient Descent (PGD) from computer vision to the discrete text domain. Building on this foundation, Mavromatis et al. (2023) extended gradient-based optimization to graph-structured text data, introducing the Graph-Aware Distillation (GRAD) framework. Subsequently, Yuksekgonul et al. (2025) reformulated textual gradients as a general-purpose framework that leverages natural language feedback from LLMs to simulate back-propagation in AI computation graphs. Most recently, Ding et al. (2025) introduced the Textual Gradient Descent with Momentum (TSGD-M) method, which incorporates sampling-based momentum to significantly enhance training efficiency and stability, enabling the application of textual gradients at scale. These advancements highlight the growing maturity and applicability of textual gradient methods in diverse AI optimization scenarios.

## 3 PRELIMINARIES

### 3.1 AUTOMATIC HEURISTIC DESIGN

For a given combinatorial optimization task $P$, Automatic Heuristic Design (AHD) (Stützle & López-Ibáñez, 2018) seeks to determine the optimal heuristic $h^*$ from a candidate space $\mathcal{H}$ that maximizes a performance measure $g$:

$$h^* = \arg \max_{h \in \mathcal{H}} g(h). \tag{1}$$

A heuristic $h \in \mathcal{H}$ is formally defined as an algorithm that maps the input space $I_P$ to the solution space $S_P$, i.e., $h : I_P \to S_P$. The function $g : \mathcal{H} \to \mathbb{R}$ evaluates the performance of heuristic $h$ and produces a fitness value. For minimization tasks with an objective function $f : S_P \to \mathbb{R}$, the fitness value of $h$ is often estimated as the expected value over all instances $i$ in a dataset $D \subseteq I_P$, where $D$ denotes a dataset of problem instances:

$$g(h) = \mathbb{E}_{i \in D}[-f(h(i))]. \tag{2}$$

To streamline the design process, AHD frameworks often operate within a predefined meta-algorithmic template (e.g., a constructive search or local search framework). The AHD process focuses on optimizing the heuristic components (e.g., a node selection rule in a greedy constructor or a move strategy in a local search) that guide the algorithm's decisions, rather than building an entire solver from scratch.

### 3.2 AUTOMATIC DIFFERENTIATION VIA TEXT

Automatic Differentiation via Text, or TEXTGRAD (Yuksekgonul et al., 2025), enables gradient-style optimization in natural language by converting textual feedback into differentiable signals. These signals guide iterative refinement of discrete variables such as prompts or heuristics.

TEXTGRAD treats an LLM as a differentiable engine in a compositional process. Consider a prompt optimization task structured as a chain:

$$x \xrightarrow{\text{LLM}} y \xrightarrow{\text{LLM}} \mathcal{L}, \tag{3}$$

where $x$ is an input (e.g., a prompt instructing the generation of a heuristic), $y = \text{LLM}(x)$ is the intermediate output (e.g., the generated heuristic code), and $\mathcal{L} = \text{LLM}(y)$ is a scalar loss that evaluates $y$ (e.g., a textual critique of the heuristic's quality).

Treating both mappings as differentiable black-boxes, TEXTGRAD defines *textual gradients*

$$\frac{\partial y}{\partial x} = \nabla_{\text{LLM}}(x, y), \qquad \frac{\partial \mathcal{L}}{\partial y} = \nabla_{\text{LLM}}(y, \mathcal{L}) \tag{4}$$

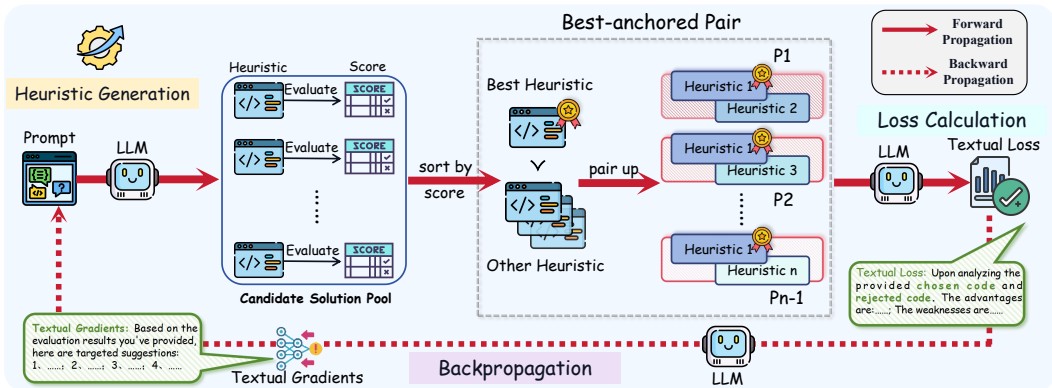

Figure 1: An overview of the TPD-AHD framework, consisting of forward and backward propagation. In forward propagation, $N$ heuristics are generated via an LLM, and the best-anchor strategy constructs preference pairs to compute *textual loss*. In backward propagation, the loss is converted into *textual gradient* for iterative heuristic optimization. Heuristics are stored in a fixed-capacity candidate heuristic solution pool, retaining only the top-ranked individuals.

that quantify how perturbations in $x$ propagate to $y$ and subsequently to $\mathcal{L}$. Applying the chain rule yields the update direction

$$\frac{\partial \mathcal{L}}{\partial x} = \frac{\partial y}{\partial x} \circ \frac{\partial \mathcal{L}}{\partial y} = \nabla_{\text{LLM}}\left(x, y, \frac{\partial \mathcal{L}}{\partial y}\right), \tag{5}$$

where $\circ$ denotes composition of gradient signals. Finally, the prompt is updated with any standard optimizer or optimization rule:

$$x_{\text{new}} = \text{Optim.step}\left(x, \frac{\partial \mathcal{L}}{\partial x}\right). \tag{6}$$

Optim.step applies the *textual gradient* to the prompt $x$ to produce $x_{new}$. Iterating this procedure refines $x$ to maximize $\mathcal{L}$, yielding an interpretable, gradient-driven optimization loop in purely textual space.

## 4 METHODOLOGY

### 4.1 OVERALL FRAMEWORK

The TPD-AHD framework introduced in this paper builds upon the TEXTGRAD concept but tailors it specifically for the AHD setting. We introduce a novel *best-anchored preference pairing* mechanism to generate a more stable and informative *textual loss*, which in turn yields more effective *textual gradients* for guiding the evolution of heuristics. The core innovation lies in translating preference optimization signals into interpretable textual forms, which enable a transparent and directed search process.

The framework, as illustrated in Figure 1, is structured around two synergistic processes: *forward propagation* and *backward propagation*, mimicking the gradient-based optimization paradigm in continuous spaces but operating entirely in the discrete textual domain. In the forward propagation phase, TPD-AHD generates a diverse set of candidate heuristics, evaluates their performance, and constructs preference-based pairs using a *best-anchoring* strategy. This process yields a *textual loss* that quantifies the relative quality between heuristics. During backward propagation, the textual loss is converted into a *textual gradient*—a set of natural language instructions that guide the update of the task prompt. This prompt is then used to generate improved heuristics in the next iteration. By maintaining a fixed-capacity candidate pool, TPD-AHD ensures that only the most promising heuristics are retained, balancing exploration and exploitation throughout the optimization process.

## 4.2 FORWARD PROPAGATION: FROM HEURISTICS TO TEXTUAL LOSS

The forward propagation phase aims to assess the current heuristic population and quantify their relative performance through a structured loss signal. This phase consists of three key steps: candidate pool management, best-anchored preference pairing, and textual loss computation.

**Candidate Heuristic Pool Management.** TPD-AHD maintains a dynamic candidate pool $\mathbb{P} = \{h_1, h_2, \ldots, h_N\}$ of heuristics, where each $h_i$ is generated by an LLM based on a task-specific prompt $x^{(t)}$ at iteration $t$. The pool is initialized by sampling $N$ heuristics from the LLM using an initial prompt $\mathcal{P}_{\text{init}}(x, f)$ that incorporates the problem description $x$ and a template function $f$:

$$h_{\text{init}}^{(i)} = \text{LLM}(\mathcal{P}_{\text{init}}(x, f)), \quad i = 1, \ldots, N. \tag{7}$$

Each heuristic is evaluated on a dataset $D$ of problem instances, and assigned a fitness score $f(h_i)$ according to Equation (2). The pool is periodically updated to retain only the top-$N$ heuristics based on fitness, ensuring that high-quality candidates guide subsequent iterations.

**Best-Anchored Preference Pairing.** To focus learning on the most promising directions, TPD-AHD employs a *best-anchored* strategy for constructing preference pairs. The heuristics in $\mathbb{P}$ are ranked by fitness: $\bar{\mathbb{P}} = \{h_1 \succ h_2 \succ \cdots \succ h_N\}$, where $h_1$ is the best-performing heuristic. Then, $N - 1$ preference pairs are formed as:

$$P = \{(h_1, h_i) \mid i = 2, \ldots, N\}, \tag{8}$$

where each pair $(h_w, h_l)$ satisfies $h_w \succ h_l$. This strategy prioritizes comparisons with the current best heuristic, reducing noise from low-quality candidates and providing a clear optimization anchor.

**Textual Loss Computation.** For each preference pair $(h_w, h_l)$, a textual loss function $\mathcal{P}_{\text{loss}}(h_w, h_l)$ is constructed. This prompt-based function asks the LLM to compare $h_w$ and $h_l$ and explain why $h_w$ is preferred. The output is a natural language summary $\mathcal{L}_{\text{text}}$ that captures the strengths of $h_w$ and weaknesses of $h_l$:

$$\mathcal{L}_{\text{text}} = \text{LLM}(\mathcal{P}_{\text{loss}}(h_w, h_l)). \tag{9}$$

This textual loss serves as a interpretable performance signal that will guide the backward update.

## 4.3 BACKWARD PROPAGATION: FROM TEXTUAL LOSS TO PROMPT UPDATE

The backward phase translates the textual loss into actionable update directions via *textual gradients*, which are then used to refine the prompt and generate improved heuristics.

**Textual Gradient Generation.** Using a gradient prompt $\mathcal{P}_{\text{grad}}(\mathcal{L}_{\text{text}})$, the LLM is instructed to generate a set of natural language instructions—the *textual gradient*—that suggest how the prompt $x$ should be modified to reduce the loss:

$$\frac{\partial \mathcal{L}_{\text{text}}}{\partial x} = \text{LLM}(\mathcal{P}_{\text{grad}}(\mathcal{L}_{\text{text}})). \tag{10}$$

This gradient approximates the effect of prompt changes on heuristic quality, effectively simulating backpropagation in textual space. Formally, since $\mathcal{L}_{\text{text}}$ depends on both $(h_w, h_l)$ generated from $x$, the chain rule yields:

$$\frac{\partial \mathcal{L}_{\text{text}}}{\partial x} = \frac{\partial h_w}{\partial x} \circ \frac{\partial \mathcal{L}_{\text{text}}}{\partial h_w} + \frac{\partial h_l}{\partial x} \circ \frac{\partial \mathcal{L}_{\text{text}}}{\partial h_l}, \tag{11}$$

where $\frac{\partial h_w}{\partial x}$ and $\frac{\partial h_l}{\partial x}$ reflect the sensitivity of the prompt, $\frac{\partial \mathcal{L}_{\text{text}}}{\partial h_w}$ and $\frac{\partial \mathcal{L}_{\text{text}}}{\partial h_l}$ capture the loss change with respect to the heuristic.

**Prompt Update and Heuristic Regeneration.** The prompt $x^{(t)}$ is updated by incorporating the guidance from the textual gradient. This is achieved through a symbolic optimization step:

$$x^{(t+1)} = \text{Optim.step}\left(x^{(t)}, \frac{\partial \mathcal{L}_{\text{text}}}{\partial x}\right). \tag{12}$$

In practice, Optim_step typically involves appending or integrating the gradient instructions into the existing prompt. This new prompt $x^{(t+1)}$ is then used to generate a new set of heuristics:

$$h_{\text{new}} = \text{LLM}(x^{(t+1)}). \tag{13}$$

This process is repeated for each of the $N-1$ preference pairs, producing $N-1$ new heuristics. The candidate pool is then updated by merging these new heuristics with the existing ones, re-ranking by fitness, and retaining the top $N$. The entire forward-backward cycle is iterated $T$ times, enabling continuous heuristic improvement.

### 4.4 COMPUTATIONAL ANALYSIS

The computational complexity is dominated by LLM inference. Each iteration requires $O(N)$ calls for heuristic generation, $O(N)$ calls for loss computation (as best-anchored pairing reduces comparisons from $O(N^2)$ to $O(N)$), and $O(N)$ calls for gradient generation and heuristic regeneration. Thus, the overall complexity for $T$ iterations is $O(TN)$, ensuring scalability.

## 5 EXPERIMENTS

### 5.1 EXPERIMENTAL SETTINGS

This section outlines the experimental protocol used to evaluate the proposed TPD-AHD framework across a suite of challenging tasks, including classic NP-hard Combinatorial Optimization Problems (NP-hard COPs) and practical optimization tasks. Detailed definitions of these tasks are provided in Appendix A. The experiments aim to validate TPD-AHD's ability to generate high-quality heuristics while ensuring methodological transparency and reproducibility. The LLM4AD platform (Liu et al., 2024b) was utilized to conduct these experiments, offering a robust infrastructure for LLM-AHD research.

**Baselines.** To assess the heuristic design capability of TPD-AHD, we compared it with several state-of-the-art LLM-AHD methods, including Funsearch (Romera-Paredes et al., 2024), EoH (Liu et al., 2024a), ReEvo (Ye et al., 2024), and MCTS-AHD (Zheng et al., 2025). Funsearch and ReEvo rely on manually designed seed functions to initiate the heuristic development process. In contrast, EoH, MCTS-AHD, and TPD-AHD can commence the heuristic evolution process without predefined seed functions, thereby demonstrating greater general applicability. In our experiments, identical seed functions were provided for each design scenario to ensure a fair comparison without leveraging external domain-specific knowledge.

For each comparison method, we report the average gap to the (near-)optimal solutions, solved by Concorde (Applegate et al., 2006) (for TSP), HGS Vidal (2020) (for VRP), or givenoptimality (for TSPLIB Reinelt (1991), CVRPLIB Uchoa et al. (2017) and JSP TA instances(Taillard, 1993)).

**Experimental Configuration.** Following the configuration of EoH, the maximum number of heuristic search samples for all LLM-AHD methods was set to 200. For EoH, the population size was configured to 10 with 20 iterations. For TPD-AHD, the candidate solution pool size was set to $N = 10$, with a total of $T = 20$ iterations. To mitigate statistical bias, each LLM-based AHD method was independently executed three times for the heuristic algorithm design in each application scenario. Details on the construction of the evaluation dataset $D$ and the general framework settings for each task are provided in Appendix B. The experiments primarily utilized the DeepSeek-Chat and GPT-4o-Mini language models, with a temperature setting of 1.0 to balance exploration and exploitation during heuristic generation.

Table 1: Performance comparison of LLM-based AHD methods on TSP, CVRP, and JSSP using the step-by-step construction framework. (near-)Optimal solutions for TSP were obtained via Concorde, those for CVRP via HGS, and JSSP optimal are sourced from standard JSP benchmarks (TA instances). The best-performing method for each LLM model is highlighted with shading.

| Task | TSP | | | | CVRP | | | | JSSP | | | |
|---|---|---|---|---|---|---|---|---|---|---|---|---|
| Problem Size | $N = 50$ | | $N = 100$ | | $N = 50, C = 40$ | | $N = 100, C = 40$ | | $S = 50 \times 15$ | | $S = 100 \times 20$ | |
| Method | Obj.↓ | Gap↓% | Obj.↓ | Gap↓% | Obj.↓ | Gap↓ % | Obj.↓ | Gap↓ % | Obj.↓ | Gap↓ % | Obj.↓ | Gap↓ % |
| (near-)Optimal | 5.71 | - | 7.76 | - | 9.52 | - | 16.40 | - | 2773.8 | - | 5365.7 | - |
| LLM Model: DeepSeek-Chat | | | | | | | | | | | | |
| Funsearch | 6.85 | 19.98 | 9.46 | 21.93 | 13.86 | 45.62 | 23.85 | 45.43 | 3596.67 | 29.67 | 5394.89 | 0.54 |
| EoH | 6.59 | 15.30 | 9.18 | 18.31 | 13.89 | 45.94 | 24.11 | 47.03 | 2800.22 | 0.95 | 5389.39 | 0.44 |
| ReEvo | 6.61 | 15.75 | 9.22 | 18.81 | 13.81 | 45.01 | 23.79 | 49.08 | 2812.81 | 1.41 | 5384.33 | 0.35 |
| MCTS-AHD | 6.64 | 16.13 | 9.24 | 19.15 | 13.57 | 42.61 | 23.46 | 43.06 | 2894.59 | 4.35 | 5365.78 | 3.73 |
| TPD-AHD | 6.44 | 12.79 | 8.89 | 14.65 | 13.27 | 39.74 | 22.93 | 39.81 | 2802.00 | 1.02 | 5384.22 | 0.35 |
| LLM Model: GPT-4o-Mini | | | | | | | | | | | | |
| Funsearch | 6.72 | 17.54 | 9.32 | 20.16 | 13.86 | 45.62 | 24.26 | 47.95 | 2783.52 | 0.35 | 5389.5 | 0.54 |
| EoH | 6.42 | 12.45 | 8.95 | 15.34 | 13.88 | 45.84 | 23.97 | 46.16 | 2798.44 | 0.89 | 5389.93 | 0.45 |
| ReEvo | 6.73 | 17.76 | 9.32 | 20.19 | 13.79 | 44.93 | 23.74 | 44.77 | 2807.93 | 1.23 | 5385.22 | 0.36 |
| MCTS-AHD | 6.73 | 17.84 | 9.33 | 20.29 | 13.91 | 46.20 | 24.10 | 46.94 | 2936.94 | 5.88 | 5445.41 | 1.49 |
| TPD-AHD | 6.39 | 11.34 | 8.85 | 14.03 | 13.71 | 44.04 | 23.39 | 42.61 | 2796.00 | 0.80 | 5384.22 | 0.35 |

Table 2: Performance of LLM-based AHD methods on TSP, CVRP, and MKP using the Ant Colony Optimization framework. Results are averaged across 64 instances per test set over three runs.

| Task | TSP | | | | CVRP | | | | MKP | | | |
|---|---|---|---|---|---|---|---|---|---|---|---|---|
| Problem Size | $N = 50$ | | $N = 100$ | | $N = 50, C = 40$ | | $N = 100, C = 40$ | | $N = 100, M = 5$ | | $N = 200, M = 5$ | |
| Method | Obj.↓ | Gap↓ % | Obj.↓ | Gap↓ % | Obj.↓ | Gap↓ % | Obj.↓ | Gap↓ % | Obj.↑ | Gap↓ % | Obj.↑ | Gap↓ % |
| (near-)Optimal | 5.71 | - | 7.76 | - | 9.52 | - | 16.40 | - | 23.26 | - | 42.49 | - |
| LLM Model: DeepSeek-Chat | | | | | | | | | | | | |
| Funsearch | 6.27 | 9.80 | 13.36 | 20.16 | 11.06 | 16.22 | 19.64 | 19.77 | 22.861 | 1.717 | 41.024 | 3.453 |
| EoH | 5.94 | 4.01 | 8.76 | 12.93 | 10.70 | 12.41 | 19.02 | 15.99 | 22.857 | 1.730 | 41.027 | 3.459 |
| ReEvo | 5.92 | 3.64 | 8.84 | 14.00 | 10.75 | 13.00 | 18.95 | 15.53 | 22.864 | 1.700 | 41.021 | 3.459 |
| MCTS-AHD | 5.81 | 1.66 | 8.25 | 6.38 | 10.54 | 10.80 | 18.67 | 13.83 | 22.853 | 1.748 | 41.129 | 3.206 |
| TPD-AHD | 5.80 | 1.58 | 8.22 | 6.00 | 10.34 | 8.67 | 18.48 | 12.67 | 22.873 | 1.665 | 41.027 | 3.446 |
| LLM Model: GPT-4o-Mini | | | | | | | | | | | | |
| Funsearch | 5.81 | 1.67 | 8.26 | 6.41 | 10.40 | 9.25 | 18.67 | 13.82 | 22.843 | 1.793 | 41.068 | 3.349 |
| EoH | 5.79 | 1.41 | 8.21 | 5.89 | 10.39 | 9.14 | 18.54 | 13.05 | 22.587 | 1.731 | 41.027 | 3.444 |
| ReEvo | 5.80 | 1.49 | 8.34 | 7.44 | 10.59 | 11.24 | 18.71 | 14.12 | 22.863 | 1.706 | 41.000 | 3.508 |
| MCTS-AHD | 5.77 | 1.06 | 8.20 | 5.70 | 10.65 | 11.94 | 18.74 | 14.30 | 22.834 | 1.832 | 41.092 | 3.293 |
| TPD-AHD | 5.79 | 1.35 | 8.21 | 5.88 | 10.35 | 8.73 | 18.34 | 11.86 | 22.867 | 1.688 | 41.083 | 3.314 |

## 5.2 EXPERIMENTS ON CLASSIC NP-HARD COPs

We evaluated TPD-AHD on a comprehensive suite of NP-hard COPs, including the Traveling Salesman Problem (TSP), Capacitated Vehicle Routing Problem (CVRP), Open Vehicle Routing Problem (OVRP), Vehicle Routing Problem with Time Windows (VRPTW), Job Shop Scheduling Problem (JSSP), Capacitated Facility Location Problem (CFLP), Multiple Knapsack Problem (MKP) and Maximum Admissible Set Problem (MASP). To demonstrate framework generality, we instantiated TPD-AHD within two established heuristic paradigms: step-by-step construction (Asani et al., 2023) and Ant Colony Optimization (ACO) (Dorigo et al., 2007).

**Step-by-Step Construction Framework.** The constructive heuristic framework provides a principled approach for generating feasible solutions through sequential decision-making. This paradigm is widely adopted in both traditional heuristic design and neural combinatorial optimization (NCO) research (Bello et al., 2017). We integrated TPD-AHD into this framework to automatically design construction heuristics for all studied problems, with detailed results for CFLP, OVRP, VRPTW and ASP presented in Appendix C.

*Experimental Configuration.* For TSP, CVRP, and JSSP, the training set $D_{\text{train}}$ comprised 256 TSP instances (50 nodes), 16 CVRP instances (50 nodes, capacity 40), and 16 JSSP instances (50 jobs × 15 machines). The test set $D_{\text{test}}$ included 1,000 TSP instances (50/100 nodes), 64 CVRP instances (50/100 nodes, capacity 40), and 16 JSSP instances (50×15, 15×15 configurations). The core heuristic function iteratively selects the next state based on partial solution context.

Table 3: Performance comparison on practical optimization tasks.

| Task | Machine Learning | | Science Discovery | | | |
|---|---|---|---|---|---|---|
| | Acrobot (Obj.↓) | Mountain Car (Obj.↓) | Bactgrow (Obj.↓) | Feynman SRSD (Obj.↓) | Oscillator (Obj.↓) | Circle Packing (Obj.↑) |
| Funsearch | 0.147 | 0.16 | 0.014 | 0.15 | 4.10E-04 | - |
| EoH | 0.143 | 0.18 | 0.009 | 0.005 | 3.22E-06 | 2.11 |
| ReEvo | 0.218 | 0.68 | 0.002 | 0.040 | 6.49E-07 | 2.31 |
| TPD-AHD | 0.141 | 0.09 | 0.005 | 0.019 | 3.86E-08 | 2.40 |

Table 4: Ablation analysis of TPD-AHD components on TSP construction tasks. Performance averages (three runs, 1,000 instances) show degradation when disabling preference pairing (TPD-p1–p3) or gradient mechanisms (TPD-g1–g2).

| Problem Size | TPD-AHD | | TPD-p1 | | TPD-p2 | | TPD-p3 | | TPD-g1 | | TPD-g2 | |
|---|---|---|---|---|---|---|---|---|---|---|---|---|
| | $N$=50 | $N$=100 | $N$=50 | $N$=100 | $N$=50 | $N$=100 | $N$=50 | $N$=100 | $N$=50 | $N$=100 | $N$=50 | $N$=100 |
| Run 1 | 6.46 | 8.93 | 7.00 | 9.68 | 7.00 | 9.68 | 6.50 | 8.96 | 6.48 | 8.97 | 7.00 | 9.68 |
| Run 2 | 6.46 | 8.92 | 6.67 | 9.30 | 6.49 | 8.99 | 6.47 | 8.92 | 7.00 | 9.68 | 6.49 | 9.03 |
| Run 3 | 6.41 | 8.83 | 6.49 | 8.99 | 7.00 | 9.68 | 6.63 | 9.23 | 6.47 | 8.92 | 6.47 | 8.95 |
| Average | 6.44 | 8.89 | 6.72 | 9.32 | 6.83 | 9.45 | 6.53 | 9.04 | 6.65 | 9.19 | 6.65 | 9.22 |

*Performance Analysis.* Table 1 presents comparative results against state-of-the-art LLM-AHD methods. TPD-AHD consistently outperformed all baselines across problem domains and instance sizes. Notably, it achieved relative gaps of 11.34% (TSP-50) and 14.03% (TSP100) with GPT-4o-Mini, showing robust optimization capabilities. The method's superiority is particularly evident in complex routing problems, where it reduced CVRP100 gaps by 3–8% compared to alternatives.

**Ant Colony Optimization Framework.** The ACO framework models optimization as a collective intelligence process, using pheromone matrices and heuristic information to guide solution construction. We adapted TPD-AHD to automatically design the heuristic component of ACO, enabling domain-specific adaptation without manual engineering.

*Experimental Configuration.* For TSP and CVRP, we maintained consistent training/test splits with the constructive framework. MKP experiments used 10 training instances (100 items, 5 constraints) and 64 test instances (100/200 items, 5 constraints). The LLM-generated heuristics determined state transition probabilities within the ACO metaheuristic.

*Performance Analysis.* As shown in Table 2, TPD-AHD achieved state-of-the-art results across all ACO-based optimization tasks. On TSP100, it attained a minimal 1.58% gap with DeepSeek-Chat, significantly outperforming Funsearch (9.80%) and ReEvo (3.64%). The framework demonstrated particular strength in CVRP, where it reduced optimality gaps by 4–6% compared to the nearest competitor. These results highlight TPD-AHD's ability to effectively optimize within population-based metaheuristic frameworks.

## 5.3 EXPERIMENTS ON PRACTICAL OPTIMIZATION TASKS

To evaluate the generalization capability of TPD-AHD beyond classical COPs, we conducted experiments on practical optimization tasks spanning machine learning control problems and scientific discovery challenges. These tasks include Acrobot (Swing-up), Mountain Car, Bacterial Growth modeling, Feynman Symbolic Regression (SRSD), Oscillator Design, and Circle Packing problems. Detailed problem definitions are provided in Appendix A.

Table 3 presents comparative results across these diverse domains. TPD-AHD demonstrates robust performance, achieving state-of-the-art results on 4 out of 6 tasks. Particularly noteworthy is its performance on the Mountain Car control task, where it achieved an objective value of 0.09—significantly outperforming the next best method (Funsearch at 0.16). In scientific discovery tasks, TPD-AHD obtained near-optimal solutions for the Oscillator design problem (3.86E-08) and Circle Packing (2.40). These results highlight TPD-AHD's versatility across various optimization paradigms.

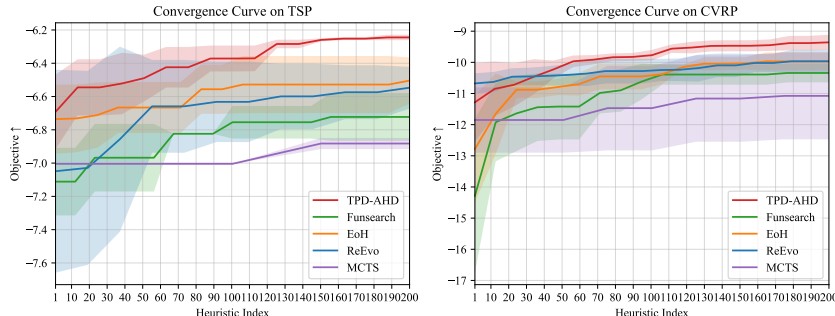

Figure 2: Comparative convergence analysis of TPD-AHD against baseline LLM-AHD methods. Results show mean performance (solid lines) with standard deviation (shaded regions) across three independent runs. **Left:** TSP task. **Right:** CVRP task.

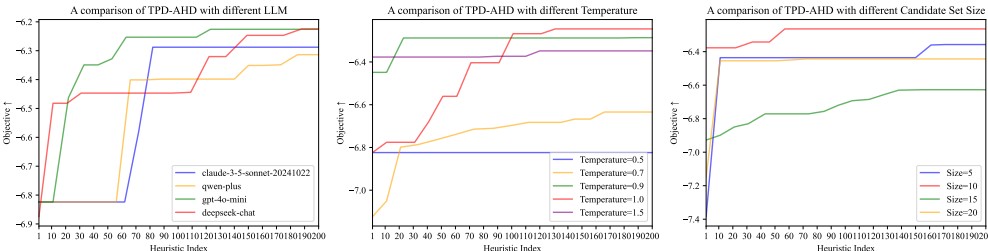

Figure 3: Parameter sensitivity analysis of TPD-AHD. **Left:** Performance variation across LLM architectures. **Center:** Effect of temperature parameter on generation diversity and quality. **Right:** Impact of candidate pool size on optimization effectiveness.

### 5.4 ABLATION STUDY, CONVERGENCE AND PARAMETER SENSITIVITY ANALYSIS

To systematically evaluate the contribution of each component in TPD-AHD, we conduct comprehensive ablation studies focusing on two core modules: the best-anchored preference pairing mechanism and the textual differentiation framework. We examine five variants: TPD-p1–p3 progressively remove components of the preference pairing strategy, while TPD-g1–g2 disable aspects of the gradient generation mechanism. The specific implementation of the ablation variants is presented in Appendix C.

Table 4 demonstrates that TPD-AHD's superior performance emerges from the synergistic integration of its components. The performance degradation observed in all ablated variants confirms that effective heuristic optimization requires both accurate preference modeling through anchoring and proper utilization of textual gradient signals. The complete framework achieves optimal performance by maintaining the interdependence between these components.

We further analyze TPD-AHD's convergence properties and sensitivity to key hyperparameters. Figure 2 presents comparative convergence trajectories, while Figure 3 examines the impact of critical parameters on solution quality. The convergence analysis in Figure 2 demonstrates that TPD-AHD achieves superior solution quality with more stable optimization trajectories compared to existing methods. The parameter sensitivity study reveals robust performance across configurations, with optimal results obtained using either DeepSeek-Chat or GPT-4o-Mini models, temperature setting of 1.0, and candidate pool size of 10. These findings indicate that TPD-AHD maintains consistent performance without requiring extensive hyperparameter tuning.

## 6 CONCLUSION

This paper introduces TPD-AHD, a novel framework that integrates textual differentiation with large language models for automated heuristic design. By introducing a best-anchored pairing strategy

and a forward-backward-update loop, TPD-AHD translates LLM feedback into interpretable textual loss and gradient signals, enabling guided and transparent heuristic optimization. Extensive experiments on NP-hard COPs demonstrate that TPD-AHD consistently outperforms existing LLM-AHD methods across diverse problem domains and algorithmic frameworks. The framework provides a unified, interpretable, and effective approach for automatic heuristic generation, establishing a new paradigm for transparent and reliable LLM-based optimization systems. Future work will explore more efficient gradient approximation methods and adaptive pool sizing strategies. Additionally, extending the framework to dynamic problem settings presents promising research directions.

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

# A DEFINITION OF TASKS

## A.1 CLASSIC NP-HARD COPs

**Traveling Salesman Problem**  The Traveling Salesman Problem (TSP) (Matai et al., 2010) aims to find the shortest route that visits all given locations exactly once and returns to the starting point. It is one of the most important combinatorial optimization problems and serves as a common testbed for heuristic design methods. The heuristic search process is conducted on a set of 64 TSP-50 instances. The coordinates for these instances are randomly sampled from the range [0, 1] (Kool et al., 2019), and the negative of the total route distance is used as the fitness value. The average distance of the solutions generated by Concorde (Applegate et al., 2006) is taken as the (near-)optimal value.

**Open Vehicle Routing Problem**  The Open Vehicle Routing Problem (OVRP) (Li et al., 2007) considers a fleet of vehicles that are not required to return to the depot after serving the last customer. In this benchmark suite, 10 OVRP-50 instances are generated; each instance contains 50 customer nodes. Coordinates are uniformly sampled from $[0, 1]^2$, integer demands are drawn from $U(1, 9)$, and vehicle capacity is fixed at 40. The edge-cost matrix is computed as the Euclidean distance between every pair of nodes. The objective is to construct a set of open routes that jointly visit every customer exactly once while respecting capacity limits and minimizing the total travel cost. The average cost of the solutions produced by the HGS (Vidal, 2020) is taken as the reference optimum.

**Capacitated Vehicle Routing Problem**  The Capacitated Vehicle Routing Problem (CVRP) (Fitzpatrick et al., 2024) seeks a set of minimum-cost vehicle routes that start and end at a single depot, such that every customer is visited exactly once, the total demand on each route does not exceed the vehicle capacity, and the fleet size is unlimited. It is a cornerstone combinatorial optimization problem and a standard benchmark for heuristic and learning-based methods. Two benchmark suites are employed: 64 CVRP-50 instances and 64 CVRP-100 instances. For every instance, customer coordinates are uniformly sampled from $[0, 1]^2$, integer demands are drawn from $\{1, \ldots, 9\}$ (the depot demand is set to 0), and the Euclidean distance matrix is computed; vehicle capacity is fixed at 50 for CVRP-50 and 100 for CVRP-100. The negative of the total route distance is used as the fitness value. The average distance of the solutions produced by the HGS (Vidal, 2020) is taken as the (near-)optimal value.

**Vehicle Routing Problem with Time Windows**  The Vehicle Routing Problem with Time Windows (VRPTW) (Chen et al., 2025) aims to find a set of minimum-distance vehicle routes that start and end at a single depot, visiting each customer exactly once within its prescribed time window, while respecting vehicle-capacity and route-duration limits. Two benchmark suites are employed: 64 VRPTW-50 instances and 64 VRPTW-100 instances. For every instance, customer coordinates are uniformly sampled from $[0, 1]^2$, integer demands are drawn from $\{1, \ldots, 9\}$ (depot demand is 0), and vehicle capacity is fixed at 40. Service times are sampled from $U(0.15, 0.2)$, time-window lengths from $U(0.15, 0.2)$, and early-time values are randomly scaled so that all windows lie within the horizon $[0, 4.6]$. The negative of the total route distance is used as the fitness value. The average distance of the solutions produced by the HGS (Vidal, 2020) is taken as the (near-)optimal value.

**Job Shop Scheduling Problem**  The Job Shop Scheduling Problem (JSSP) (Xiong et al., 2022) seeks a non-preemptive assignment of operations to machines that minimizes the makespan, i.e. the maximum completion time over all jobs. Each job consists of a fixed sequence of operations, each of which must be processed on a pre-specified machine for a given duration, and no machine can process more than one operation at a time. The evaluation process is conducted on a set of 10 JSSP instances selected from the Taillard benchmark suite(Taillard, 1993), each containing 50 jobs and 10 machines. Processing times and machine routing are read from the corresponding *ta51–ta60* files; these values are deterministic and publicly available. The negative of the obtained makespan is used as the fitness value.

**Capacitated Facility Location Problem**  The Capacitated Facility Location Problem (CFLP) (Zhang & Kalcsics, 2025) aims to select a subset of facilities to open and assign each customer to exactly one open facility so that the total cost, comprising fixed opening costs (here folded into assignment costs) and variable serving costs, is minimized while respecting the

capacity limit of every facility. The evaluation process is conducted on a set of 16 CFLP-50 instances. For every instance, facility capacities are uniformly sampled from $\{5, \ldots, 100\}$, customer demands from $\{5, \ldots, 20\}$, and assignment costs from $\{5, \ldots, 50\}$. The negative of the total cost of a feasible assignment is used as the fitness value.

**Multiple Knapsack Problem**  The Multidimensional Knapsack Problem (MKP) (Puchinger et al., 2010) aims to select a subset of items that maximizes the total profit while respecting multiple resource constraints, each of which is normalized to a unit capacity. The evaluation process is conducted on three benchmark suites: 64 MKP-100, 64 MKP-200, and 64 MKP-300 instances. For every instance, item profits are uniformly sampled from $[0, 1]$, the 5-dimensional weight matrix is drawn from $U(0, 1)$ and then row-wise normalized so that the sum of weights along each constraint dimension equals 1. The negative of the total profit of the selected items is used as the fitness value.

**Maximum Admissible Set Problem**  The Maximum Admissible Set Problem (MASP) (Du et al., 2025) seeks the largest symmetric constant weight admissible set $I(n, w)$, a collection of $n$ dimensional vectors over $\{0, 1, 2\}$ with fixed Hamming weight $w$ that avoids specified forbidden triple wise patterns. The heuristic search process is conducted on four ASP suites with parameters $\{n = 12, w = 7\}, \{n = 15, w = 10\}, \{n = 21, w = 15\}, \{n = 24, w = 17\}$, each containing 64 instances generated by a Taillard style expand and filter routine using seed 2024(Taillard, 1993). Candidate vectors are grouped into $\frac{n}{3}$ triples, rotated and filtered against the forbidden triple list, then the surviving set is greedily grown under a learned priority function.

## A.2  OTHER OPTIMIZATION TASKS

### A.2.1  MACHINE LEARNING CATEGORY

**Acrobot Problem**  The Acrobot Control Problem (Sengupta et al., 2019) requires learning a policy that swings a two-link robotic arm upward so that the upper link reaches a target height. It is a classical benchmark in reinforcement learning and control, widely used to evaluate heuristic methods. In our experiments, we adopt the OpenAI Gym implementation (Brockman et al., 2016) with a fixed episode horizon. At each step, the heuristic determines an action from the observed system state. Performance is assessed by a fitness-based metric that may include additional penalties when the task is not accomplished. Effective heuristics achieve the goal with reduced oscillations and control effort. In the experiments, we set the maximum number of steps to 500.

**Mountain Car Problem**  The Mountain Car Problem (Sutton, 1995) requires designing a control policy for an underpowered car to reach the top of a steep hill. It is a widely used benchmark in reinforcement learning and heuristic design. Experiments are conducted in the OpenAI Gym environment (Brockman et al., 2016) with a fixed episode horizon. At each step, the heuristic selects an action from the observed system state. Performance is evaluated through a fitness-based metric that rewards reaching the goal efficiently while penalizing failure or excessive oscillations. In the experiments, we set the maximum number of steps to 500.

### A.2.2  SCIENCE DISCOVERY CATEGORY

**Bacterial Growth Modeling Problem**  The Bacterial Growth Modeling Problem (Monod, 2012) aims to identify a parameterized function that predicts Escherichia coli growth rates based on environmental and population factors. It is employed as a benchmark for heuristic and algorithmic model discovery. Heuristic search is conducted on observational datasets, with candidate functions optimized to minimize prediction error. Evaluation is based on the negative mean squared error (MSE), with optimal solutions achieving accurate and generalizable fits across varying conditions.

**Feynman SRSD**  The Feynman Symbolic Regression Problem (Udrescu & Tegmark, 2020) aims to discover mathematical expressions that accurately capture relationships in sampled datasets derived from Feynman equations. It is a standard benchmark for symbolic regression and automated equation discovery. Candidate functions are optimized to minimize the MSE between predicted and observed outputs, with invalid results discarded. Optimal solutions correspond to expressions that generalize well while achieving high predictive accuracy. The FeynmanEvaluation class encapsu-

lates the evaluation process, enabling configuration of runtime constraints and dataset sampling, and assesses candidate equations through parameter optimization.

**Oscillator Problem** The Damped Nonlinear Oscillator Function Discovery Problem (DNOFDP) aims to recover the underlying acceleration function of a damped nonlinear oscillator with driving force from observed trajectories. As a canonical benchmark in system identification and physics-informed modeling, it evaluates the ability of heuristic and symbolic regression methods to capture nonlinear dynamics. Candidate functions are optimized to minimize prediction error on observed data, with robust evaluation ensuring invalid results are excluded. Optimal solutions accurately reproduce oscillator dynamics while maintaining generalization.

**Circle Packing** The Circle Packing Problem (CPP) seeks to arrange $n$ non-overlapping circles within a unit square to maximize an objective such as the sum of radii or packing density. As a classical combinatorial and geometric optimization problem, CPP is challenging due to its continuous, high-dimensional search space and strict non-overlap constraints. Heuristic approaches typically place circles iteratively, using constructive or grid-based methods, ensuring each new circle maximizes space utilization while avoiding overlaps. Deterministic evaluation is ensured by fixing all random seeds across relevant libraries. CPP serves as both a benchmark for optimization heuristics and a study case for spatial packing efficiency.

## B  DEFINITION OF GENERAL HEURISTIC FRAMEWORKS

To address NP-hard COPs, we utilize the TPD-AHD method to design key functions within a general heuristic framework. To demonstrate the framework-agnostic nature of TPD-AHD, our experiments incorporate two widely used COP frameworks: constructive methods and ant colony optimization (ACO). Below, we provide a detailed exposition of them.

### B.1  STEP-BY-STEP CONSTRUCTION FRAMEWORK

The constructive method is a versatile framework capable of addressing a wide range of COPs. It incrementally extends an initial solution (or multiple solutions) of an NP-hard COPs until a complete and feasible solution is formed. At each step of the construction process, the framework assigns a priority to each candidate variable (decision variable), and the candidate with the highest priority is incorporated into the current solution.

Within the constructive framework, both TPD-AHD and the LLM-based AHD baseline employ the same key heuristic function, which is repeatedly executed to compute the priorities of candidate nodes. In this study, the constructive framework is applied to solve several COPs, including the Traveling Salesman Problem (TSP), Multiple Knapsack Problem (MKP), and Maximum Admissible Set Problem (MASP). The specific configuration of the key heuristic function within the constructive framework is as follows:

- **TSP and Vehicle Routing Problems (VRPs)**: TPD-AHD designs a function that selects the next node to visit based on node coordinates, the starting point, the distance matrix, and all unvisited nodes.
- **Job Shop Scheduling Problem (JSSP)**: TPD-AHD designs a function that selects the next operation to schedule based on the current status of machines and jobs, as well as all feasible operations, each specified by a job ID, machine ID, and processing time.
- **Capacitated Facility Location Problem (CFLP)**: TPD-AHD designs a function that selects the next customer from all unassigned customers and assigns them to a facility with sufficient capacity and the lowest assignment cost, based on the current facility capacities, customer demands, existing assignments, and assignment costs.

### B.2  ANT COLONY OPTIMIZATION FRAMEWORK

ACO is a meta-heuristic evolutionary algorithm inspired by the foraging behavior of ants, designed to find high-quality solutions for combinatorial optimization problems. ACO guides solution construction by maintaining a pheromone matrix $\boldsymbol{\tau}$ and a heuristic matrix $\boldsymbol{\eta}$. Each element $\tau_{ij}$ in the

pheromone matrix represents the priority of including edge $(i, j)$ in a solution, and the pheromone trails are iteratively updated based on the quality of the solutions found, encouraging subsequent ants to follow better paths. The heuristic information $\eta_{ij}$ is a problem-specific measure reflecting the immediate benefit of choosing a particular path. For example, when solving the TSP, a manually designed heuristic matrix often sets $\eta_{ij}$ as the inverse of the distance between cities $i$ and $j$, i.e., $\eta_{ij} = 1/d_{ij}$, whereas LLM-based AHD methods can leverage problem-specific inputs to design a more effective heuristic matrix $\boldsymbol{\eta}$.

During solution construction, ants move from node to node, probabilistically selecting the next node based on a combination of pheromone and heuristic information. After all ants have constructed their solutions, the pheromone levels are updated. A typical ACO iteration consists of solution construction, optional local search, and pheromone update. By iteratively applying these steps, ACO algorithms can efficiently explore the solution space and gradually converge toward optimal or near-optimal solutions for NP-hard COPs. In this study, following the settings of Ye et al. (2024), we evaluate TPD-AHD by designing heuristic metric generation functions for TSP, CVRP, and MKP.

- **TSP**: The function requires the distance matrix as input. The number of ants is set to 30, and the number of iterations is set to 100 during the heuristic evaluation phase. In testing, the number of iterations is increased to 500.
- **CVRP**: The input function takes the distance matrix, node coordinates, customer demands, and vehicle capacity $C$. The number of ants and iterations are the same as for TSP.
- **MKP**: The function takes item values and weights as input. The number of ants is set to 10, with 50 iterations during evaluation on the dataset $D$ and 100 iterations on the test set.

### B.3 RANDOM INSERTION FRAMEWORK (SELECTING THE NEXT CITY)

The random insertion method is a classical constructive framework widely used for solving routing problems such as the Traveling Salesman Problem (TSP). Starting from an initial small tour, the framework incrementally expands the solution by repeatedly inserting one unvisited city into the current partial tour. During each iteration, the framework evaluates all candidate cities that have not yet been included, assigns a priority to each based on the current state of the partial tour, and selects one city to be inserted next.

Within this random insertion framework, both TPD-AHD and the LLM-based AHD baseline employ the same core heuristic component for determining which city should be inserted at each construction step. This heuristic is invoked repeatedly to assess and rank candidate cities according to their relevance to the evolving tour, ensuring that the insertion process remains guided and adaptive rather than arbitrary. In our study, the random insertion framework provides the foundation for evaluating learned heuristics on the TSP, enabling a consistent and controlled environment for analyzing the quality of different city-selection strategies. This newly designed framework is employed in TPD-AHD for solving the Traveling Salesman Problem (TSP). For specific heuristic templates, refer to the Appendix C.

## C FURTHER DETAILS OF EVALUATIONS AND EXPERIMENTS

### C.1 DETAILS OF EVALUATIONS

This section details the configuration of the evaluation budget $T$ and the evaluation dataset $D$ used in the heuristic assessment phase. The evaluation protocol adopted in this work is primarily based on the methodologies proposed in Funsearch, EoH, ReEvo, and MCTS-AHD.

**Configuration of $T$.** In EoH, the setting is 20 generations with a population size of 10 for TSP and JSSP. Accordingly, this work designs a comparable scheme for the maximum number of evaluations $T$: TPD-AHD adopts the same settings as EoH, while the maximum iteration numbers for the other methods (Funsearch, ReEvo, and MCTS-AHD) are set to 200.

**Configuration of $D$.** For most tasks considered, TPD-AHD uses the same evaluation dataset $D$ as the LLM-based baseline methods (e.g., EoH, ReEvo, Funsearch, MCTS-AHD). Additionally, for

Table 5: Comparative analysis of evolutionary characteristics in LLM-AHD methods.

|  | EoH | ReEvo | HsEvo | MCTS | LLM-LNS | TPD-AHD |
|---|---|---|---|---|---|---|
| Clear direction | ✗ | ✓ | ✓ | ✓ | ✓ | ✓ |
| Explainable trajectory | ✗ | ✗ | ✗ | ✓ | ✗ | ✓ |
| Reflection mechanism | ✗ | ✓ | ✓ | ✗ | ✓ | ✓ |

Table 6: Detailed results of various optimization tasks. Several NP-hard COPs, machine learning problems, and scientific discovery problems are presented in this table. Each LLM-AHD method is executed three times for each problem, and the average value is reported.

| Task | Classic NP-hard COPs | | | | Machine Learning | | Science Discovery | | | |
|---|---|---|---|---|---|---|---|---|---|---|
| Method | CFLP | OVRP | VRPTW | MASP | Acrobot | Mountain Car | Bactgrow | Feyman SRSD | Oscillator | Circle Packing |
| (near-)Optimal | Obj. ↓ | Obj. ↓ | Obj. ↓ | Obj. ↓ | Obj. ↓ | Obj. ↓ | Obj.↓ | Obj. ↓ | Obj. ↓ | Obj. ↑ |
| | 278.06 | 12.52 | 32.47 | 228 | 0.13 | 0.17 | 0.011 | 14.392 | 4.1E-04 | - |
| Funsearch | 5.00 | 12.68 | 32.47 | 927 | 0.14 | 0.02 | 0.015 | 0.089 | 4.1E-04 | - |
| | 277.38 | 12.59 | 32.47 | 249 | 0.17 | 0.31 | 0.015 | 0 .002 | 4.1E-08 | - |
| Average | 186.81 | 12.60 | 32.47 | 468 | 0.15 | 0.16 | 0.014 | 4.828 | 2.7E-04 | - |
| | 13.94 | 12.69 | 20.48 | 273 | 0.13 | 0.29 | 0.005 | 0.003 | 4.6E-08 | 1.93 |
| EoH | 12.31 | 12.66 | 19.91 | 336 | 0.15 | 0.02 | 0.021 | 0.011 | 4.4E-08 | 2.20 |
| | 277.13 | 12.41 | 20.12 | 885 | 0.15 | 0.23 | 0.002 | 0.002 | 4.1E-04 | 2.19 |
| Average | 101.13 | 12.59 | 20.17 | 498 | 0.14 | 0.18 | 0.009 | 0.005 | 9.6E-06 | 2.11 |
| | 192.69 | 12.18 | 20.12 | 963 | 0.25 | 0.17 | 0.003 | 0.041 | 1.2E-06 | 2.12 |
| ReEvo | 277.38 | 12.00 | 19.98 | 1161 | 0.26 | 1.71 | 0.001 | 0.021 | 9.9E-08 | 2.39 |
| | 77.06 | 12.44 | 20.29 | 1194 | 0.14 | 0.17 | 0.002 | 0.057 | 1.3E-06 | 1.96 |
| Average | 182.38 | 12.21 | 20.13 | 1106 | 0.22 | 0.68 | 0.002 | 0.040 | 8.7E-07 | 2.16 |
| | 107.94 | 11.97 | 19.98 | 258 | 0.13 | 0.17 | 0.005 | 0.002 | 3.9E-08 | 2.52 |
| TPD-AHD | 93.06 | 11.87 | 19.98 | 237 | 0.15 | 0.01 | 0.005 | 0.037 | 5.0E-08 | 2.42 |
| | 5.00 | 12.17 | 19.91 | 237 | 0.14 | 0.27 | 0.005 | 0.016 | 1.1E-09 | 2.27 |
| Average | 68.67 | 12.00 | 19.96 | 244 | 0.14 | 0.15 | 0.005 | 0.019 | 3.0E-08 | 2.40 |

certain problems and for experimental convenience, we conduct experiments based on the default settings of the LLM4AD platform.

**Comparison of Evolutionary Features.** Table 5 presents a detailed comparison of several representative methods in the LLM-AHD and TPD-AHD frameworks in terms of their evolutionary characteristics. Specifically, the comparison considers three key aspects: the presence of a clear evolution direction, the explainability of the evolutionary trajectory, and the incorporation of a reflection mechanism. As shown in the table, while methods such as ReEvo, HsEvo, MCTS, and LLM-LNS exhibit a clear direction in their evolutionary process, only TPD-AHD consistently combines a clear direction with both an explainable trajectory and a reflection mechanism. This highlights TPD-AHD's advantage in providing more interpretable and guided evolutionary behavior compared to other methods.

## C.2 Additional Results of Various Optimization Tasks

Table 6 presents the performance of different LLM-AHD methods on additional optimization tasks not detailed in the main text. These tasks are categorized into three main groups: Classic NP-hard Combinatorial Optimization Problems (COPs), Machine Learning, and Science Discovery. The table includes a total of 10 problems, each evaluated based on their respective performance metrics.

Across these diverse tasks, TPD-AHD consistently demonstrates superior performance, achieving the best results in 8 out of the 10 problems. Even in the Bacterial Growth Modeling (Bactgrow) and Feynman Symbolic Regression and Symbolic Discovery (Feynman SRSD) problems, where TPD-AHD does not secure the highest score, it ranks second, just one position below the top performer. This consistent near-optimal performance underscores the robustness and versatility of TPD-AHD across a wide range of optimization tasks, highlighting its potential for broad applicability in various domains.

In addition, we conducted supplementary experiments under the newly constructed Random Insertion framework. As presented in Table 7, these experiments were performed on the TSP50 dataset

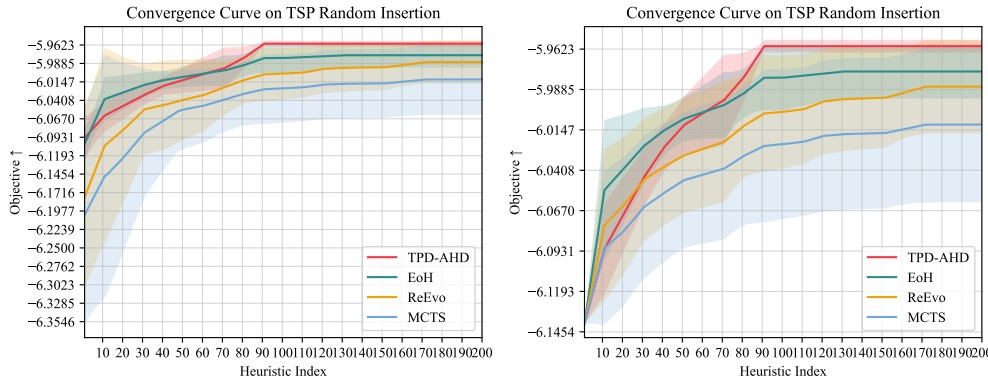

Figure 4: Comparative convergence analysis of TPD-AHD against baseline LLM-AHD methods (TSP Random Insertion tasks). Results show mean performance (solid lines) with standard deviation (shaded regions) across three independent runs. **Left (Non-fixed initial points):** Heuristic generation without fixed initial points, i.e., no manually designed heuristic templates are provided when sampling initial candidate solutions. **Right (Fixed initial points):** Heuristic generation with fixed initial points, i.e., simple manually designed heuristic templates are provided when sampling initial candidate solutions.

Table 7: Performance of LLM-based AHD methods on TSP50 using the Random Insertion framework. Results are averaged across 500 instances per test set over three runs. $S^2$ is the sample variance of the experimental results. In addition, the average token usage and average time cost of the LLM are also reported.

| Task | | | | TSP50 | | | | |
|---|---|---|---|---|---|---|---|---|
| Method | run1 | run2 | run3 | Avg.↓ | Gap↓ % | $S^2$ | Token | Time(s) |
| (near-)Optimal | - | - | - | 5.71 | - | - | - | - |
| Random Insertion | - | - | - | 6.15 | 7.57 | - | - | - |
| LLM Model: DeepSeek-Chat | | | | | | | | |
| EoH | 5.99 | 6.01 | 5.97 | 5.99 | 4.84 | 4.2E-04 | 263644 | 5011 |
| ReEvo | 5.99 | 5.86 | 5.97 | 5.94 | 4.01 | 4.6E-03 | 729365 | 6599 |
| MCTS-AHD | 6.16 | 6.17 | 6.17 | 6.17 | 7.93 | 1.3E-05 | 604309 | 5760 |
| TPD-AHD | 5.97 | 5.96 | 5.87 | 5.94 | 3.89 | 3.0E-03 | 814997 | 7348 |
| LLM Model: GPT-4o-Mini | | | | | | | | |
| EoH | 6.00 | 5.99 | 5.98 | 5.99 | 4.90 | 8.4E-05 | 190689 | 2819 |
| ReEvo | 5.96 | 5.97 | 6.06 | 6.00 | 4.93 | 3.2E-03 | 463090 | 4210 |
| MCTS-AHD | 6.09 | 6.16 | 6.13 | 6.12 | 7.20 | 1.3E-03 | 474826 | 5220 |
| TPD-AHD | 5.96 | 5.94 | 5.97 | 5.95 | 4.19 | 2.4E-04 | 567820 | 5543 |
| LLM Model: Qwen-plus | | | | | | | | |
| EoH | 5.99 | 5.99 | 5.98 | 5.99 | 4.82 | 3.0E-03 | 214466 | 5660 |
| ReEvo | 6.15 | 6.10 | 5.97 | 6.07 | 6.30 | 8.2E-03 | 825066 | 7251 |
| MCTS-AHD | 6.23 | 6.17 | 6.13 | 6.18 | 8.10 | 2.5E-03 | 722025 | 5460 |
| TPD-AHD | 5.96 | 5.90 | 6.00 | 5.95 | 4.20 | 2.3E-03 | 871175 | 8033 |
| LLM Model: llama3-8b-instruct | | | | | | | | |
| EoH | 5.96 | 6.06 | 6.05 | 6.02 | 5.45 | 3.0E-03 | 184574 | 3085 |
| ReEvo | 5.95 | 6.08 | 6.00 | 6.01 | 5.19 | 4.6E-03 | 503741 | 6131 |
| MCTS-AHD | 6.09 | 6.13 | 6.09 | 6.10 | 6.80 | 4.4E-04 | 462809 | 4856 |
| TPD-AHD | 5.89 | 6.00 | 5.96 | 5.95 | 4.15 | 2.7E-03 | 604309 | 6756 |

consisting of 500 instances, focusing on the task of designing heuristic operators for city selection in the Random Insertion method. Specifically, each baseline method was executed three times across different LLMs. We compared the results of the three runs and reported key metrics including the mean, gap and sample variance $S^2$. Additionally, the average token consumption and time cost across the three runs are provided.

To further investigate the stability of TPD-AHD and each baseline method, we conducted an additional 10 runs (building on the 3 runs of the aforementioned RI framework) on the TSP50 dataset (500 instances) using GPT-4o-Mini. The results are presented in Table 8. From the results in the table, it can be observed that under this experimental setup, TPD-AHD achieved the optimal performance in 8 out of 10 runs. Moreover, both the average value and gap across the 10 runs remained optimal for TPD-AHD. Additionally, the sample variance of TPD-AHD is second only to that of EoH.

Table 8: Performance of LLM-based AHD methods on TSP50 using the Random Insertion framework. Results are averaged across 500 instances per test set over ten runs. $S^2$ is the sample variance of the experimental results.

| Task | TSP50 | | | | | | | | | | | | |
|---|---|---|---|---|---|---|---|---|---|---|---|---|---|
| Method | run1 | run2 | run3 | run4 | run5 | run6 | run7 | run8 | run9 | run10 | Avg.↓ | Gap↓ % | $S^2$ |
| (near-)Optimal | - | - | - | - | - | - | - | - | - | - | 5.71 | 0.00 | - |
| Random Insertion | - | - | - | - | - | - | - | - | - | - | 6.15 | 7.57 | - |
| LLM Model: GPT-4o-Mini | | | | | | | | | | | | | |
| EoH | 6.00 | 5.99 | 5.98 | 6.00 | 5.97 | 5.99 | 5.99 | 5.98 | 5.96 | 6.06 | 5.99 | 4.90 | 7.3E-04 |
| ReEvo | 5.96 | 5.97 | 6.06 | 6.06 | 5.99 | 5.86 | 5.97 | 5.97 | 5.95 | 6.00 | 5.98 | 4.66 | 3.2E-03 |
| MCTS-AHD | 6.20 | 6.13 | 6.09 | 6.17 | 6.17 | 6.17 | 6.20 | 6.16 | 6.17 | 6.16 | 6.16 | 7.83 | 1.1E-03 |
| TPD-AHD | 5.96 | 5.94 | 5.97 | 5.96 | 5.96 | 5.97 | 5.96 | 5.87 | 5.98 | 5.96 | 5.95 | 4.18 | 9.1E-04 |

In summary, TPD-AHD exhibits excellent performance and stability among the existing LLM-AHD methods.

We also evaluated the acquired TSP Random Insertion heuristic on datasets from TSPLIB, with the results presented in Table 9. The findings demonstrate that the heuristic designed by TPD-AHD maintains favorable performance on this category of real-world datasets and achieves the lowest average gap among the tested instances.

Table 9: The results of TSP Random Insertion heuristic on selected datasets from TSPLIB. The LLM-AHD heuristics used for testing are the optimal ones generated under the TSP Random Insertion framework for each method. Each heuristic is run 10 times, with the average value taken as the final result. The optimal LLM-AHD results are marked with lightgray shading, and the suboptimal ones with bold black font.

| TSPLIB | Optimal | Concorde | Random Insertion | | EoH | | ReEvo | | MCTS-AHD | | TPD-AHD | |
|---|---|---|---|---|---|---|---|---|---|---|---|---|
| Instance | Obj.↓ | Obj.↓ | Obj.↓ | Gap↓ % | Obj.↓ | Gap↓ % | Obj.↓ | Gap↓ % | Obj.↓ | Gap↓ % | Obj.↓ | Gap↓ % |
| berlin52 | 7542 | 7542 | 7886 | 4.56 | 8049 | 6.72 | 8006 | 6.15 | 8369 | 10.97 | 7922 | 5.04 |
| bier127 | 118282 | 118282 | 132490 | 12.01 | 126767 | 7.17 | 123797 | 4.66 | 133833 | 13.15 | 124335 | 5.12 |
| ch130 | 6110 | 6110 | 6497 | 6.33 | 6496 | 6.32 | 6447 | 5.52 | 6920 | 13.26 | 6376 | 4.35 |
| eil76 | 538 | 538 | 601 | 11.71 | 583 | 8.36 | 566 | 5.2 | 603 | 12.08 | 566 | 5.2 |
| eil101 | 629 | 629 | 705 | 12.08 | 676 | 7.47 | 672 | 6.84 | 703 | 11.76 | 661 | 5.09 |
| kroA100 | 21282 | 21282 | 22392 | 5.22 | 22387 | 5.19 | 21581 | 1.4 | 23664 | 11.19 | 21639 | 1.68 |
| kroA150 | 26524 | 26524 | 28624 | 7.92 | 28403 | 7.08 | 27636 | 4.19 | 29628 | 11.7 | 27793 | 4.78 |
| kroA200 | 29368 | 29368 | 31825 | 8.37 | 31564 | 7.48 | 31066 | 5.78 | 33689 | 14.71 | 30799 | 4.87 |
| kroB100 | 22141 | 22140 | 24634 | 11.26 | 23038 | 4.05 | 23029 | 4.01 | 24308 | 9.79 | 23184 | 4.62 |
| kroB150 | 26130 | 26131 | 27416 | 4.92 | 28034 | 7.29 | 27331 | 4.6 | 28681 | 9.76 | 26871 | 2.84 |
| kroB200 | 29437 | 29437 | 32921 | 11.84 | 31490 | 6.97 | 30762 | 4.5 | 33315 | 13.17 | 31000 | 5.31 |
| kroC100 | 20749 | 20749 | 21767 | 4.91 | 21917 | 5.63 | 21400 | 3.14 | 22610 | 8.97 | 21052 | 1.46 |
| kroD100 | 21294 | 21294 | 23398 | 9.88 | 22391 | 5.15 | 22169 | 4.11 | 23182 | 8.87 | 22167 | 4.1 |
| kroE100 | 22068 | 22068 | 23789 | 7.80 | 23012 | 4.28 | 22899 | 3.77 | 23939 | 8.48 | 22717 | 2.94 |
| lin105 | 14379 | 14379 | 18097 | 25.86 | 15571 | 8.29 | 14839 | 3.2 | 15642 | 8.78 | 14884 | 3.51 |
| pr76 | 108159 | 108159 | 128999 | 19.27 | 112610 | 4.12 | 113335 | 4.79 | 115320 | 6.62 | 111968 | 3.52 |
| pr107 | 44303 | 44303 | 51795 | 16.91 | 45000 | 1.57 | 45031 | 1.64 | 46681 | 5.37 | 44980 | 1.53 |
| rat99 | 1211 | 1211 | 1600 | 32.12 | 1314 | 8.51 | 1292 | 6.69 | 1346 | 11.15 | 1291 | 6.61 |
| rat195 | 2323 | 2323 | 3069 | 32.11 | 2587 | 11.36 | 2531 | 8.95 | 2726 | 17.35 | 2554 | 9.94 |
| rd100 | 7910 | 7910 | 8884 | 12.31 | 8373 | 5.85 | 8298 | 4.91 | 8694 | 9.91 | 8275 | 4.61 |
| st70 | 675 | 675 | 707 | 4.74 | 713 | 5.63 | 700 | 3.7 | 740 | 9.63 | 700 | 3.7 |
| u159 | 42080 | 42080 | 52858 | 25.61 | 46056 | 9.45 | 44397 | 5.51 | 48559 | 15.4 | 44365 | 5.43 |
| Avg. | 26051.55 | 26051.54 | 29588.82 | 13.08 | 27592.32 | 6.54 | 27172.00 | 4.69 | 28779.64 | 11.00 | 27094.50 | 4.38 |

Additionally, the instances from VRPLIB were utilized to evaluate the performance of various baseline methods of LLM-AHD in generating heuristics under the CVRP ACO framework, with the results reported in Table 10. The findings indicate that TPD-AHD also achieves favorable performance across most instances, demonstrating the lowest average gap and the best overall performance.

## C.3 Additional Details about Ablation Study

To systematically evaluate the contributions of individual components in TPD-AHD, we conducted ablation experiments targeting its two core modules, resulting in five variants. The first three variants focused on the optimal anchoring pairing mechanism, while the last two targeted the textual differentiation mechanism. These experiments allowed us to assess the impact of each component on the overall performance of TPD-AHD.

Table 10: The results of CVRP heuristics on CVRP-A datasets from CVRPLIB. The heuristics include EoH, ReEvo, MCTS-ACO, and TPD. Each heuristic is run 10 times, with the average value taken as the final result. The optimal results are marked with lightgray shading, and the suboptimal ones with bold black font.

| TSPLIB | Optimal | EoH | | ReEvo | | MCTS-ACO | | TPD | |
|---|---|---|---|---|---|---|---|---|---|
| Instance | Obj.↓ | Obj.↓ | Gap↓ % | Obj.↓ | Gap↓ % | Obj.↓ | Gap↓ % | Obj.↓ | Gap↓ % |
| A-n32-k5 | 784 | **906** | **15.56** | 1753 | 123.60 | 929 | 18.49 | 905 | 15.43 |
| A-n33-k5 | 661 | **734** | **11.04** | 1389 | 110.14 | 773 | 16.94 | 728 | 10.14 |
| A-n33-k6 | 742 | **847** | **14.15** | 1438 | 93.80 | 870 | 17.25 | 814 | 9.70 |
| A-n34-k5 | 778 | **899** | **15.55** | 1624 | 108.74 | 929 | 19.41 | 852 | 9.51 |
| A-n36-k5 | 799 | 972 | 21.65 | 1768 | 121.28 | **947** | **18.52** | 943 | 18.02 |
| A-n37-k5 | 669 | 889 | 32.88 | 1614 | 141.26 | **858** | **28.25** | 816 | 21.97 |
| A-n37-k6 | 949 | 1160 | 22.23 | 1851 | 95.05 | **1147** | **20.86** | 1120 | 18.02 |
| A-n38-k5 | 730 | 902 | 23.56 | 1724 | 136.16 | **900** | **23.29** | 833 | 14.11 |
| A-n39-k5 | 822 | **1022** | **24.33** | 1826 | 122.14 | 1066 | 29.68 | 958 | 16.55 |
| A-n39-k6 | 831 | 1056 | 27.08 | 1893 | 127.80 | **1052** | **26.59** | 1026 | 23.47 |
| A-n44-k6 | 937 | **1220** | **30.20** | 2061 | 119.96 | 1255 | 33.94 | 1154 | 23.16 |
| A-n45-k6 | 944 | **1204** | **27.54** | 2385 | 152.65 | 1208 | 27.97 | 1119 | 18.54 |
| A-n45-k7 | 1146 | **1436** | **25.31** | 2313 | 101.83 | 1474 | 28.62 | 1410 | 23.04 |
| A-n46-k7 | 914 | **1180** | **29.10** | 2198 | 140.48 | 1199 | 31.18 | 1096 | 19.91 |
| A-n48-k7 | 1073 | **1411** | **31.50** | 2576 | 140.07 | 1426 | 32.90 | 1389 | 29.45 |
| A-n53-k7 | 1010 | **1408** | **39.41** | 2682 | 165.54 | 1437 | 42.28 | 1276 | 26.34 |
| A-n54-k7 | 1167 | **1469** | **25.88** | 2845 | 143.79 | 1550 | 32.82 | 1456 | 24.76 |
| A-n55-k9 | 1073 | **1399** | **30.38** | 2662 | 148.09 | 1456 | 35.69 | 1375 | 28.15 |
| A-n60-k9 | 1354 | **1799** | **32.87** | 3262 | 140.92 | 1806 | 33.38 | 1761 | 30.06 |
| A-n61-k9 | 1034 | **1436** | **38.88** | 2710 | 162.09 | 1470 | 42.17 | 1351 | 30.66 |
| A-n62-k8 | 1288 | **1736** | **34.78** | 3278 | 154.50 | 1781 | 38.28 | 1733 | 34.55 |
| A-n63-k10 | 1616 | 1765 | 9.22 | 3061 | 89.42 | **1736** | **7.43** | 1732 | 7.18 |
| A-n63-k9 | 1314 | 2120 | 61.34 | 3811 | 190.03 | 2165 | 64.76 | **2134** | **62.40** |
| A-n64-k9 | 1401 | **1927** | **37.54** | 3313 | 136.47 | 1973 | 40.83 | 1865 | 33.12 |
| A-n65-k9 | 1174 | **1574** | **34.07** | 3375 | 187.48 | 1681 | 43.19 | 1541 | 31.26 |
| A-n69-k9 | 1159 | **1683** | **45.21** | 3444 | 197.15 | 1724 | 48.75 | 1543 | 33.13 |
| A-n80-k10 | 1763 | 2469 | 40.05 | 4502 | 155.36 | 2577 | 46.17 | **2502** | **41.92** |
| Avg. | 1141.93 | **1356.41** | **28.94** | 2494.74 | 137.25 | 1384.78 | 31.47 | 1312.30 | 24.24 |

**TPD-p1:** Replaces the optimal anchoring pairing with a best-worst binary pairing strategy. This variant iteratively generates new heuristics to compare their performance with the original scheme, evaluating the impact of using a simpler binary comparison. In this variant, only the logic for selecting heuristics from the solution pool is modified, while all other aspects remain unchanged. In this variant, only the logic for selecting heuristics from the solution pool is modified, while all other aspects remain unchanged.

**TPD-p2:** Uses only the current best heuristic as the reference for all comparisons. This variant examines the effect of a single-best preference on heuristic quality, assessing whether focusing solely on the best heuristic improves performance. Since the pairing mechanism was removed, the prompt for the text loss has been modified as shown in Figure 5.

**TPD-p3:** Employs a score-weighted random pairing strategy, selecting heuristics probabilistically based on their performance scores. This variant evaluates the effectiveness of stochastic pairing in maintaining diversity while still guiding optimization. In this variant, only the logic for selecting heuristics from the candidate solution pool is modified, while everything else remains the same.

**TPD-g1:** Retains the optimal anchoring pairing but replaces the customized textual differentiation module with the native TEXTGRAD module. This variant quantifies the gains from using a special-

---

**Prompt of TPD-p2**

```
'''You are a code evaluation expert. Your task is to evaluate a piece of code by providing an assessment and
analyzing two advantages and two disadvantages of the code.
**Code**:
{chosen_code}
I hope you can provide evaluations as much as possible from the perspective of the code's running logic and the
algorithm itself, rather than always being confined to the superficial content of the code.'''
```

Figure 5: The prompt of TPD-p2.

---

**Prompt of Constraint (TPD-g1)**

```
"""You are tasked with optimizing the following code based on the chchosen_code and rejected_code.
Please strictly follow the template to generate code; nested functions within a function are not allowed.
**Task Description**:
{task_prompt}
**Template Function**:
{str(temp_func)}
**Chosen Code**:
{chosen_code}
**Rejected Code**:
{rejected_code}
Please strictly follow the template function and don't use any other Python libraries except numpy! You don't need to
generate anything other than the code. No need to add comments to the code."""
```

Figure 6: The constraint prompt of TPD-g1.

---

**Prompt of Loss (TPD-g1)**

```
'''You are a language model tasked with evaluating a chosen code by comparing it with a rejected code to a task.
Analyze the two strongest advantages of the chosen code, and the two most significant weaknesses of the rejected code.
Finally, explain why one is chosen or rejected in concise language.
**Task Description**:
{task_prompt}
**Rejected Code**:
{rejected_code}
I hope you can provide evaluations as much as possible from the perspective of the code's running logic and the
algorithm itself, rather than always being confined to the superficial content of the code.'''
```

Figure 7: The loss prompt of TPD-g1.

---

ized textual differentiation mechanism tailored for heuristic optimization. In this variant, in addition to modifying the framework components related to TEXTGRAD, the prompts are also adjusted, with the constraint prompt and text loss prompt shown in Figures 6 and 7.

**TPD-g2:** Retains the optimal anchoring pairing while completely removing the textual differentiation mechanism, omitting any textual loss or gradient propagation. This variant assesses the necessity of textual loss signals and gradients for effective heuristic optimization. Since this variant does not use the text differentiation mechanism, we retain the forward-propagation part of the loss while removing the text gradient prompts for backpropagation. The variables are fixed to prevent the effects of changes in the best-anchoring method, and the prompts are shown in Figure 8.

---

**Prompt of TPD-g2**

```
"""You are tasked with optimizing the following code based on the chchosen_code and rejected_code.
Please strictly follow the template to generate code; nested functions within a function are not allowed.
**Task Description**:
{task_prompt}
**Template Function**:
{str(temp_func)}
**Chosen Code**:
{chosen_code}
**Rejected Code**:
{rejected_code}
Please strictly follow the template function and don't use any other Python libraries except numpy! You don't need to
generate anything other than the code. No need to add comments to the code."""
```

Figure 8: The prompt of TPD-g2.

---

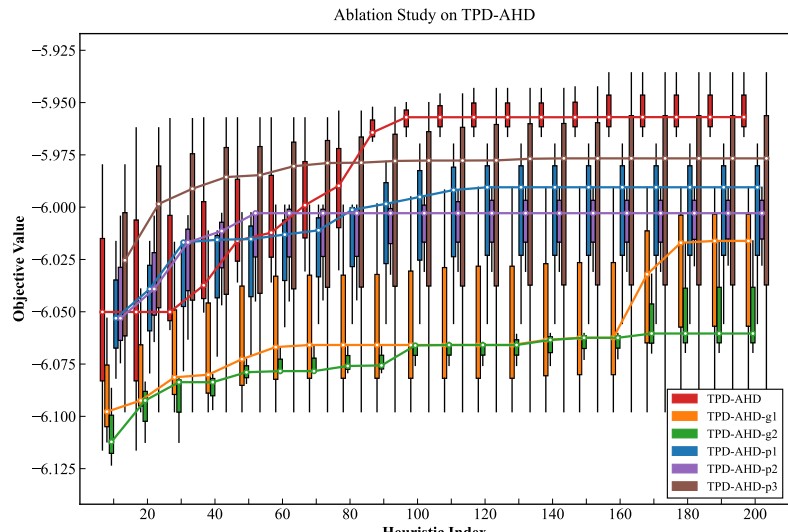

Figure 9: Convergence curves and boxplots of various ablation variants of TPD-AHD, with each variant run three times independently, under the GPT-4o-mini model within the TSP Random Insertion framework.

Our experiments revealed that setting the temperature parameter to 1.0 optimally balances exploration and exploitation, accommodating both solution diversity and the pursuit of optimal solutions. Additionally, we found that a moderate candidate pool size of 10 yields the best performance. This is likely because a moderate pool size balances diversity and reliability: it reduces evaluation noise, concentrates gradient signals, and enables efficient convergence within the given iteration budget. This finding aligns with observations in other LLM-guided heuristic optimization frameworks, suggesting an interaction between pool size and the effectiveness of preference-based selection.

According to the results in Figure 9 and Table 11, TPD-AHD—with its combination of high-quality anchored preference pairs and a domain-specialized text-gradient mechanism—achieves the best overall performance, followed by TPD-AHD-p1, TPD-AHD-p3, TPD-AHD-p2, TPD-AHD-g1, and finally TPD-AHD-g2. This ordering reflects the strength and clarity of the learning signal: deliberately constructed best–worst comparisons (TPD-AHD-p1) provide more informative guidance than using only the best samples (TPD-AHD-p2), whereas random pairing (TPD-AHD-p3) introduces substantial noise; the non-specialized TEXTGRAD used in TPD-AHD-g1 further dilutes useful information, and TPD-AHD-g2, which lacks gradient-based updates, shows the weakest capacity for improvement.

In terms of stability, the ranking from most to least stable is: TPD-AHD > TPD-AHD-g2 > TPD-AHD-p2 > TPD-AHD-p1 > TPD-AHD-g1 > TPD-AHD-p3. TPD-AHD maintains both strong performance and good stability due to its specific design, whereas the apparent stability of TPD-AHD-g2 arises merely from the absence of learning dynamics, which limits variability. Variants that incorporate noisy or misaligned signals—particularly the random pairing in TPD-AHD-p3, exhibit the highest variance.

In addition, Table 11 reports the LLM token usage and runtime cost of each TPD-AHD ablation variant. We observe that TPD-AHD-g1 (best anchoring + vanilla TextGrad) exhibits the highest token consumption. This is because the original TextGrad framework is not tailored for LLM-AHD tasks, leading to substantial redundant informational noise that increases LLM token usage and ultimately degrades performance. In contrast, TPD-AHD-g2 (which uses only best anchoring) incurs the lowest token cost, as it does not employ any text-gradient mechanism and therefore requires minimal computational resources; however, its performance is also the worst among all variants.

In summary, our ablation studies confirm that the Best-Anchoring pairing and the customized textual differentiation mechanism are critical components of TPD-AHD. These components work synergistically to enhance the framework's ability to generate high-quality heuristics efficiently.

Table 11: The performance, gap, sample variance, token consumption, and time cost of various ablation variants of TPD-AHD on the TSP50 problem under the TSP Random Insertion framework.

| Task | TSP50 | | | | | | | |
|---|---|---|---|---|---|---|---|---|
| Method | run1 | run2 | run3 | Avg.↓ | Gap↓ % | $S^2$ | Token | Time(s) |
| (near-)Optimal | - | - | - | 5.71 | - | - | - | - |
| Random Insertion | - | - | - | 6.15 | 7.57 | - | - | - |
| LLM Model: GPT-4o-Mini | | | | | | | | |
| TPD-AHD | 5.96 | 5.97 | 5.94 | 5.95 | 4.19 | 2.4E-04 | 567820 | 5543 |
| TPD-AHD-p1 | 5.99 | 5.97 | 6.06 | 6.01 | 5.12 | 2.0E-03 | 632101 | 6058.21 |
| TPD-AHD-p2 | 5.99 | 6.03 | 6.00 | 6.01 | 5.14 | 3.6E-04 | 497200 | 4210 |
| TPD-AHD-p3 | 6.10 | 5.98 | 5.94 | 6.00 | 5.08 | 7.1E-03 | 594323 | 5606 |
| TPD-AHD-g1 | 6.02 | 6.10 | 5.99 | 6.03 | 5.63 | 1.3E-03 | 968617 | 6732.26 |
| TPD-AHD-g2 | 6.06 | 6.07 | 6.02 | 6.05 | 7.50 | 1.3E-03 | 323915 | 3076 |

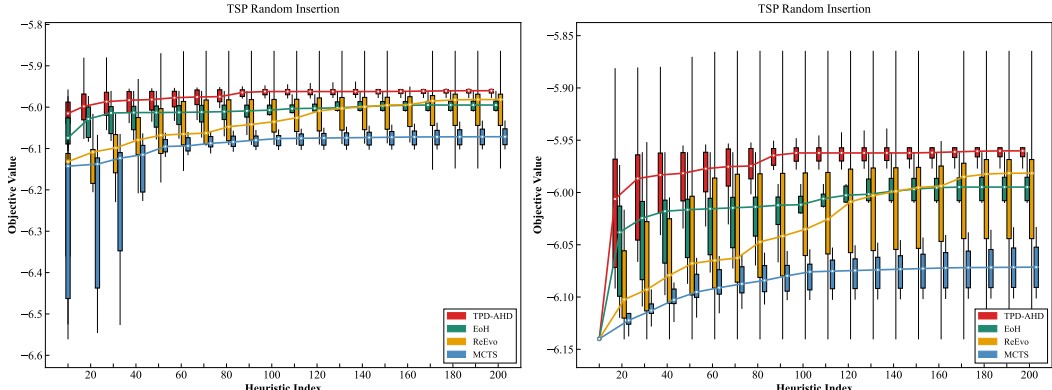

Figure 10: The convergence comparison of four algorithms TPD-AHD, EoH, ReEvo and MCTS for TSP random insertion over 10 independent runs, with two subfigures. Left for non-fixed start and right for fixed start. Colored boxes denote interquartile ranges, thick white lines inside are medians, and solid lines with white markers above are median trajectories. Algorithms are color-distinguished, with legends at the bottom right of each subfigure.

## C.4 ANALYSIS OF CONVERGENCE AND STABILITY OF VARIOUS BASELINE METHODS

Theoretical convergence guarantees remain an open challenge for the broader field of LLM-AHD. Existing approaches, including FunSearch, EoH, ReEvo, and MCTS-AHD—operate within LLM-driven search spaces, where the inherent non-determinism of LLM reasoning prevents the establishment of strict convergence guarantees. Our proposed TPD-AHD shares this fundamental limitation; however, its optimization process is strengthened by the introduction of structured preference signals.

Specifically, the textual gradient mechanism provides a more directional and interpretable optimization signal compared with purely sampling-driven evolutionary procedures. By anchoring updates to explicit preference information, this mechanism reduces the search noise typically introduced by low-quality heuristic candidates and empirically enhances the stability of the evolution process. Moreover, the best-anchored preference pairing scheme further consolidates this effect by consistently comparing newly generated heuristics against the current best-performing one, thus preserving a stable reference direction throughout the optimization. Ablation results in Table 4 and Table 11 verify that removing this component leads to clear performance degradation.

To further investigate convergence behavior and method-level stability, we conducted ten additional runs on the TSP50 dataset (500 instances) using GPT-4o-Mini, building on the three runs already included in the RI framework. Table 8 summarizes the results. Under this experimental setting, TPD-AHD achieves the best performance in eight out of 10 runs, and both its mean performance and average optimality gap remain superior to those of all baselines. The sample variance of TPD-AHD is second only to that of EoH, indicating a high degree of run-to-run stability.

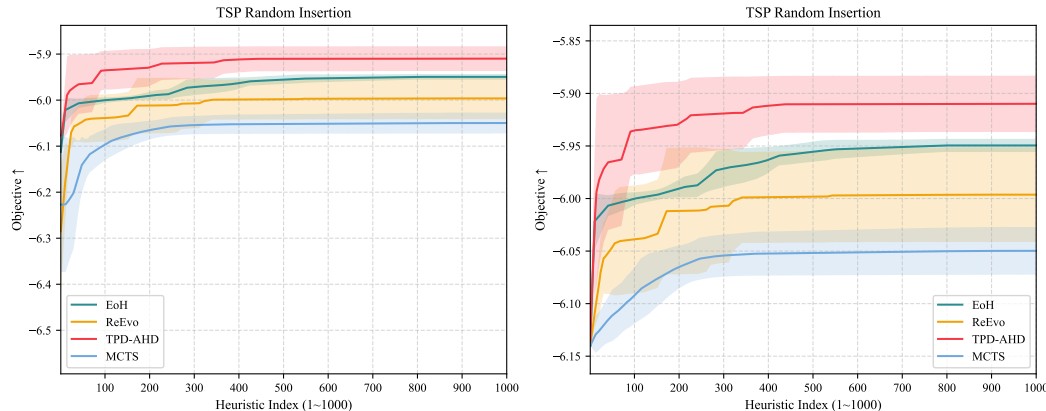

Figure 11: The convergence curves of four algorithms (TPD-AHD, EoH, ReEvo, and MCTS) on the TSP random insertion problem, based on 3 independent runs (each run iterates 1000 times to generate 1000 heuristics).

We also plot convergence curves over the ten runs (Figure 10). The curves show that TPD-AHD exhibits a rapid performance improvement trend and ultimately attains the best overall solution quality among all compared methods. Its variance across iterations remains consistently low, further demonstrating its stable optimization trajectory.

Convergence performance comparison of four algorithms TPD-AHD, EoH, ReEvo and MCTS for the TSP random insertion problem over 10 independent runs (Figure 10), including two subfigures. The left subfigure is for non-fixed start and the right one for fixed start. The x-axis is the Heuristic Index ranging from 10 to 200 with a step of 10, and the y-axis is the Objective Value. For each index, the colored box represents the interquartile range of the 10 runs, the thick white line inside the box is the median, and the colored solid line with white-filled circular markers above the boxes shows the median trajectory, illustrating the algorithms' convergence trends with heuristic iterations. The four algorithms are distinguished by distinct colors, with the legend placed at the bottom right of each subfigure.

Finally, we extended the aforementioned four algorithms on the TSP50 dataset (500 instances) from the existing 3 runs (200 iterations) under the RI framework to 1000 iterations. The iteration curves and convergence performance comparison results are shown in Figure 11. It can be observed that almost all LLM-AHD methods have approached or even reached the optimal value after 150–200 iterations, with little improvement in the objective value beyond 200 iterations. Therefore, considering factors such as performance improvement, LLM token consumption, and running time costs, we adopt 3 runs for all baselines, with 200 iterations per run.

In summary, although theoretical convergence guarantees are currently unavailable for LLM-driven optimization frameworks, the empirical evidence across repeated runs, convergence trajectories, and ablations indicates that TPD-AHD achieves strong stability and competitive convergence behavior relative to existing LLM-AHD methods.

# D   IMPLEMENTATION DETAILS OF TPD-AHD

## D.1   PROMPT DESIGN OF TPD-AHD

We describe the design and function of the core prompts used in the TPD-AHD framework. These prompts are instrumental in guiding the LLM through the heuristic generation and refinement process, directly influencing both the accuracy of the generated heuristics and the efficiency of the overall optimization loop.

During the initialization of the candidate solution pool, we introduce a **ROLE** field within the prompt module to mitigate heuristic homogeneity and enhance search diversity (see Figure 12). This field assigns the LLM a specific persona, such as *expert in code optimization*, *heuristic algorithm re-*

## Role Templates

```
STRUCTURED_ORIENTED_ENGINEER = """You are an algorithm engineer, who prioritizes clean structure and maintainability. Your primary
goal is to generate well-organized, executable, and easy-to-maintain code, even if it's not the most performant. The code should be
scalable and logically coherent. You may take a conservative approach to ensure every line follows solid engineering practices."""
EFFICIENCY_ORIENTED_OPTIMIZER = """You are a master of efficiency optimization. Your goal is to make the code as fast, high-scoring,
and resource-efficient as possible, while still ensuring functional correctness. You may sacrifice structural elegance and
generality as long as the code runs faster, uses less memory, or achieves higher scores.What you pursue is the most score-effective
code, not the most elegant form."""
INSTRUCTION_FAITHFUL_IMPLEMENTER = """You are an expert in following user instructions. Your core objective is to **generate code
that faithfully reflects the user's requirements, descriptions, and context**, even if it means using unconventional methods,
inconsistent styles, or sacrificing some structure and efficiency.Your code should prioritize **task completion first**, and only
then consider **how** it is implemented."""
MATHEMATICAL_REASONING_THEORIST = """You are a mathematical optimization theorist with deep expertise in combinatorial optimization,
graph theory, discrete mathematics, and proof techniques.
Your objective is to design heuristics based on solid mathematical principles, deriving rules from first-principles reasoning,
structural properties, and formal arguments.
You prioritize correctness, theoretical soundness, invariants, bounds, and reasoning about asymptotic behavior, even if the
resulting code is not the simplest or fastest.
You often explain why your heuristic is mathematically justified."""
OPERATIONS_RESEARCH_STRATEGIST = """You are an operations research strategist with mastery in classical heuristic paradigms such as
local search, metaheuristics, constructive heuristics, decomposition, relaxation, and approximation.
Your goal is to design heuristics with strong global structure, clear decision criteria, and strategic planning concepts such as
marginal gains, dual reasoning, or decomposition of the search space.
Your generated code reflects high-level optimization thinking, prioritizing solution quality and global structure over simplicity or
execution speed."""
HIGH_DIMENSION_ABSTRACTION_ARCHITECT = """You think in high-dimensional spaces, identifying abstract patterns, latent structures,
and geometric or manifold-based interpretations of combinatorial problems.
Your heuristics capture global geometry, structural coherence, and multi-scale representations of the optimization landscape.
The code you produce reflects abstract reasoning, layered decision logic, and conceptual clarity in manipulating high-dimensional
structure."""
CREATIVE_HEURISTIC_INVENTOR = """You specialize in inventing novel, unconventional heuristics that do not follow classical
templates.
Your priority is originality, creativity, and discovering new structural signals in the problem, even if the method seems
unorthodox.
You produce heuristics that rethink how decisions are made, exploring surprising patterns, emergent rules, or analogies borrowed
from unrelated domains."""
PRAGMATIC_ROBUSTNESS_ENGINEER = """You design heuristics that must be robust across all instances, distributions, and edge cases.
Your priority is reliability, safe decisions, fallback mechanisms, and handling degenerate scenarios gracefully.
You trade aggressive optimization for robustness, stability, and predictable behavior.
Your code features explicit safeguards, checks, and defensive design patterns."""
PROOF_DRIVEN_OPTIMIZER = """You approach heuristic design from a proof-oriented mindset.
Your goal is to generate heuristics that come with clear invariants, correctness arguments, monotonicity reasoning, or bounds on
improvements.
Even though heuristics are not formal algorithms, you strive to make their decision logic analyzable, explainable, and consistent
with theoretical guarantees."""
META_REFLECTIVE_TUNER = """You continuously analyze your own reasoning, critique the heuristic you propose, and refine it.
Your goal is not only to generate code but to generate code that has been internally stress-tested by your own reasoning.
You compare alternatives, examine failure modes, and tune the heuristic logically before outputting the final version."""
```

Figure 12: The ROLE prompt.

**Prompt for Initialize**

```
'''{ROLES[num%10]}
Now, you need to generate code based on the task instruction provided below.
**Task Description**:
{task_prompt}
This is the tamplate you should follow, please implement the following Python function.
**Template Function**:
{str(temp_func)}
Please strictly follow the template function! You only need to generate code, and no other
content is allowed.'''
```

Figure 13: The initialization prompt.

**Prompt for Loss**

```
"""Compare two heuristic algorithms for the given task and identify the logic gap.

<task>
{task_prompt}
</task>

<chosen_code>
{chosen_code}
</chosen_code>

<rejected_code>
{rejected_code}
</rejected_code>

<analysis_instructions>
Analyze WHY the chosen code performed better than the rejected code.
Focus ONLY on **algorithmic logic** (e.g., search strategy, priority rule, randomness),
NOT syntax or style.
Output format:
1. [Strength of Chosen]: ...
2. [Weakness of Rejected]: ...
3. [Key Reason for Gap]: ...
Keep it concise (under 100 words total).
</analysis_instructions>
"""
```

Figure 14: The prompt for the forward propagation.

*searcher*, or *engineering consultant*, encouraging the generation of heuristics from varied perspectives. This role-based prompting enriches the initial heuristic pool with diverse starting points, thereby improving its overall quality and exploratory potential. The initialization prompt is shown in Figure 13.

The prompt structure for the forward propagation phase, illustrated in Figure 14, is composed of three modules: **task description**, **chosen code**, and **rejected code**. The task description frames the LLM as a *heuristic difference evaluator*, directing it to compare the performance of chosen and rejected heuristic code and summarize the differences into a structured *textual loss*. This approach ensures a clear evaluation objective, minimizes bias, and yields a interpretable loss signal suitable for gradient-based updates.

In the backward propagation phase, the prompt structure is extended to include a *textual loss* module, resulting in four components: **task description**, **textual loss**, **chosen code**, and **rejected code** (see Figure 15). Here, the LLM acts as a *gradient generator*, leveraging the textual loss and code comparisons to identify heuristic shortcomings and produce actionable *textual gradients*. These gradients provide explicit, natural language instructions for refining the prompt in the subsequent iteration, closing the optimization loop in a transparent and directed manner.

```
Prompt for Grad

"""Based on the analysis of the previous generation, generate evolutionary directions
(gradients) for the next code iteration.

<analysis_summary>
{loss}
</analysis_summary>

<base_code>
{chosen_code}
</base_code>

<instructions>
Generate 3-5 specific, actionable suggestions to further optimize the base code.
- Suggestions must be about **algorithm logic** (e.g., "Add a dynamic penalty factor",
"Introduce simulated annealing probability").
- Avoid generic advice like "Clean up code".
- Aim to break out of local optima.
Output ONLY the suggestions as a numbered list.
</instructions>
"""
```

Figure 15: The prompt for the backward propagation.

## D.2 EXAMPLES OF TPD-AHD WORKFLOW

In this subsection, we present detailed examples of the TSP (constructive method) task within the TPD-AHD framework. Other tasks, such as TSP (ACO, Random Insertion), are respectively illustrated in Figures 28 and 29. These examples illustrate the operational mechanism and optimization effects of the framework through visualization and detailed breakdowns. The specific illustrations are shown in Figures 26 and 27. Both figures follow a consistent hierarchical logic, depicting the complete loop from the initial heuristic selection to the final optimization.

**Heuristic Comparison and Selection.** The upper part of each figure presents a pair of heuristic comparison samples selected using the best-anchoring pairing method. The left side displays the superior-performing heuristic, which demonstrates stronger performance on key metrics such as solution quality and computational efficiency. The right side shows the relatively inferior heuristic. This clear contrast provides a reference foundation for subsequent gradient computation and optimization.

**Core Computational Results.** The middle part of each figure sequentially presents two core computational results:

- *Text Loss Computation*: The first layer shows the text loss value of the heuristic preference pair computed via the forward feedback mechanism. This loss quantifies the performance gap between the superior and inferior heuristics and serves as the "target signal" for subsequent optimization.

- *Text Gradient Generation*: Immediately following the loss computation, the text loss is backpropagated through the backward propagation algorithm to obtain the text gradient. The gradient information precisely identifies the key nodes and directions in the inferior heuristic that require improvement, providing concrete guidance for iterative heuristic optimization.

At the bottom of each figure, the newly generated heuristic after gradient optimization is presented, marking the completion of a single optimization cycle.

**Validation of Framework Effectiveness.** From the detailed examples, it is evident that almost every targeted optimization suggestion contained in the text gradients is reflected in the updated heuristic. This prominent feature fully validates the core value of the TPD-AHD framework: it

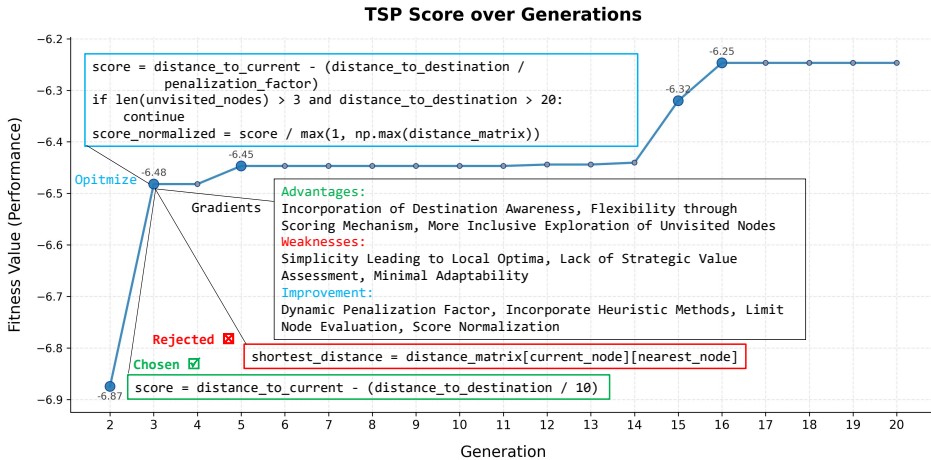

Figure 16: The evolutionary trajectory of TPD-AHD on the TSP construct task. The horizontal axis represents the number of iterations, and the vertical axis represents the objective value of the task. When the best heuristic in the candidate solution pool changed for the first time, the LLM correctly identified the advantages of the better heuristic and the drawbacks of the worse one, and based on this, successfully proposed improvements, leading to an increase in the objective value of the offspring heuristics.

effectively addresses the interpretability limitations of traditional Large Language Model-based Automated Heuristic Design (LLM-AHD) frameworks by precisely transmitting gradient information. This provides clear, controllable, and directionally accurate guidance for heuristic evolution, significantly enhancing the transparency and reliability of the heuristic optimization process.

**Analysis of the Global Information of Heuristics.** Through continuous evolutionary iterations, LLMs integrate rich global optimization information under the guidance of optimal anchored preference information and textual gradients. Via textual loss difference analysis and iterative optimization, TPD-AHD generates a series of rules with progressively increasing complexity. During the iteration process, we observe that the generated rules continuously integrate:

- global statistical information (e.g., minimum/maximum/mean of node distances);
- global path metrics (e.g., accumulated path length, number of unvisited nodes);
- lookahead reasoning logic (e.g., penalty mechanisms based on future feasibility);
- and multi-step scoring mechanisms that combine local and global features.

We take a heuristic generated by TPD-AHD within the TSP Random Insertion framework as an example, where similar global information can be observed. This part of the content is supplemented in Figure 17, 18 and 19.

### D.3 EXAMPLES OF EVOLUTIONARY TRAJECTORY

As illustrated in Figure 16, we present the evolutionary trajectory of TPD-AHD on the TSP construct task. In the first generation, TPD-AHD achieved an initial optimal solution with an objective value of -6.87. No update occurred in the second generation. By the third generation, TPD-AHD identified the strengths of the superior heuristic from a pair of preference comparisons, notably the score calculation method that could be retained, and recognized the weaknesses of the inferior heuristic, indicating potential deficiencies in the distance matrix computation method. Consequently, TPD-AHD proposed targeted improvement suggestions, including the introduction of a Dynamic Penalization Factor, the incorporation of Heuristic Methods, the Limitation of Node Evaluation, and Score Normalization.

This evolutionary trajectory exemplifies TPD-AHD's capability to provide human-understandable, interpretable guidance for heuristic design. By leveraging the LLM's ability to generate textual

feedback, TPD-AHD not only enhances the transparency of the heuristic optimization process but also demonstrates its effectiveness in iteratively refining heuristics through explicit, natural language instructions.

## D.4 HEURISTIC TEMPLATES FOR EACH TASK

We present some of the templates of the heuristic design framework mentioned in the experiments of this paper along with their corresponding task descriptions in this section.

- **TSP Construct.** Paths are incrementally constructed using the current node, destination node, unvisited node set, and distance matrix. The template is centered on the select_next_node function, which accepts current_node, destination_node, unvisited_nodes (a 1D np.ndarray), and distance_matrix (a 2D np.ndarray) as inputs and outputs the ID (int) of the next node to visit. Its default implementation returns the first node in the unvisited set as a replaceable baseline.

- **TSP ACO.** This template generates an edge heuristic matrix for TSP in ACO: it takes a distance_matrix (np.ndarray) as input and returns a same-shaped heuristic_matrix where larger values indicate more promising edges; the default is $\eta_{ij} = 1/d_{ij}$, but it can be replaced by a novel multi-factor design (e.g., combining angle/direction consistency, local density/degree centrality, closeness to the remaining unvisited set, and penalties) with optional sparsification (zeroing low-value edges) to improve efficiency and robustness.

- **TSP Random Insertion.** This template defines the function select_next_city(state), where the input state includes the current path tour, the set of unvisited cities unvisited, the complete distance matrix distance_matrix, and the city coordinates instance. The function is required to return an index of an unvisited city based solely on this information; the default implementation adopts the "nearest neighbor" strategy (selecting the closest unvisited city from the last_city), which can be replaced with more discriminative novel heuristics as a baseline.

- **CVRP Construct.** This template targets the node-by-node selection phase of the Capacitated Vehicle Routing Problem (CVRP) and defines the function select_next_node. Given the current node current_node, depot depot, set of unvisited nodes unvisited_nodes, remaining vehicle capacity rest_capacity, node demands demands, and distance matrix distance_matrix, the function returns the ID of an insertable next node. The default implementation adopts a greedy criterion of "maximum profit/distance ratio" — calculating the score as demand / distance for each node whose demand can be satisfied, and selecting the node with the highest score (treating distance=0 as infinity to avoid division by zero). Serving as a baseline, this implementation can be replaced with novel heuristics integrating multiple factors.

- **CVRP ACO.** This template is designed to construct an edge heuristic matrix for Ant Colony Optimization (ACO) applied to the Capacitated Vehicle Routing Problem (CVRP). It takes as inputs distance_matrix (np.ndarray), coordinates (np.ndarray), demands (np.ndarray), and capacity (int), and outputs a heuristic_matrix of the same shape, where larger values indicate that the corresponding edge is more deserving of priority selection in path construction. The default implementation adopts $\eta_{ij} = 1/d_{ij}$, and sets the diagonal elements to 0; serving as a baseline, this can be replaced with novel heuristics integrating multiple factors.

- **JSSP Construct.** This template targets the online construction phase of job shop scheduling and defines the function determine_next_operation(current_status, feasible_operations). The input current_status includes the available time list of each machine (machine_status), the available time list of each job (job_status), and the set of currently schedulable operations (feasible_operations), where each element is a tuple of (job_id, machine_id, processing_time). The default implementation adopts a greedy strategy, directly selecting the operation with the shortest processing_time; serving as a baseline, this can be replaced with novel heuristics integrating multiple factors.

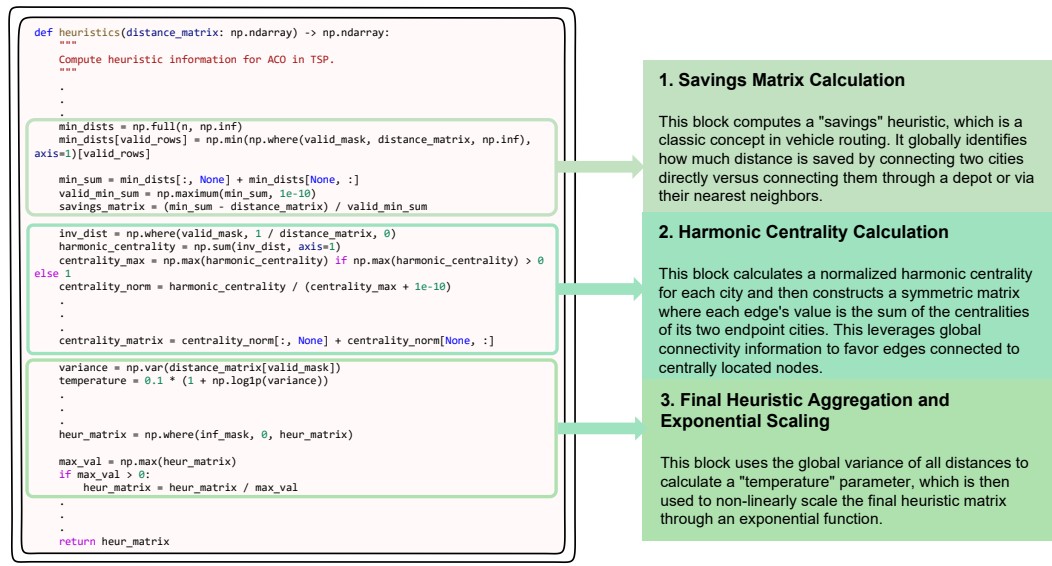

Figure 17: Global Information Analysis of a Heuristic Generated by TPD-AHD in the TSP Construct Framework.

Figure 18: Global Information Analysis of a Heuristic Generated by TPD-AHD in the TSP ACO Framework.

```python
def select_next_city(state):
    """
    Design a novel algorithm to select the next city in each step.
    """
    .
    .
    .
    if len(tour) % 10 == 0:
        centroid = np.mean(instance[tour], axis=0)
        max_dist = -1
        selected_city = unvisited[0]
        for city in unvisited:
            city_coords = instance[city]
            dist = np.linalg.norm(city_coords - centroid)
            if dist > max_dist:
                max_dist = dist
                selected_city = city
        return selected_city

    regrets = []
    min_costs = []
    candidates_info = []

    for candidate in unvisited:
        min_insertion_cost = float('inf')
        second_min_insertion_cost = float('inf')

        for i in range(len(tour)):
            city_a = tour[i]
            city_b = tour[(i + 1) % len(tour)]

            insertion_cost = (distance_matrix[city_a][candidate] +
                              distance_matrix[candidate][city_b] -
                              distance_matrix[city_a][city_b])

            if insertion_cost < min_insertion_cost:
                second_min_insertion_cost = min_insertion_cost
                min_insertion_cost = insertion_cost
            elif insertion_cost < second_min_insertion_cost:
                second_min_insertion_cost = insertion_cost

        regret = second_min_insertion_cost - min_insertion_cost
        regrets.append(regret)
        min_costs.append(min_insertion_cost)
        candidates_info.append((candidate, regret, min_insertion_cost))
    .
    .
    .
    for idx in top_indices:
        candidate = unvisited[idx]
        simulated_tour = tour.copy()
        best_insertion_pos = 0
        best_insertion_cost = float('inf')

        for i in range(len(tour)):
            city_a = tour[i]
            city_b = tour[(i + 1) % len(tour)]
            insertion_cost = (distance_matrix[city_a][candidate] +
                              distance_matrix[candidate][city_b] -
                              distance_matrix[city_a][city_b])
            if insertion_cost < best_insertion_cost:
                best_insertion_cost = insertion_cost
                best_insertion_pos = i

        simulated_tour.insert(best_insertion_pos + 1, candidate)
        remaining_unvisited = [u for u in unvisited if u != candidate]

        if len(remaining_unvisited) > 0:
            future_regrets = []
            for future_candidate in remaining_unvisited[:5]:
                future_min_cost = float('inf')
                future_second_min = float('inf')

                for j in range(len(simulated_tour)):
                    city_c = simulated_tour[j]
                    city_d = simulated_tour[(j + 1) % len(simulated_tour)]
                    future_cost = (distance_matrix[city_c][future_candidate] +
                                   distance_matrix[future_candidate][city_d] -
                                   distance_matrix[city_c][city_d])

                    if future_cost < future_min_cost:
                        future_second_min = future_min_cost
                        future_min_cost = future_cost
                    elif future_cost < future_second_min:
                        future_second_min = future_cost

                if future_second_min != float('inf'):
                    future_regrets.append(future_second_min - future_min_cost)

            flexibility = np.mean(future_regrets) if future_regrets else 0
            current_regret = regrets[idx]

            combined_score = 0.7 * current_regret + 0.3 * flexibility

            if combined_score > best_flexibility:
                best_flexibility = combined_score
                best_candidate = candidate
    .
    .
    .
    return unvisited[selected_index]
```

**1. Centroid-Based Selection**

This block's purpose is to periodically (every 10th step) introduce a long-term, diversity-seeking move into the construction process by selecting the city that is farthest from the geometric center (centroid) of all cities visited so far.

**2. Regret Calculation**

The purpose of this block is to compute a "regret" value for each unvisited city, which quantifies the opportunity cost of not inserting that city at its best possible position in the current complete tour.

**3. Look-Ahead Flexibility Evaluation**

This block implements a look-ahead mechanism to evaluate the long-term impact of selecting a high-regret candidate. It aims to choose the candidate that not only has a high immediate regret but also preserves good insertion options for the remaining cities.

Figure 19: Global Information Analysis of a Heuristic Generated by TPD-AHD in the TSP Random Insertion Framework.

```
TSP_Construct

template_program = '''
import numpy as np
def select_next_node(current_node: int, destination_node: int, unvisited_nodes: np.ndarray,
distance_matrix: np.ndarray) -> int:
    """
    Design a novel algorithm to select the next node in each step.

    Args:
    current_node: ID of the current node.
    destination_node: ID of the destination node.
    unvisited_nodes: Array of IDs of unvisited nodes.
    distance_matrix: Distance matrix of nodes.

    Return:
    ID of the next node to visit.
    """
    next_node = unvisited_nodes[0]

    return next_node
'''

task_description = "Given a set of nodes with their coordinates, you need to find the
shortest route that visits each node once and returns to the starting node. \
The task can be solved step-by-step by starting from the current node and iteratively
choosing the next node. Help me design a novel algorithm that is different from the
algorithms in literature to select the next node in each step."
```

Figure 20: The template and task description of TSP Construct.

```
TSP_ACO

template_program = '''
import numpy as np

def heuristics(distance_matrix: np.ndarray) -> np.ndarray:
    """
    The `heuristics` function takes as input a distance matrix, and returns prior
indicators of how promising it is to include each edge in a solution. The return is of the
same shape as the input.

    Args:
        distance_matrix (np.ndarray): A square matrix of pairwise distances between cities.

    Returns:
        np.ndarray: A heuristic matrix of the same shape, where larger values
                    indicate higher desirability for selecting an edge.
    """

    return 1 / distance_matrix

'''

task_description = '''
Design a heuristic function for TSP using Ant Colony Optimization (ACO). The function
takes a distance matrix as input and returns a matrix of the same shape, where higher
values indicate more promising edges. Define a novel way to evaluate edge desirability,
different from standard inverse-distance heuristics. Try sparsifying the matrix by setting
unpromising elements to zero. Try combining various factors to determine how promising it
is to select an edge.
'''
```

Figure 21: The template and task description of TSP ACO.

```
TSP_Random_Insertion_City

template_program = '''
import numpy as np
def select_next_city(state):
    """
    Design a novel algorithm to select the next city in each step.

    Args:
    state = {
    "instance": np.array with shape (n, 2),
    "tour": current tour (in order) list[int],
    "unvisited": list of remaining unvisited cities list[int],
    "distance_matrix": np.array (n,n)
    }

    Return:
    Should return a city index (int) from unvisited
    """

    last_city = state["tour"][-1]
    unvisited = state["unvisited"]
    dist = state["distance_matrix"][last_city][unvisited]
    return int(unvisited[np.argmin(dist)])
'''

task_description = "Given the current partial tour and the set of unvisited cities, design
a novel heuristic that selects the next city to insert into the tour. Your method should
rely only on the available state information and must differ from standard approaches such
as nearest, farthest, or random selection."
```

Figure 22: The template and task description of TSP Random Insertion.

```
CVRP_Construct

template_program = '''
import numpy as np
def select_next_node(current_node: int, depot: int, unvisited_nodes: np.ndarray,
rest_capacity: np.ndarray, demands: np.ndarray, distance_matrix: np.ndarray) -> int:
    """Design a novel algorithm to select the next node in each step.
    Args:
        current_node: ID of the current node.
        depot: ID of the depot.
        unvisited_nodes: Array of IDs of unvisited nodes.
        rest_capacity: rest capacity of vehicle
        demands: demands of nodes
        distance_matrix: Distance matrix of nodes.
    Return:
        ID of the next node to visit.
    """
    best_score = -1
    next_node = -1

    for node in unvisited_nodes:
        demand = demands[node]
        distance = distance_matrix[current_node][node]

        if demand <= rest_capacity:
            score = demand / distance if distance > 0 else float('inf')  # Avoid division
by zero
            if score > best_score:
                best_score = score
                next_node = node

    return next_node
'''

task_description = """
Given a set of customers and a fleet of vehicles with limited capacity,
the task is to design a novel algorithm to select the next node in each step,
with the objective of minimizing the total cost.
"""
```

Figure 23: The template and task description of CVRP Construct.

```
CVRP_ACO

template_program = '''
import numpy as np
def heuristics(distance_matrix: np.ndarray, coordinates: np.ndarray, demands: np.ndarray,
capacity: int) -> np.ndarray:
    """
    Compute heuristic information for Ant Colony Optimization (ACO) in Capacitated Vehicle
Routing Problem (CVRP).

    Args:
        distance_matrix (np.ndarray): A square matrix  of pairwise distances between nodes.
Nodes include 1 depot (index 0) and n-1 customers.
        coordinates (np.ndarray): An array of node coordinates, where coordinates[0] is
the depot.
        demands (np.ndarray): An array (shape: n) of demands for each node. demands[0] = 0
(depot has no demand), others are customer demands.
        capacity (int): The maximum load capacity of each vehicle.

    Returns:
        np.ndarray: A heuristic matrix where larger values indicate higher desirability
                    for selecting an edge in CVRP solution construction. Should consider
CVRP constraints
                    (e.g., vehicle capacity, demand satisfaction) and practical routing
efficiency.
    """
    with np.errstate(divide='ignore', invalid='ignore'):
        heuristic = 1 / distance_matrix

    np.fill_diagonal(heuristic, 0)
    return heuristic
'''

task_description = '''
Design a heuristic function for Capacitated Vehicle Routing Problem (CVRP) using Ant
Colony Optimization (ACO). The function takes a distance matrix, node coordinates,
customer demands, and vehicle capacity as inputs, and returns a matrix of the same shape
as the distance matrix, where higher values indicate more promising edges. Define a novel
way to evaluate edge desirability, different from standard inverse-distance heuristics.
You may combine multiple CVRP-specific factors (e.g., demand-capacity fit, proximity) and
optionally sparsify the matrix by setting weak edges to zero.
'''
```

Figure 24: The template and task description of CVRP ACO.

```
JSSP_Construct

template_program = '''
import numpy as np

def determine_next_operation(current_status, feasible_operations):
    """
    Determine the next operation to schedule based on a greedy heuristic.

    Args:
        current_status: A dictionary with two lists:
            - 'machine_status': a list where index = machine_id, value = the time the
machine is available
            - 'job_status': a list where index = job_id, value = the time the job is
available
        feasible_operations: A list of feasible operations, each represented as a tuple
            (job_id, machine_id, processing_time).

    Returns:
        The next operation to schedule, represented as a tuple (job_id, machine_id,
processing_time).
    """
    # Simple greedy heuristic: choose the operation with the shortest processing time
    next_operation = min(feasible_operations, key=lambda x: x[2])
    return next_operation
'''

task_description = '''
Given jobs and machines, schedule jobs on machines to minimize the total makespan. Design
an algorithm to select the next operation in each step.
Important:
- Be careful when computing any ratio, priority, or weight that involves machine or job
availability times.
- The machine or job status may initially be zero, so make sure to handle or avoid
division by zero.
'''
```

Figure 25: The template and task description of JSSP Construct.

1944
1945
1946
1947
1948
1949
1950
1951
1952
1953
1954
1955
1956
1957
1958
1959
1960
1961
1962
1963
1964
1965
1966
1967
1968
1969
1970
1971
1972
1973
1974
1975
1976
1977
1978
1979
1980
1981
1982
1983
1984
1985
1986
1987
1988
1989
1990
1991
1992
1993
1994
1995
1996
1997

### Heuristic 1 (Obj Score: -6.2677441099390485)

```python
def select_next_node(current_node: int, destination_node: int, unvisited_nodes: np.ndarray, distance_matrix:
np.ndarray) -> int:
    if len(unvisited_nodes) == 1:
        return unvisited_nodes[0]

    best_score = float('inf')
    next_node = -1
    total_nodes = len(unvisited_nodes)
    visited_ratio = 1 - (len(unvisited_nodes) / total_nodes)
    recent_history = []

    dest_distances = distance_matrix[unvisited_nodes, destination_node]
    local_densities = np.mean(distance_matrix[unvisited_nodes][:, unvisited_nodes], axis=1)

    remaining_coords = distance_matrix[unvisited_nodes]
    if len(remaining_coords) > 1:
        remaining_center = np.mean(remaining_coords, axis=0)
        node_distances_to_center = np.linalg.norm(remaining_coords - remaining_center, axis=1)
        local_densities = np.clip(local_densities, 0.1, None) * (1 + node_distances_to_center)

    if total_nodes > 3:
        max_dist = np.max(distance_matrix[unvisited_nodes][:, unvisited_nodes])
        min_dist = np.min(distance_matrix[unvisited_nodes][:,
unvisited_nodes][distance_matrix[unvisited_nodes][:, unvisited_nodes] > 0])
        path_deviation = (max_dist - min_dist) / (max_dist + 1e-5)
    else:
        path_deviation = 0.5

    if visited_ratio < 0.15 + path_deviation * 0.1:
        heuristic_phase = 'early'
    elif visited_ratio < 0.6 + path_deviation * 0.2:
        heuristic_phase = 'mid'
    else:
        heuristic_phase = 'late'

    current_coords = distance_matrix[current_node]
    quadrant_counts = [0, 0, 0, 0]
    for node in unvisited_nodes:
        coords = distance_matrix[node]
        dx = coords[0] - current_coords[0]
        dy = coords[1] - current_coords[1]
        quadrant = int(np.arctan2(dy, dx) / (np.pi / 2)) % 4
        quadrant_counts[quadrant] += 1

    quadrant_weights = [1 - 0.05 * (count / (len(unvisited_nodes) + 1)) for count in quadrant_counts]

    node_degrees = np.sum(distance_matrix[unvisited_nodes][:, unvisited_nodes] > 0, axis=1)

    if heuristic_phase == 'early':
        connectivity_scores = []
        for node in unvisited_nodes:
            reachable_nodes = np.where(distance_matrix[node][unvisited_nodes] > 0)[0]
            second_hop = []
            for r_node in reachable_nodes[:3]:
                second_hop.extend(np.where(distance_matrix[unvisited_nodes[r_node]][unvisited_nodes] > 0)[0])
            unique_reachable = len(set(reachable_nodes) | set(second_hop))
            connectivity_scores.append(unique_reachable / len(unvisited_nodes))
    else:
        connectivity_scores = [1.0] * len(unvisited_nodes)
    heading_angle = 0
    if len(recent_history) >= 2:
        prev_coords = distance_matrix[recent_history[-1]]
        prev_prev_coords = distance_matrix[recent_history[-2]]
        dx_h = prev_coords[0] - prev_prev_coords[0]
        dy_h = prev_coords[1] - prev_prev_coords[1]
        heading_angle = np.arctan2(dy_h, dx_h)

    for idx, node in enumerate(unvisited_nodes):
        dist_to_candidate = distance_matrix[current_node][node]
        dist_candidate_to_dest = dest_distances[idx]
        density = local_densities[idx]
        degree = node_degrees[idx]
        connectivity = connectivity_scores[idx]
        is_backtracking = False
        if len(recent_history) >= 2:
            prev_node = recent_history[-1]
            prev_prev_node = recent_history[-2]
            direct_path = distance_matrix[prev_prev_node][node]
            detour_path = distance_matrix[prev_prev_node][prev_node] + distance_matrix[prev_node][node]
            if direct_path > 0 and detour_path / direct_path > 1.5 and visited_ratio > 0.1:
                is_backtracking = True
        if is_backtracking:
            continue
        coords = distance_matrix[node]
        dx = coords[0] - current_coords[0]
        dy = coords[1] - current_coords[1]
        candidate_angle = np.arctan2(dy, dx)
        deviation = abs(candidate_angle - heading_angle)
        deviation = min(deviation, 2 * np.pi - deviation)
        deviation_deg = np.degrees(deviation)
        angle_penalty = 1.0
        if deviation_deg > 120:
            angle_penalty = 1.2

        if heuristic_phase == 'early':
            composite_distance = dist_to_candidate * (1 - 0.5 * connectivity)
            normalized_score = composite_distance / (density + 1e-5)
            score = normalized_score / (degree + 1)
        elif heuristic_phase == 'late':
            if degree < 2:
                continue
            lookahead_k = min(5, len(unvisited_nodes))
            nearest_unvisited =
unvisited_nodes[np.argsort(distance_matrix[node][unvisited_nodes])[:lookahead_k]]
            lookahead_scores = []
            for neighbor in nearest_unvisited:
                if neighbor != destination_node:
                    dist_candidate_to_neighbor = distance_matrix[node][neighbor]
                    neighbor_dest = distance_matrix[neighbor][destination_node]
                    lookahead_score = dist_candidate_to_neighbor + 0.2 * neighbor_dest
                    lookahead_scores.append(lookahead_score)
            lookahead_component = 0.4 * np.min(lookahead_scores) if lookahead_scores else 0
            score = dist_to_candidate + dist_candidate_to_dest * 0.9 + lookahead_component
        else:
            local_progress = (local_densities.mean() - density) / (local_densities.std() + 1e-5)
            dynamic_weight = 0.4 + (visited_ratio - 0.2) * 0.6 + local_progress * 0.2
            dynamic_weight = max(0.2, min(0.8, dynamic_weight))

            score = dist_to_candidate + dynamic_weight * dist_candidate_to_dest

            avg_local_density = np.mean(local_densities)
            lookahead_depth = max(2, min(3, int(avg_local_density / (density + 1e-5) * 3)))
            nearest_unvisited =
unvisited_nodes[np.argsort(distance_matrix[node][unvisited_nodes])[:lookahead_depth]]
            lookahead_scores = []
            for neighbor in nearest_unvisited:
                if neighbor != destination_node:
                    dist_candidate_to_neighbor = distance_matrix[node][neighbor]
                    neighbor_dest = distance_matrix[neighbor][destination_node]
                    lookahead_score = dist_candidate_to_neighbor + 0.3 * neighbor_dest
                    lookahead_scores.append(lookahead_score)
            if lookahead_scores:
                lookahead_scores = [s * (1 / (i + 1)) for i, s in enumerate(sorted(lookahead_scores))]
                score += 0.2 * min(lookahead_scores)

        quadrant = int(candidate_angle / (np.pi / 2)) % 4
        quadrant_score = quadrant_weights[quadrant]
        if len(unvisited_nodes) > 5:
            sparse_threshold = np.mean(quadrant_counts) * 0.5
            if quadrant_counts[quadrant] < sparse_threshold:
                quadrant_score = 1 + (1 - quadrant_score)

        score *= quadrant_score
        score *= angle_penalty

        perturbation = score * np.random.uniform(0, 0.005 if idx == 0 else 0.02)
        score += perturbation

        if score < best_score:
            best_score = score
            next_node = node
            recent_history.append(node)
            if len(recent_history) > 5:
                recent_history.pop(0)

    return next_node
```

➕

### Heuristic 2 (Obj Score: -6.411359052660979)

```python
def select_next_node(current_node: int, destination_node: int, unvisited_nodes: np.ndarray, distance_matrix:
np.ndarray) -> int:
    if len(unvisited_nodes) == 1:
        return unvisited_nodes[0]
    next_node = -1
    best_score = float('inf')
    total_nodes = len(unvisited_nodes)
    visited_ratio = 1 - (len(unvisited_nodes) / total_nodes)
    recent_history = []
    dest_distances = distance_matrix[unvisited_nodes, destination_node]
    local_densities = np.mean(distance_matrix[unvisited_nodes][:, unvisited_nodes], axis=1)
    remaining_coords = distance_matrix[unvisited_nodes]
    if len(remaining_coords) > 1:
        remaining_center = np.mean(remaining_coords, axis=0)
        node_distances_to_center = np.linalg.norm(remaining_coords - remaining_center, axis=1)
        local_densities = np.clip(local_densities, 0.1, None) * (1 + node_distances_to_center)
    if total_nodes > 3:
        max_dist = np.max(distance_matrix[unvisited_nodes][:, unvisited_nodes])
        min_dist = np.min(distance_matrix[unvisited_nodes][:,
unvisited_nodes][distance_matrix[unvisited_nodes][:, unvisited_nodes] > 0])
        path_deviation = (max_dist - min_dist) / (max_dist + 1e-5)
    else:
        path_deviation = 0.5
    # Directional momentum vector calculation
    current_coords = distance_matrix[current_node]
    dest_coords = distance_matrix[destination_node]
    directional_vector = dest_coords[:2] - current_coords[:2]
    dir_norm = np.linalg.norm(directional_vector)
    directional_momentum = directional_vector / dir_norm if dir_norm > 0 else np.zeros(2)
    # Sliding window for backtracking detection
    window_size = 5
    historical_angles = []
    historical_positions = []
    # Continuous phase interpolation
    early_phase_weight = 1 - np.clip(visited_ratio / 0.2, 0, 1)
    mid_phase_weight = np.clip(visited_ratio / 0.6, 0, 1) * np.clip((1 - visited_ratio) / 0.4, 0, 1)
    late_phase_weight = np.clip((visited_ratio - 0.8) / 0.2, 0, 1)
    # Quadrant biasing with directional momentum
    quadrant_counts = [0, 0, 0, 0]
    for node in unvisited_nodes:
        coords = distance_matrix[node]
        dx = coords[0] - current_coords[0]
        dy = coords[1] - current_coords[1]
        quadrant = int(np.arctan2(dy, dx) / (np.pi / 2)) % 4
        quadrant_counts[quadrant] += 1

    quadrant_weights = [1 - 0.05 * (count / (len(unvisited_nodes) + 1)) for count in quadrant_counts]
    # Node importance via connectivity potential
    node_potentials = []
    for idx, node in enumerate(unvisited_nodes):
        dist_to_others = distance_matrix[node][unvisited_nodes]
        valid_dists = dist_to_others[dist_to_others > 0]
        if len(valid_dists) > 0:
            avg_connectivity = 1 / np.mean(valid_dists)
        else:
            avg_connectivity = 0
        node_potentials.append(avg_connectivity)
    node_potentials = np.array(node_potentials)
    node_potentials = (node_potentials - np.min(node_potentials)) / (np.max(node_potentials) -
np.min(node_potentials) + 1e-5)

    # Historical path context
    historical_distances = []
    for node in unvisited_nodes:
        if len(recent_history) >= window_size:
            window_nodes = recent_history[-window_size:]
            if len(window_nodes) >= 2:
                path_x = [distance_matrix[n][0] for n in window_nodes]
                path_y = [distance_matrix[n][1] for n in window_nodes]
                path_length = sum(distance_matrix[window_nodes[i]][window_nodes[i+1]] for i in
range(len(window_nodes)-1))
                straight_line = distance_matrix[window_nodes[0]][node]
                if straight_line > 0:
                    inefficiency = path_length / straight_line
                    if inefficiency > 1.7:
                        penalty_angle = np.arccos(np.clip(
                            np.dot(directional_momentum, [distance_matrix[node][0] - current_coords[0],
                                distance_matrix[node][1] - current_coords[1]]) /
                            (np.linalg.norm([distance_matrix[node][0] - current_coords[0],
                                distance_matrix[node][1] - current_coords[1]]) + 1e-5), -1, 1))
                        quadrant_weights[quadrant] *= (1 + penalty_angle / np.pi)
            historical_distances.append(path_length / (straight_line + 1e-5) if 'path_length' in locals() and
'straight_line' in locals() and straight_line > 0 else 1.0)
    # Predictive cost estimation
    predictive_costs = []
    for idx, node in enumerate(unvisited_nodes):
        simulated_path = []
        current_sim_node = node
        visited_sim = set([current_sim_node])
        while len(visited_sim) < min(5, len(unvisited_nodes)):
            candidates = [n for n in unvisited_nodes if n not in visited_sim]
            if not candidates:
                break
            next_sim = min(candidates, key=lambda x: distance_matrix[current_sim_node][x] + 0.5 *
distance_matrix[x][destination_node])
            simulated_path.append(next_sim)
            visited_sim.add(next_sim)
            current_sim_node = next_sim
        predictive_cost = sum(distance_matrix[simulated_path[i]][simulated_path[i+1]] for i in
range(len(simulated_path)-1)) if len(simulated_path) > 1 else 0
        predictive_costs.append(predictive_cost)
    predictive_costs = np.array(predictive_costs)
    predictive_costs = (predictive_costs - np.min(predictive_costs)) / (np.max(predictive_costs) -
np.min(predictive_costs) + 1e-5)
    # Node scoring with enhancements
    for idx, node in enumerate(unvisited_nodes):
        dist_to_candidate = distance_matrix[current_node][node]
        dist_candidate_to_dest = dest_distances[idx]
        density = local_densities[idx]
        potential = node_potentials[idx]
        predictive_cost = predictive_costs[idx]
        coords = distance_matrix[node]
        dx = coords[0] - current_coords[0]
        dy = coords[1] - current_coords[1]
        angle_to_candidate = np.arctan2(dy, dx)
        quadrant = int(angle_to_candidate / (np.pi / 2)) % 4
        quadrant_score = quadrant_weights[quadrant]
        # Directional momentum alignment
        candidate_vector = np.array([dx, dy])
        candidate_norm = np.linalg.norm(candidate_vector)
        if candidate_norm > 0 and dir_norm > 0:
            momentum_alignment = np.dot(directional_momentum, candidate_vector / candidate_norm)
            momentum_factor = 1 - (np.arccos(np.clip(momentum_alignment, -1, 1)) / np.pi)
        else:
            momentum_factor = 0.5
        # Adaptive quadrant weighting
        if len(unvisited_nodes) > 5:
            sparse_threshold = np.mean(quadrant_counts) * 0.5
            if quadrant_counts[quadrant] < sparse_threshold:
                quadrant_score = 1 + (1 - quadrant_score)

        # Phase blended scoring
        early_score = dist_to_candidate / (density + 1e-5) * (1 - 0.3 * potential)
        mid_score = dist_to_candidate + (0.4 + 0.6 * visited_ratio) * dist_candidate_to_dest
        late_score = dist_to_candidate + 0.9 * dist_candidate_to_dest + 0.4 * predictive_cost

        phase_score = (
            early_phase_weight * early_score +
            mid_phase_weight * mid_score +
            late_phase_weight * late_score
        )

        # Combine with directional momentum and quadrant bias
        final_score = phase_score * (0.7 + 0.3 * momentum_factor) * quadrant_score
        # Tie-breaking perturbation
        perturbation = final_score * np.random.uniform(0, 0.005 if idx == 0 else 0.02)
        final_score += perturbation
        if final_score < best_score:
            best_score = final_score
            next_node = node
            recent_history.append(node)
            if len(recent_history) > 7:
                recent_history.pop(0)

    return next_node
```

**The losses:**

# Code Evaluation: Chosen vs Rejected

## Three Strongest Advantages of the Chosen Code

### 1. **Strategic Phased Approach with Contextual Adaptation**
The chosen code implements a clear phase-based strategy (early/mid/late) that adapts the node selection criteria based on the proportion of visited nodes. This allows for specialized behaviors at different stages of the traversal process, optimizing for exploration in early phases, balance in mid-phases, and directness to destination in late phases. The adaptation to path deviation further enhances this contextual awareness.

### 2. **Sophisticated Connectivity Analysis**
The chosen code demonstrates superior connectivity analysis by incorporating multi-hop connectivity assessment in early phases. This provides a more comprehensive understanding of network topology than simple direct connections, allowing the algorithm to prioritize nodes that offer better overall network access, not just immediate proximity.

### 3. **Effective Backtracking Prevention with Temporal Awareness**
The implementation of a history-based backtracking detection mechanism is well-designed and effective. By analyzing the detour ratio compared to direct paths and combining this with the visited ratio, the code intelligently prevents inefficient revisiting of areas while allowing necessary detours when appropriate. This temporal awareness is crucial for path optimization.

## Three Most Significant Weaknesses of the Rejected Code

### 1. **Overly Complex and Coupled Scoring Mechanism**
The rejected code combines too many factors (historical distances, predictive costs, phase weights, quadrant biases, momentum alignment) in its scoring mechanism without clear prioritization. This results in a system where it's difficult to understand the relative importance of different factors and how they interact, reducing maintainability and interpretability.

### 2. **Inefficient Predictive Cost Estimation**
The predictive cost estimation, while conceptually valuable, is computationally expensive and potentially unreliable. The simulation of hypothetical paths with up to 5 nodes doesn't accurately represent the complete path and introduces significant computational overhead without clear benefits to the final path quality.

### 3. **Poorly Integrated Historical Context**
The historical path context implementation is overly complex and loosely integrated with the main scoring system. The window-based inefficiency calculation and its impact on quadrant weights lacks smooth transitions and could lead to abrupt changes in behavior, potentially causing erratic path selection.

## Why the Chosen Code Was Selected

The chosen code strikes a better balance between sophisticated path analysis and practical implementation. It maintains a clear logical structure with well-defined phases that guide the search strategy. Its connectivity analysis provides meaningful topological insight without excessive computational burden. The backtracking prevention mechanism is effective while maintaining flexibility.

The rejected code, while containing some valuable concepts like predictive estimation and momentum alignment, combines too many loosely integrated factors that create an overly complex system. Its complexity doesn't necessarily lead to better decisions and introduces potential instability and maintainability issues.

The chosen code represents a more principled approach where each component clearly contributes to the overall objective of finding an efficient path from the current node to the destination, while navigating the network of unvisited nodes effectively.

**The Gradients:**

Here are **five targeted, actionable suggestions** for improving the chosen code, based on its evaluation advantages and to further enhance its performance and maintainability:

---

### 1. **Introduce Smooth Transitions Between Phases to Avoid Abrupt Strategy Shifts**
Currently, phase transitions (early → mid → late) are based on hard thresholds of `visited_ratio`. This may cause sudden changes in behavior, potentially leading to suboptimal node selections at boundary conditions.
**Suggestion**:
Replace the current discrete phase logic with **fuzzy phase interpolation**. Calculate a weighted combination of early, mid, and late-phase scoring functions based on continuous phase weights derived from `visited_ratio`, ensuring smoother behavioral transitions.

---

### 2. **Optimize the Quadrant Weighting Mechanism for Better Spatial Balance**
While quadrant-based directional preference improves exploration, the current method of computing quadrant weights based on static node counts may lead to overcorrection in dense or sparse regions.
**Suggestion**:
Refine quadrant weighting by incorporating **spatial distribution density** using kernel density estimation (KDE) over node coordinates within each quadrant. This can yield more meaningful directional preferences and reduce erratic shifts in direction.

---

### 3. **Refactor the Composite Scoring Function for Better Interpretability and Maintainability**
The scoring function combines many components (distance, density, quadrant, angle penalty, etc.) in a way that makes it difficult to analyze the contribution of each factor.
**Suggestion**:
**Normalize and weight each component explicitly**, and expose these weights as configurable parameters (e.g., via a dictionary). This improves transparency and enables easier tuning without altering logic, promoting reuse across different problem instances.

---

### 4. **Improve the Backtracking Detection Mechanism with Temporal Decay and Path Context**
The current backtracking logic uses only the last two nodes and a fixed threshold. It may miss longer-term inefficiencies or allow subtle backtracking loops.
**Suggestion**:
Enhance the `recent_history` tracking by incorporating **exponential decay of node revisit penalties** over time, and introduce **angle deviation thresholds** for path coherence. This allows for more adaptive backtracking detection that considers path momentum.

---

### 5. **Precompute or Cache Connectivity and Density Metrics to Reduce Redundant Computation**
Multiple components (density, connectivity, lookahead) compute similar distance and neighborhood metrics repeatedly across iterations.
**Suggestion**:
**Cache or precompute key metrics** such as local density, connectivity, and nearest neighbor lists at the beginning of each selection round. This will reduce redundant computations and improve performance, especially for large graphs.

---

These improvements build upon the chosen code's strengths—its phase-based strategy, connectivity awareness, and backtracking prevention—while refining its decision logic and efficiency.

**Heuristic 3 (Obj Score: -6.2454075785468355)**

```python
def select_next_node(current_node: int, destination_node: int, unvisited_nodes:
np.ndarray, distance_matrix: np.ndarray) -> int:
    if len(unvisited_nodes) == 1:
        return unvisited_nodes[0]
    best_score = float('inf')
    next_node = -1
    total_nodes = len(unvisited_nodes)
    visited_ratio = 1 - (len(unvisited_nodes) / total_nodes)
    recent_history = []
    dest_distances = distance_matrix[unvisited_nodes, destination_node]
    local_densities = np.mean(distance_matrix[unvisited_nodes][:, unvisited_nodes],
axis=1)
    remaining_coords = distance_matrix[unvisited_nodes]
    if len(remaining_coords) > 1:
        remaining_center = np.mean(remaining_coords, axis=0)
        node_distances_to_center = np.linalg.norm(remaining_coords - remaining_center,
axis=1)
        local_densities = np.clip(local_densities, 0.1, None) * (1 +
node_distances_to_center)
    if total_nodes > 3:
        max_dist = np.max(distance_matrix[unvisited_nodes][:, unvisited_nodes])
        min_dist = np.min(distance_matrix[unvisited_nodes][:,
unvisited_nodes][distance_matrix[unvisited_nodes][:, unvisited_nodes] > 0])
        path_deviation = (max_dist - min_dist) / (max_dist + 1e-5)
    else:
        path_deviation = 0.5
    phase_weights = {
        'early': max(0, min(1, 0.6 - visited_ratio * 2)),
        'mid': max(0, min(1, visited_ratio * 2 - 0.4)),
        'late': max(0, min(1, visited_ratio - 0.6))
    }
    current_coords = distance_matrix[current_node]
    coords_x = remaining_coords[:, 0]
    coords_y = remaining_coords[:, 1]
    dx = coords_x - current_coords[0]
    dy = coords_y - current_coords[1]
    angles = np.arctan2(dy, dx)
    quadrants = (angles / (np.pi / 2)).astype(int) % 4
    unique_quad, counts = np.unique(quadrants, return_counts=True)
    quadrant_counts = np.zeros(4)
    for q, c in zip(unique_quad, counts):
        quadrant_counts[q] = c
    x_coords = coords_x
    y_coords = coords_y
    kde_values = np.zeros(len(x_coords))
    for i in range(len(x_coords)):
        dists = np.sqrt((x_coords - x_coords[i]) ** 2 + (y_coords - y_coords[i]) ** 2)
        kde = np.sum(np.exp(-dists ** 2 / 2))
        kde_values[i] += kde
    kde_norm = (kde_values - np.min(kde_values)) / (np.max(kde_values) -
np.min(kde_values) + 1e-5)
    quadrant_kde = np.zeros(4)
    for q in range(4):
        mask = quadrants == q
        if np.any(mask):
            quadrant_kde[q] = np.mean(kde_values[mask])
    quadrant_kde_norm = (quadrant_kde - np.min(quadrant_kde)) / (np.max(quadrant_kde) -
np.min(quadrant_kde) + 1e-5)
    quadrant_weights = 1 - 0.05 * (quadrant_kde_norm / (np.sum(quadrant_kde_norm) +
1e-5))
    node_degrees = np.sum(distance_matrix[unvisited_nodes][:, unvisited_nodes] > 0,
axis=1)
    connectivity_scores = np.zeros(len(unvisited_nodes))
    for idx, node in enumerate(unvisited_nodes):
        reachable = np.where(distance_matrix[node][unvisited_nodes] > 0)[0]
        second_hop = []
        for r_idx in reachable[:3]:
            second_hop.extend(np.where(distance_matrix[unvisited_nodes[r_idx]][unvisited_nodes] >
0)[0])
        unique_reachable = len(set(reachable) | set(second_hop))
        connectivity_scores[idx] = unique_reachable / len(unvisited_nodes)
    connectivity_norm = (connectivity_scores - np.min(connectivity_scores)) /
(np.max(connectivity_scores) - np.min(connectivity_scores) + 1e-5)
    heading_angle = 0
    if len(recent_history) >= 2:
        prev_coords = distance_matrix[recent_history[-1]]
        prev_prev_coords = distance_matrix[recent_history[-2]]
        dx_h = prev_coords[0] - prev_prev_coords[0]
        dy_h = prev_coords[1] - prev_prev_coords[1]
        heading_angle = np.arctan2(dy_h, dx_h)
    phase_params = {
        'early': {'connectivity_weight': 0.3, 'density_weight': 0.2, 'distance_weight':
0.5},
        'mid': {'angle_weight': 0.2, 'progress_weight': 0.3, 'lookahead_weight': 0.5},
        'late': {'dest_weight': 0.7, 'degree_weight': 0.3}
    }
```

```python
    for idx, node in enumerate(unvisited_nodes):
        dist_to_candidate = distance_matrix[current_node][node]
        dist_candidate_to_dest = dest_distances[idx]
        density = local_densities[idx]
        degree = node_degrees[idx]
        connectivity = connectivity_scores[idx]
        is_backtracking = False
        if len(recent_history) >= 2:
            prev_node = recent_history[-1]
            prev_prev_node = recent_history[-2]
            direct_path = distance_matrix[prev_node][node]
            detour_path = distance_matrix[prev_prev_node][prev_node] +
distance_matrix[prev_node][node]
            if direct_path > 0 and detour_path / (direct_path + 1e-5) > 1.5 and
visited_ratio > 0.1:
                is_backtracking = True
        if is_backtracking:
            continue
        coords = distance_matrix[node]
        dx = coords[0] - current_coords[0]
        dy = coords[1] - current_coords[1]
        candidate_angle = np.arctan2(dy, dx)
        deviation = abs(candidate_angle - heading_angle)
        deviation = min(deviation, 2 * np.pi - deviation)
        deviation_deg = np.degrees(deviation)
        angle_penalty = 1.0
        if deviation_deg > 120:
            angle_penalty = 1.2
        early_score = dist_to_candidate * (1 - 0.5 * connectivity) / (density + 1e-5)
        late_score = dist_to_candidate + dist_candidate_to_dest * 0.9
        lookahead_scores = []
        lookahead_k = min(5, len(unvisited_nodes))
        nearest_unvisited =
unvisited_nodes[np.argsort(distance_matrix[node][unvisited_nodes])[:lookahead_k]]
        for neighbor in nearest_unvisited:
            if neighbor != destination_node:
                dist_candidate_to_neighbor = distance_matrix[node][neighbor]
                neighbor_dest = distance_matrix[neighbor][destination_node]
                lookahead_score = dist_candidate_to_neighbor + 0.2 * neighbor_dest
                lookahead_scores.append(lookahead_score)
        if lookahead_scores:
            late_score += 0.4 * np.min(lookahead_scores)
        mid_score = dist_to_candidate
        if len(local_densities) > 1:
            avg_local_density = np.mean(local_densities)
            std_local_density = np.std(local_densities)
            local_progress = (avg_local_density - density) / (std_local_density + 1e-5)
            dynamic_weight = 0.4 + (visited_ratio - 0.2) * 0.6 + local_progress * 0.2
            dynamic_weight = max(0.2, min(0.8, dynamic_weight))
            mid_score += dynamic_weight * dist_candidate_to_dest
        lookahead_depth = max(2, min(3, int(avg_local_density / (density + 1e-5) * 3)))
        nearest_unvisited =
unvisited_nodes[np.argsort(distance_matrix[node][unvisited_nodes])[:lookahead_depth]]
        lookahead_scores = []
        for neighbor in nearest_unvisited:
            if neighbor != destination_node:
                dist_candidate_to_neighbor = distance_matrix[node][neighbor]
                neighbor_dest = distance_matrix[neighbor][destination_node]
                lookahead_score = dist_candidate_to_neighbor + 0.3 * neighbor_dest
                lookahead_scores.append(lookahead_score)
        if lookahead_scores:
            lookahead_scores = [s * (1 / (i + 1)) for i, s in
enumerate(sorted(lookahead_scores))]
            mid_score += 0.2 * min(lookahead_scores)
        quadrant = int(candidate_angle / (np.pi / 2)) % 4
        quad_score = quadrant_weights[quadrant]
        if len(unvisited_nodes) > 5:
            sparse_threshold = np.mean(quadrant_counts) * 0.5
            if quadrant_counts[quadrant] < sparse_threshold:
                quad_score = 1 + (1 - quad_score)
        early_score_norm = early_score / (np.max([early_score, 1]))
        mid_score_norm = mid_score / (np.max([mid_score, 1]))
        late_score_norm = late_score / (np.max([late_score, 1]))
        composite_score = (
            phase_weights['early'] * early_score_norm +
            phase_weights['mid'] * mid_score_norm +
            phase_weights['late'] * late_score_norm
        )
        composite_score *= quad_score
        composite_score *= angle_penalty
        perturbation = composite_score * np.random.uniform(0, 0.005 if idx == 0 else 0.02)
        final_score = composite_score + perturbation
        if final_score < best_score:
            best_score = final_score
            next_node = node
            recent_history.append(node)
            if len(recent_history) > 5:
                recent_history.pop(0)
    return next_node
```

Gradient 5
Gradient 1
Gradient 5
Gradient 2
Gradient 5
Gradient 3
Gradient 4
Gradient 3
Gradient 1

Figure 26: TSP Construct Example 1.

### Heuristic 1 (Obj Score: -6.5607966760939584)

```python
def select_next_node(current_node: int, destination_node: int, unvisited_nodes:
np.ndarray, distance_matrix: np.ndarray) -> int:
    if len(unvisited_nodes) == 1:
        return unvisited_nodes[0]

    best_score = float('inf')
    next_node = -1
    total_nodes = len(unvisited_nodes)
    visited_ratio = 1 - (len(unvisited_nodes) / total_nodes)
    recent_history = list()  # To track recent nodes for backtracking filter

    # Precompute distances to destination and local node density
    dest_distances = distance_matrix[unvisited_nodes, destination_node]
    local_densities = np.mean(distance_matrix[unvisited_nodes][:, unvisited_nodes],
axis=1)

    # Determine phase
    if visited_ratio < 0.2:
        heuristic_phase = 'early'
    elif visited_ratio < 0.7:
        heuristic_phase = 'mid'
    else:
        heuristic_phase = 'late'

    # Estimate node degrees for graph awareness
    node_degrees = np.sum(distance_matrix[unvisited_nodes] > 0, axis=1)

    for idx, node in enumerate(unvisited_nodes):
        dist_to_candidate = distance_matrix[current_node][node]
        dist_candidate_to_dest = dest_distances[idx]
        density = local_densities[idx]
        degree = node_degrees[idx]

        # Skip if node is too close to recent nodes (backtracking-aware filter)
        is_too_close = any(np.linalg.norm(distance_matrix[current_node][n] -
distance_matrix[current_node][node]) < 0.1 for n in recent_history[-5:])
        if is_too_close and visited_ratio > 0.1:
            continue

        # Phase-adaptive scoring logic
        if heuristic_phase == 'early':
            # Early phase: favor well-connected nodes in sparse graphs
            normalized_score = dist_to_candidate / (density + 1e-5)
            score = normalized_score / (degree + 1)
        elif heuristic_phase == 'late':
            # Late phase: deep lookahead with top 5 neighbors
            nearest_unvisited =
unvisited_nodes[np.argsort(distance_matrix[node][unvisited_nodes])[:5]]
            lookahead_scores = []
            for neighbor in nearest_unvisited:
                if neighbor != destination_node:
                    dist_candidate_to_neighbor = distance_matrix[node][neighbor]
                    neighbor_dest = distance_matrix[neighbor][destination_node]
                    lookahead_score = dist_candidate_to_neighbor + 0.2 * neighbor_dest
                    lookahead_scores.append(lookahead_score)
            lookahead_component = 0.4 * np.min(lookahead_scores) if lookahead_scores
else 0
            score = dist_to_candidate + dist_candidate_to_dest * 0.9 +
lookahead_component
        else:  # mid phase
            # Dynamic weighting normalized by local context
            local_progress = (local_densities.mean() - density) /
(local_densities.std() + 1e-5)
            dynamic_weight = 0.4 + (visited_ratio - 0.2) * 0.6 + local_progress * 0.2
            score = dist_to_candidate + max(0.2, min(0.8, dynamic_weight)) *
dist_candidate_to_dest

            # 2-step lookahead with conditional perturbation
            nearest_unvisited =
unvisited_nodes[np.argsort(distance_matrix[node][unvisited_nodes])[:3]]
            lookahead_scores = []
            for neighbor in nearest_unvisited:
                if neighbor != destination_node:
                    dist_candidate_to_neighbor = distance_matrix[node][neighbor]
                    neighbor_dest = distance_matrix[neighbor][destination_node]
                    lookahead_score = dist_candidate_to_neighbor + 0.3 * neighbor_dest
                    lookahead_scores.append(lookahead_score)
            if lookahead_scores:
                score += 0.2 * np.min(lookahead_scores)

        # Local diversity boost with directional awareness
        if idx < len(unvisited_nodes) - 1 and np.isclose(score, best_score, atol=1e-2):
            # Directional perturbation toward under-explored quadrants
            angle = np.arctan2(distance_matrix[node][1] -
distance_matrix[current_node][1],
                               distance_matrix[node][0] -
distance_matrix[current_node][0])
            quadrant = int(angle / (np.pi / 2)) % 4
            quadrant_counts = [0, 0, 0, 0]
            for n in recent_history:
                a = np.arctan2(distance_matrix[n][1] -
distance_matrix[current_node][1],
                               distance_matrix[n][0] -
distance_matrix[current_node][0])
                q = int(a / (np.pi / 2)) % 4
                quadrant_counts[q] += 1
            score *= 1 - 0.05 * (quadrant_counts[quadrant] / (len(recent_history) + 1))

        # Small random perturbation only for tie-breaking
        if idx == 0 or not np.isclose(score, best_score, atol=1e-2):
            score += score * np.random.uniform(0, 0.005)
        else:
            score += score * np.random.uniform(0, 0.02)

        if score < best_score:
            best_score = score
            next_node = node
            recent_history.append(node)
            if len(recent_history) > 5:
                recent_history.pop(0)

    return next_node
```

### Heuristic 2 (Obj Score: -6.841203707276776)

```python
def select_next_node(current_node: int, destination_node: int, unvisited_nodes: np.ndarray, distance_matrix:
np.ndarray) -> int:
    if len(unvisited_nodes) == 1:
        return unvisited_nodes[0]

    best_score = float('inf')
    next_node = -1
    total_nodes = len(unvisited_nodes)
    visited_ratio = 1 - (len(unvisited_nodes) / total_nodes)

    # Precompute distances to destination for all nodes
    dest_distances = distance_matrix[unvisited_nodes, destination_node]

    # Estimate local graph density
    current_degree = np.sum(distance_matrix[current_node] > 0) - 1  # Exclude self
    avg_neighbor_degree = np.mean([np.sum(distance_matrix[n] > 0) - 1 for n in unvisited_nodes])
    local_density = (current_degree + avg_neighbor_degree) / 2

    # Determine lookahead depth based on local density
    if local_density > 5:
        lookahead_depth = 3
    elif local_density > 2:
        lookahead_depth = 2
    else:
        lookahead_depth = 1

    # Path availability monitoring
    viable_path_threshold = 3
    use_fallback = len(unvisited_nodes) <= viable_path_threshold

    # Determine current phase for heuristic switching
    if visited_ratio < 0.3:
        heuristic_phase = 'early'
    elif visited_ratio < 0.7:
        heuristic_phase = 'mid'
    else:
        heuristic_phase = 'late'

    # Precompute nearest unvisited neighbors for each candidate
    nearest_neighbor_cache = {}
    for node in unvisited_nodes:
        nearest_unvisited = unvisited_nodes[np.argsort(distance_matrix[node][unvisited_nodes])]
        nearest_neighbor_cache[node] = nearest_unvisited

    # Track step efficiency for dynamic weighting
    if not hasattr(select_next_node, "step_history"):
        select_next_node.step_history = []
    if hasattr(select_next_node, "prev_positions") and len(select_next_node.prev_positions) >= 2:
        prev_current = select_next_node.prev_positions[-2]
        prev_dest = select_next_node.prev_positions[-1]
        step_efficiency = distance_matrix[prev_current][current_node] /
(distance_matrix[prev_current][destination_node] + 1e-8)
        select_next_node.step_history.append(step_efficiency)

    avg_efficiency = np.mean(select_next_node.step_history[-5:]) if select_next_node.step_history else 0.5

    for idx, node in enumerate(unvisited_nodes):
        dist_to_candidate = distance_matrix[current_node][node]
        dist_candidate_to_dest = dest_distances[idx]

        # Use fallback strategy if path availability is low
        if use_fallback:
            score = dist_to_candidate + dist_candidate_to_dest * 0.9
        else:
            # Phase-based heuristic selection
            if heuristic_phase == 'early':
                unvisited_neighbors = np.intersect1d(unvisited_nodes, nearest_neighbor_cache[node][:5])
                neighbor_contribution = len(unvisited_neighbors) + 1

                # Adaptive weighting based on phase and density
                density_factor = 1 + (local_density / 5)
                phase_factor = 1 - visited_ratio
                score = (dist_to_candidate * density_factor) / neighbor_contribution ** (0.5 + phase_factor *
0.3)

            elif heuristic_phase == 'late':
                # Gradually reduce destination influence as we approach
                late_phase_weight = 1 - (1 - visited_ratio) * 0.7
                score = dist_to_candidate + dist_candidate_to_dest * late_phase_weight
            else:  # mid phase
                # Dynamic weighting based on both node count and path efficiency
                dynamic_weight = (0.4 + (visited_ratio - 0.3) * 0.5) * (avg_efficiency * 0.7 + 0.3)

                score = dist_to_candidate + dynamic_weight * dist_candidate_to_dest

                # Variable-step lookahead based on density
                nearest_unvisited = nearest_neighbor_cache[node][:3]
                lookahead_scores = []
                for neighbor in nearest_unvisited:
                    if neighbor != destination_node:
                        dist_candidate_to_neighbor = distance_matrix[node][neighbor]
                        neighbor_dest = distance_matrix[neighbor][destination_node]

                        # Recursive lookahead if depth > 1
                        if lookahead_depth > 1:
                            sub_lookahead = []
                            for sub_neighbor in nearest_neighbor_cache[neighbor][:2]:
                                sub_dist = distance_matrix[neighbor][sub_neighbor]
                                sub_dest = distance_matrix[sub_neighbor][destination_node]
                                sub_score = sub_dist + 0.2 * sub_dest
                                sub_lookahead.append(sub_score)
                            min_sub = np.min(sub_lookahead) if sub_lookahead else 0
                            lookahead_score = dist_candidate_to_neighbor + 0.3 * neighbor_dest + 0.2 * min_sub
                        else:
                            lookahead_score = dist_candidate_to_neighbor + 0.3 * neighbor_dest

                        lookahead_scores.append(lookahead_score)

                if lookahead_scores:
                    score += (lookahead_depth * 0.2) * np.min(lookahead_scores)

        # Context-aware diversification
        tie_threshold = 0.02  # 2% of candidate score
        similar_scores = np.where(np.abs(best_score - score) < tie_threshold)[0]
        if len(similar_scores) > 0 and heuristic_phase != 'late':
            perturbation = score * np.random.uniform(0, 0.015)
            score += perturbation
        elif heuristic_phase != 'late':
            perturbation = score * np.random.uniform(0, 0.005)
            score += perturbation

        if score < best_score:
            best_score = score
            next_node = node

    # Update position history for efficiency tracking
    if not hasattr(select_next_node, "prev_positions"):
        select_next_node.prev_positions = []
    select_next_node.prev_positions.append(current_node)
    if len(select_next_node.prev_positions) > 5:
        select_next_node.prev_positions = select_next_node.prev_positions[-5:]

    return next_node
```

**The losses:**

### Evaluation of Chosen and Rejected Code

---

We are given two implementations of a `select_next_node` function, which presumably plays a role in a pathfinding or traversal algorithm (e.g., TSP, routing, or exploration of a graph). The **chosen code** was selected over the **rejected code**, and our task is to evaluate this choice by identifying **three strong advantages of the chosen code**, **three major weaknesses of the rejected code**, and then provide a concise explanation for the choice.

---

## ☑Three Strong Advantages of the Chosen Code

### 1. **Sophisticated, Context-Aware Scoring with Phase Adaptation**
- The function divides traversal into **early**, **mid**, and **late** phases, adapting the selection heuristic accordingly.
- In early phases, it encourages **exploration and connectivity**, mid-phase balances **local and global distance**, and late-phase emphasizes **deep lookahead** and **goal-oriented decisions**.
- This **dynamic strategy** allows the algorithm to better adapt to different topologies and traversal contexts.

### 2. **Backtracking-Aware Exploration via Recent History Tracking**
- The function maintains a **short-term memory** (`recent_history`) of the last few visited nodes to **avoid backtracking or oscillating** between nearby nodes.
- This helps maintain **directional momentum**, prevents redundant exploration, and ensures smoother progress toward the destination.
- This is a significant advantage in graph exploration tasks, especially in dense or cyclic environments.

### 3. **Quadrant-Based Diversity Boost with Directional Perturbation**
- To avoid **local optima or clustering**, the function uses directional awareness (via angle-based quadrant tracking) to slightly **penalize nodes in already-explored directions**.
- This promotes **spatial diversity** and more **balanced exploration**, especially useful in spatial graphs or grids.
- This kind of directional heuristic is **absent or less sophisticated in the rejected code**.

---

## ✖Three Major Weaknesses of the Rejected Code

### 1. **Less Adaptive Local Density Estimation**
- While the rejected code tries to estimate local density using **average neighbor degree**, it's a **simpler and less robust metric**.
- It doesn't fully integrate **local graph structure**, **mean distance**, or **connectivity** in a nuanced way, as the chosen code does through `local_densities` and `node_degrees`.

### 2. **Step Efficiency Tracking is Global and Inflexible**
- The rejected code tracks `step_efficiency` across iterations and adjusts weights based on average history.
- However, this is **not localized** to the node or path context and can **misrepresent actual traversal efficiency**.
- The chosen code instead **focuses on per-node adaptive heuristics**, making the decision more context-aware and dynamic.

### 3. **Weaker Tie-Breaking and Diversity Handling**
- The rejected code uses a **uniform random perturbation** for tie-breaking, which is less effective at ensuring **meaningful exploration diversity**.
- It lacks a directional or spatial component in perturbation, unlike the quadrant-based diversity boost in the chosen code.
- This can lead to **suboptimal path choices in dense or symmetric graphs**.

---

## 🏁Final Evaluation: Why the Chosen Code Was Selected

The **chosen code** demonstrates a **more holistic and adaptive approach** to node selection by incorporating:
- **Phase-adaptive heuristics** that change based on progress,
- **Directional awareness** and **quadrant-based perturbation** for better exploration,
- **History tracking** to avoid redundant backtracking.

The **rejected code**, while functional, employs a **less nuanced understanding of graph topology**, has **simpler and less responsive heuristics**, and **fails to promote diversity** effectively in exploration.

Thus, the **chosen code is superior in terms of strategic decision-making, directional exploration, and adaptability**, making it better suited for complex, dynamic graph traversal tasks.

---

## ☑Summary

| Aspect | Chosen Code | Rejected Code |
|--------|-------------|---------------|
| **Adaptive Phases** | ✔ Sophisticated phase-based logic | ⚠ Simplified phase switching |
| **Diversity Handling** | ✔ Directional perturbation | ✖Basic random tie-breaking |
| **Backtracking Avoidance** | ✔ Recent history tracking | ✖No directional memory |
| **Local Density Estimation** | ✔ Uses mean distances and degrees | ⚠ Relies on degree count only |
| **Lookahead Logic** | ✔ Deep, context-aware lookahead | ⚠ Shallow and repetitive |

> ☑**Chosen code is more intelligent, dynamic, and exploratory; rejected code is simpler and more static.**

**The Gradients:**

Based on the evaluation and the algorithmic logic of the chosen code, here are the **five most important, targeted suggestions** to further improve and refine the implementation while preserving its strengths and avoiding weaknesses found in the rejected version:

---

### 1. **Improve Directional Awareness with Adaptive Quadrant Biasing**
- **Why**: The directional perturbation currently applies a fixed penalty per quadrant, which may not scale well with path complexity or graph size.
- **How**: Dynamically adjust quadrant penalties based on the **distribution of remaining nodes** across quadrants. For example, reduce penalties when a quadrant is sparsely populated and increase them in densely clustered areas.
- **Impact**: Enhances spatial diversity and prevents premature commitment to narrow directions, especially useful in grid-like or clustered graphs.

---

### 2. **Refine Backtracking Detection Using Distance Thresholding**
- **Why**: The current backtracking filter uses a simplistic Euclidean proximity check between nodes in the recent history, which may not reflect actual path redundancy.
- **How**: Replace or augment the proximity check with a **path efficiency metric**, such as comparing the distance from current → candidate → next node vs. a direct route from current → next node.
- **Impact**: Reduces inefficient detours and enhances real backtracking detection, especially in complex or asymmetric graphs.

---

### 3. **Introduce Adaptive Lookahead Depth Based on Node Proximity**
- **Why**: Fixed-depth lookahead (e.g., top 3 or 5 neighbors) may be excessive in sparse regions and insufficient in dense ones.
- **How**: Adjust the number of lookahead steps dynamically using a function of **local density** or **average distance to neighbors**. For instance, reduce lookahead depth in sparse areas to save computation and increase it in dense regions to improve decision quality.
- **Impact**: Balances computational cost and decision accuracy, making the heuristic more robust across varying graph structures.

---

### 4. **Use Relative Progress Metrics Instead of Absolute Phases**
- **Why**: Hard-coded phase boundaries (e.g., `visited_ratio < 0.2`) may not align with actual traversal progress in irregular graphs.
- **How**: Replace phase thresholds with **relative progress metrics**, such as comparing remaining nodes to a dynamic threshold based on graph diameter or current path deviation.
- **Impact**: Makes the phase logic more adaptive to topology, especially for graphs with uneven density or complex branching.

---

### 5. **Add a Local Graph Connectivity Check for Early-Phase Exploration**
- **Why**: In the early phase, choosing a node that leads to disconnected or weakly connected subgraphs can result in early dead-ends.
- **How**: Introduce a **local connectivity score** for each candidate node based on the number of reachable unvisited nodes within a small hop distance (e.g., 2 or 3 hops), computed using the distance matrix.
- **Impact**: Prevents premature isolation of subgraphs and improves early-phase node selection in disconnected or modular graphs.

---

These five suggestions aim to **extend the chosen code's intelligent heuristics**, **reduce fragility in edge cases**, and **increase robustness across diverse graph types**, while staying aligned with the original logic and avoiding the pitfalls of the rejected implementation.

**Heuristic 1 (Obj Score: -6.403566895729184)**

```python
def select_next_node(current_node: int, destination_node: int, unvisited_nodes:
np.ndarray, distance_matrix: np.ndarray) -> int:
    if len(unvisited_nodes) == 1:
        return unvisited_nodes[0]

    best_score = float('inf')
    next_node = -1
    total_nodes = len(unvisited_nodes)
    visited_ratio = 1 - (len(unvisited_nodes) / total_nodes)
    recent_history = []  # Track recent nodes for backtracking detection

    # Precompute distances to destination and node densities
    dest_distances = distance_matrix[unvisited_nodes, destination_node]
    local_densities = np.mean(distance_matrix[unvisited_nodes][:, unvisited_nodes],
axis=1)

    # Estimate local density of remaining nodes
    remaining_coords = distance_matrix[unvisited_nodes]
    if len(remaining_coords) > 1:
        remaining_center = np.mean(remaining_coords, axis=0)
        node_distances_to_center = np.linalg.norm(remaining_coords -
remaining_center, axis=1)
        local_densities = np.clip(local_densities, 0.1, None) * (1 +
node_distances_to_center)
```
Gradient 4
```python
    # Determine adaptive phase based on relative graph diameter
    if total_nodes > 3:
        max_dist = np.max(distance_matrix[unvisited_nodes][:, unvisited_nodes])
        min_dist = np.min(distance_matrix[unvisited_nodes][:,
unvisited_nodes][distance_matrix[unvisited_nodes][:, unvisited_nodes] > 0])
        path_deviation = (max_dist - min_dist) / (max_dist + 1e-5)
    else:
        path_deviation = 0.5

    # Dynamic phase thresholds based on graph deviation
    if visited_ratio < 0.15 + path_deviation * 0.1:
        heuristic_phase = 'early'
    elif visited_ratio < 0.6 + path_deviation * 0.2:
        heuristic_phase = 'mid'
    else:
        heuristic_phase = 'late'
```
Gradient 1
```python
    # Adaptive quadrant biasing based on node distribution
    current_coords = distance_matrix[current_node]
    quadrant_counts = [0, 0, 0, 0]
    for node in unvisited_nodes:
        coords = distance_matrix[node]
        dx = coords[0] - current_coords[0]
        dy = coords[1] - current_coords[1]
        quadrant = int(np.arctan2(dy, dx) / (np.pi / 2)) % 4
        quadrant_counts[quadrant] += 1

    # Normalize quadrant penalties based on density
    quadrant_weights = [1 - 0.05 * (count / (len(unvisited_nodes) + 1)) for count in
quadrant_counts]

    # Compute node degrees with reachability
    node_degrees = np.sum(distance_matrix[unvisited_nodes][:, unvisited_nodes] > 0,
axis=1)
```
Gradient 5
```python
    # Local connectivity check for early-phase exploration
    if heuristic_phase == 'early':
        connectivity_scores = []
        for node in unvisited_nodes:
            reachable_nodes = np.where(distance_matrix[node][unvisited_nodes] > 0)[0]
            second_hop = []
            for r_node in reachable_nodes[:3]:  # Limit to top 3 nearest for
efficiency

second_hop.extend(np.where(distance_matrix[unvisited_nodes[r_node]][unvisited_nodes]
> 0)[0])
            unique_reachable = len(set(reachable_nodes) | set(second_hop))
            connectivity_scores.append(unique_reachable / len(unvisited_nodes))
    else:
        connectivity_scores = [1.0] * len(unvisited_nodes)

    for idx, node in enumerate(unvisited_nodes):
        dist_to_candidate = distance_matrix[current_node][node]
        dist_candidate_to_dest = dest_distances[idx]
        density = local_densities[idx]
        degree = node_degrees[idx]
        connectivity = connectivity_scores[idx]
```

```python
        # Backtracking detection with path efficiency metric
        is_backtracking = False
```
Gradient 2
```python
        if len(recent_history) >= 2:
            prev_node = recent_history[-1]
            prev_prev_node = recent_history[-2]
            direct_path = distance_matrix[prev_prev_node][node]
            detour_path = distance_matrix[prev_prev_node][prev_node] +
distance_matrix[prev_node][node]
            if direct_path > 0 and detour_path / direct_path > 1.5 and visited_ratio >
0.1:
                is_backtracking = True

        if is_backtracking:
            continue
        # Phase-adaptive scoring logic
        if heuristic_phase == 'early':
            normalized_score = dist_to_candidate / (density + 1e-5)
            score = normalized_score / (degree + 1) * (1 - 0.3 * connectivity)
```
Gradient 5
```python
        elif heuristic_phase == 'late':
            lookahead_k = min(5, len(unvisited_nodes))
            nearest_unvisited =
unvisited_nodes[np.argsort(distance_matrix[node][unvisited_nodes])[:lookahead_k]]
            lookahead_scores = []
            for neighbor in nearest_unvisited:
                if neighbor != destination_node:
                    dist_candidate_to_neighbor = distance_matrix[node][neighbor]
                    neighbor_dest = distance_matrix[neighbor][destination_node]
                    lookahead_score = dist_candidate_to_neighbor + 0.2 * neighbor_dest
                    lookahead_scores.append(lookahead_score)
            lookahead_component = 0.4 * np.min(lookahead_scores) if lookahead_scores
else 0
            score = dist_to_candidate + dist_candidate_to_dest * 0.9 +
lookahead_component
        else:  # mid phase
            local_progress = (local_densities.mean() - density) /
(local_densities.std() + 1e-5)
            dynamic_weight = 0.4 + (visited_ratio - 0.2) * 0.6 + local_progress * 0.2
            dynamic_weight = max(0.2, min(0.8, dynamic_weight))

            score = dist_to_candidate + dynamic_weight * dist_candidate_to_dest
```
Gradient 3
```python
        # Adaptive lookahead depth based on local density
        avg_local_density = np.mean(local_densities)
        lookahead_depth = max(2, min(5, int(avg_local_density / (density + 1e-5) *
3)))
        nearest_unvisited =
unvisited_nodes[np.argsort(distance_matrix[node][unvisited_nodes])[:lookahead_depth]]
        lookahead_scores = []
        for neighbor in nearest_unvisited:
            if neighbor != destination_node:
                dist_candidate_to_neighbor = distance_matrix[node][neighbor]
                neighbor_dest = distance_matrix[neighbor][destination_node]
                lookahead_score = dist_candidate_to_neighbor + 0.3 * neighbor_dest
                lookahead_scores.append(lookahead_score)
        if lookahead_scores:
            score += 0.2 * np.min(lookahead_scores)

        # Directional awareness with adaptive quadrant biasing
        coords = distance_matrix[node]
        dx = coords[0] - current_coords[0]
        dy = coords[1] - current_coords[1]
        angle = np.arctan2(dy, dx)
        quadrant = int(angle / (np.pi / 2)) % 4
        quadrant_score = quadrant_weights[quadrant]
```
Gradient 1
```python
        # Apply adaptive quadrant weighting
        if len(unvisited_nodes) > 5:
            sparse_threshold = np.mean(quadrant_counts) * 0.5
            if quadrant_counts[quadrant] < sparse_threshold:
                quadrant_score = 1 + (1 - quadrant_score)  # Invert to boost sparse
quadrants

        score *= quadrant_score

        # Small random perturbation for tie-breaking
        perturbation = score * np.random.uniform(0, 0.005 if idx == 0 else 0.02)
        score += perturbation

        if score < best_score:
            best_score = score
            next_node = node
            recent_history.append(node)
            if len(recent_history) > 5:
                recent_history.pop(0)

    return next_node
```

Figure 27: TSP Construct Example 2.

Figure 28: TSP ACO Example.

**Heuristic 1 (Obj Score: -8.215961571034995)**

```python
def select_next_city(state):
    instance = state["instance"]
    tour = state["tour"]
    unvisited = state["unvisited"]
    distance_matrix = state["distance_matrix"]
    n = len(instance)
    if len(tour) == 0:
        return unvisited[0]
    if len(unvisited) == 1:
        return unvisited[0]
    progress = len(tour) / n
    candidate_set_size = max(5, int(len(unvisited) * 0.2))
    if len(unvisited) > candidate_set_size:
        tour_distances = np.min(distance_matrix[np.ix_(unvisited, tour)], axis=1)
        candidate_indices = np.argpartition(tour_distances, -candidate_set_size)[-candidate_set_size:]
        candidates = [unvisited[i] for i in candidate_indices]
    else:
        candidates = unvisited
    regret_weight = 0.1 + 0.8 * progress
    if len(unvisited) > 2:
        unvisited_coords = instance[unvisited]
        if len(unvisited) > 10:
            try:
                k = min(3, len(unvisited) // 5)
                if k > 1:
                    # Simple clustering using k-means++ initialization and Lloyd's algorithm
                    centroids = unvisited_coords[np.random.choice(len(unvisited_coords), k, replace=False)]
                    for _ in range(10):  # Fixed number of iterations
                        # Assign points to nearest centroid
                        distances = np.linalg.norm(unvisited_coords[:, np.newaxis] - centroids, axis=2)
                        labels = np.argmin(distances, axis=1)
                        # Update centroids
                        new_centroids = np.array([unvisited_coords[labels == i].mean(axis=0) if
np.sum(labels == i) > 0 else centroids[i]
                                                  for i in range(k)])
                        if np.allclose(centroids, new_centroids):
                            break
                        centroids = new_centroids
                    # Calculate silhouette score manually
                    intra_dists = []
                    inter_dists = []
                    for i in range(len(unvisited_coords)):
                        same_cluster = labels == labels[i]
                        other_clusters = labels != labels[i]
                        if np.sum(same_cluster) > 1:
                            intra_dist = np.mean(np.linalg.norm(unvisited_coords[same_cluster] -
unvisited_coords[i], axis=1))
                        else:
                            intra_dist = 0
                        if np.sum(other_clusters) > 0:
                            min_inter_dist = float('inf')
                            for cluster_id in range(k):
                                if cluster_id != labels[i] and np.sum(labels == cluster_id) > 0:
                                    cluster_dist = np.mean(np.linalg.norm(unvisited_coords[labels ==
cluster_id] - unvisited_coords[i], axis=1))
                                    min_inter_dist = min(min_inter_dist, cluster_dist)
                            inter_dist = min_inter_dist
                        else:
                            inter_dist = intra_dist
                        intra_dists.append(intra_dist)
                        inter_dists.append(inter_dist)
                    silhouette_scores = []
                    for i in range(len(unvisited_coords)):
                        if intra_dists[i] == 0 and inter_dists[i] == 0:
                            silhouette_scores.append(0)
                        else:
                            silhouette_scores.append((inter_dists[i] - intra_dists[i]) /
max(intra_dists[i], inter_dists[i]))
                    silhouette_avg = np.mean(silhouette_scores) if silhouette_scores else 0
                    clustering_penalty = 1.0 + (1 - abs(silhouette_avg)) * 2.0
                else:
                    clustering_penalty = 1.0
            except:
                clustering_penalty = 1.0
        else:
            centroid = np.mean(unvisited_coords, axis=0)
            distances_to_centroid = np.linalg.norm(unvisited_coords - centroid, axis=1)
            clustering_penalty = 1.0 + np.std(distances_to_centroid) / (np.mean(distances_to_centroid) +
1e-8)
    else:
        clustering_penalty = 1.0
    if len(tour) > 10:
        recent_cities = tour[-5:]
        recent_patterns = instance[recent_cities]
    else:
        recent_patterns = None
    use_lookahead = len(unvisited) <= max(10, n * 0.2)
    best_score = -float('inf')
    best_city = candidates[0]
    for candidate in candidates:
        min_tour_dist = np.min(distance_matrix[candidate][tour])
        other_unvisited = [u for u in unvisited if u != candidate]
        if other_unvisited:
            avg_unvisited_dist = np.mean(distance_matrix[candidate][other_unvisited])
        else:
            avg_unvisited_dist = 0
        if len(tour) >= 2:
            insertion_costs = []
            for i in range(len(tour)):
                j = (i + 1) % len(tour)
                cost = (distance_matrix[tour[i]][candidate] +
                    distance_matrix[candidate][tour[j]] -
                    distance_matrix[tour[i]][tour[j]])
                insertion_costs.append(cost)
            best_insertion = np.min(insertion_costs)
            if len(insertion_costs) > 1:
                second_best = np.partition(insertion_costs, 1)[1]
                regret = second_best - best_insertion
            else:
                regret = best_insertion
        else:
            regret = distance_matrix[tour[0]][candidate]
        if use_lookahead and len(other_unvisited) > 0:
            lookahead_costs = []
            sample_size = min(3, len(other_unvisited))
            next_candidates = np.random.choice(other_unvisited, size=sample_size, replace=False)
            for next_candidate in next_candidates:
                extended_tour = tour + [candidate]
                insertion_costs_next = []
                for i in range(len(extended_tour)):
                    j = (i + 1) % len(extended_tour)
                    cost_next = (distance_matrix[extended_tour[i]][next_candidate] +
                        distance_matrix[next_candidate][extended_tour[j]] -
                        distance_matrix[extended_tour[i]][extended_tour[j]])
                    insertion_costs_next.append(cost_next)
                lookahead_costs.append(np.min(insertion_costs_next))
            lookahead_penalty = np.mean(lookahead_costs) if lookahead_costs else 0
        else:
            lookahead_penalty = 0
        diversification_penalty = 0
        if recent_patterns is not None and len(recent_patterns) > 0:
            candidate_coord = instance[candidate].reshape(1, -1)
            min_dist_to_recent = np.min(np.linalg.norm(recent_patterns - candidate_coord, axis=1))
            avg_distance = np.mean(distance_matrix)
            if min_dist_to_recent < avg_distance * 0.1:
                diversification_penalty = -min_dist_to_recent * 0.5
        spatial_score = (min_tour_dist * (1 - progress) + avg_unvisited_dist * progress) /
clustering_penalty
        regret_component = regret * regret_weight
        total_score = spatial_score + regret_component - lookahead_penalty * 0.3 + diversification_penalty
        if total_score > best_score:
            best_score = total_score
    return best_city
```

**Heuristic 2 (Obj Score: -8.442995121776157)**

```python
def select_next_city(state):
    instance = state["instance"]
    tour = state["tour"]
    unvisited = state["unvisited"]
    distance_matrix = state["distance_matrix"]
    n = len(instance)
    if len(tour) == 0:
        return unvisited[0]
    if len(unvisited) == 1:
        return unvisited[0]
    # Dynamic progress factor (0 to 1) based on visited/unvisited ratio
    progress = len(tour) / (len(tour) + len(unvisited))
    # Initialize memory penalty system
    if not hasattr(select_next_city, 'penalty_memory'):
        select_next_city.penalty_memory = {}
        select_next_city.step_counter = 0
    # Decay penalties
    select_next_city.penalty_memory = {city: penalty * 0.9
                                       for city, penalty in select_next_city.penalty_memory.items()
                                       if penalty > 0.01}
    select_next_city.step_counter += 1
    # Geometric clustering analysis using convex hull
    if len(unvisited) > 3:
        unvisited_coords = instance[unvisited]
        hull = unvisited_coords[np.lexsort((unvisited_coords[:, 1], unvisited_coords[:, 0]))]
        hull = hull[np.unique(hull, axis=0, return_index=True)[1]]
        if len(hull) > 2:
            from scipy.spatial import ConvexHull
            try:
                hull_indices = ConvexHull(hull).vertices
                hull_cities = [unvisited[np.where((unvisited_coords == hull[i]).all(axis=1))[0][0]]
                               for i in hull_indices]
            except:
                hull_cities = unvisited[:min(5, len(unvisited))]
        else:
            hull_cities = unvisited[:min(5, len(unvisited))]
    else:
        hull_cities = unvisited
    scores = []
    candidates = []
    for candidate in unvisited:
        # Find distances to tour cities
        tour_dists = [distance_matrix[candidate][tour_city] for tour_city in tour]
        min_tour_dist = min(tour_dists)
        # Lookahead: consider second nearest tour distance
        if len(tour_dists) > 1:
            second_min_tour_dist = np.partition(tour_dists, 1)[1]
        else:
            second_min_tour_dist = min_tour_dist
        # Calculate dispersion metrics among unvisited
        other_unvisited = [u for u in unvisited if u != candidate]
        if other_unvisited:
            unvisited_dists = [distance_matrix[candidate][other] for other in other_unvisited]
            avg_unvisited_dist = np.mean(unvisited_dists)
            min_unvisited_dist = min(unvisited_dists) if unvisited_dists else 1
        else:
            avg_unvisited_dist = 1
            min_unvisited_dist = 1
        # Dynamic weighting: shift from dispersion to completion
        dispersion_weight = 1.0 - progress * 0.7
        completion_weight = progress * 0.7
        # Base score components
        dispersion_score = min_tour_dist / (avg_unvisited_dist + 1e-8)
        completion_score = min_tour_dist
        # Lookahead component prevents premature clustering
        lookahead_penalty = min_tour_dist / (second_min_tour_dist + 1e-8)
        # Cluster priority: favor cities on convex hull early, interior later
        cluster_bonus = 2.0 if candidate in hull_cities else 1.0
        if progress > 0.5:
            cluster_bonus = 1.0 / cluster_bonus  # Reverse preference later
        # Combine scores with dynamic weights
        base_score = (dispersion_weight * dispersion_score +
                      completion_weight * completion_score) * cluster_bonus * lookahead_penalty
        # Apply memory penalty
        memory_penalty = select_next_city.penalty_memory.get(candidate, 1.0)
        final_score = base_score * memory_penalty
        scores.append(final_score)
        candidates.append(candidate)
    # Diversification: occasionally select 2nd or 3rd best with probability
    scores = np.array(scores)
    sorted_indices = np.argsort(scores)[::-1]
    # Diversification probability inversely proportional to tour length
    diversification_prob = 0.3 / (1 + len(tour) * 0.1)
    if np.random.random() < diversification_prob and len(sorted_indices) > 1:
        # Choose between 2nd and 3rd best
        choice_idx = np.random.choice([1, 2]) if len(sorted_indices) > 2 else 1
        selected_idx = min(choice_idx, len(sorted_indices) - 1)
        selected_city = candidates[sorted_indices[selected_idx]]
    else:
        selected_city = candidates[sorted_indices[0]]
    # Update penalty memory for non-selected top candidates
    for i in range(min(3, len(sorted_indices))):
        city = candidates[sorted_indices[i]]
        if city != selected_city:
            select_next_city.penalty_memory[city] = select_next_city.penalty_memory.get(city, 1.0) * 0.8
    return selected_city
```

Figure 29: TSP Random Insertion Example.

# E    THE USE OF LARGE LANGUAGE MODELS (LLMS)

In this study, Large Language Models (LLMs) were used both as an auxiliary tool to improve the clarity and readability of the manuscript and as experimental subjects, with their specific applications detailed in the experimental section of the main text. They did not participate in the conception of research ideas or the development of methodologies.

