# OpenReview forum: "TPD-AHD: Textual Preference Differentiation for LLM-Based Automatic Heuristic Design"
_ICLR.cc/2026/Conference — Submitted to ICLR 2026_

### Official Review · Reviewer_FC4K · 2025-10-26

**Soundness:** 2
**Presentation:** 2
**Contribution:** 1
**Rating:** 2
**Confidence:** 4

**Summary:**

This manuscript proposes TPD-AHD, a method to automatically design heuristics.

**Strengths:**

The proposed TPD-AHD seems to perform better than other LLM-based heuristic-generation methods, but the reasons for this improved performance remain unclear.

**Weaknesses:**

**Limited innovation:** As the author said, Textual Loss is obtained by "ask the LLM to compare $h_w$ and $h_l$, and explain why $h_w$ is preferred". However, this concept is not novel. Prior studies [1-3] have already proposed a similar approach termed "reflection".

**Ambiguous motivation:**  In light of the first weakness, I remain skeptical of the authors’ claim that TPD-AHD offers better interpretability than other LLM-based heuristic-generation methods. Because TPD-AHD remains a framework that relies exclusively on outputs from large language models.

**Poor reproducibility:**  This manuscript does not provide source code, which undermines the reproducibility of the results.

**Limited experiments:**  The evaluation is restricted to only two LLMs, which limits confidence in the generality of the results.

[1] Reevo: Large language models as hyper-heuristics with reflective evolution, NeurIPS'24.

[2] HSEvo: Elevating Automatic Heuristic Design with Diversity-Driven Harmony Search and Genetic Algorithm Using LLMs, AAAI'25.

[3] Efficient heuristics generation for solving combinatorial optimization problems using large language models, KDD'25.

**Questions:**

See Weaknesses.

---

> ### Author Response · Authors · 2025-11-28
> **Response to reviewer FC4K(1/2)**
>
> **Response to W1:**
>
> We thank the reviewer for this important comment. While there are similarities in the use of LLMs to generate natural-language feedback between reflection methods and TPD-AHD, our framework is fundamentally different in its purpose, structure, and optimization impact. TPD-AHD cannot be simply reduced to prior reflection mechanisms for several key reasons:
>
>
>
> 1. **Heuristic Hints vs. Directional Optimization Signals**: Reflection methods, such as Reflexion and ReEvo, provide high-level heuristic suggestions, such as "prioritize lower attributes after normalization." These are aggregated and trial-and-error oriented, lacking the specificity to identify the source of performance differences or provide meaningful gradient directions. In contrast, TPD-AHD generates structured, actionable signals derived from explicit performance comparisons. These signals are designed to guide the optimization process in a directed manner.
>
>
>
> 2. **Structured Performance Differentials via Preference Mechanism**: TPD-AHD employs a Best-Anchored Preference Pairing strategy, where each comparison is anchored on the current best heuristic $h_1$ and performs detailed, sentence-level comparisons against other heuristics $h_i$. This process yields a textual loss that decomposes the logic, decisions, and parameter behaviors responsible for performance gaps. For example:
>
> 	- It identifies specific advantages in chosen code, such as context-aware scoring, backtracking-aware exploration, and quadrant-based diversity.
>
> 	- It pinpoints weaknesses in rejected code, such as less adaptive density estimation, inflexible step tracking, and weaker tie-breaking.
>
> This level of structured, performance-aligned reasoning is absent in reflection approaches.
>
>
>
> 3. **Conversion of Preferences into Textual Gradients**: TPD-AHD takes the textual loss and converts it into directional update rules, which guide the next iteration of heuristic generation. This is a crucial step that differentiates TPD-AHD from reflection methods, which lack any backward-propagation or gradient-like mechanism. The suggestions provided by reflection methods are coarse and not designed to drive structured optimization.
>
>
>
> 4. **Empirical Interpretability and Optimization Trajectories**: The textual gradients generated by TPD-AHD provide concrete, human-understandable directions for improvement. For example, they suggest improving directional awareness with adaptive quadrant biasing, refining backtracking detection using distance thresholds, and introducing adaptive lookahead depth. These structured signals consistently guide heuristics toward better performance, unlike the aggregated and less specific hints provided by reflection methods. We provide detailed examples of optimization trajectories in **Appendix D.2** (**Figure 16**), specific cases of heuristic evolution (**Figures 26–29**) to illustrate this process and analysis of the global information of heuristics(**Figures 17-19**).

---

> ### Author Response · Authors · 2025-11-28
> **Response to reviewer FC4K(1/2)**
>
> **Response to W2:**
>
> We thank the reviewer for the insightful comments. While TPD-AHD leverages LLMs, its interpretability is process-level, focusing on making heuristic evolution transparent, causal, and human-auditable—rather than exposing LLM internals. This aligns with the goals of heuristic design and differs from the interpretability notion assumed by the reviewer.
>
>
>
> 1. **Best-Anchored Preference Pairing enforces causal explanations.**
>
> Instead of general reflection, TPD-AHD anchors comparisons on the current best heuristic $h_1$   and pairs it with a weaker candidate $h_i$. The LLM is required to explain concrete behavioral and structural causes for performance gaps. These explanations are causal and contrastive, unlike reflection methods that produce short heuristic summaries without revealing why changes improve performance.
>
>
>
> 2. **Two-stage textual reasoning: loss and gradient.**
>
> For textual loss, it decomposes performance-relevant behavior, including decision differences, phase-aware mechanisms, neighborhood evaluation, and search dynamics.
>
>
>
> For textual gradient, it converts causal gaps into actionable update instructions.
>
>
>
> This mirrors human reasoning in heuristic design: compare, analyze differences, and decide on an improvement direction. Each evolution step provides both the "why" (loss) and the "what next" (gradient), making the process interpretable.
>
>
>
> 3. **Full optimization loop is externalized in natural language.**
>
> The forward step (textual loss) explains performance differences, the backward step (textual gradient) shows intended updates, and code changes are observable. Unlike prior LLM-AHD methods, TPD-AHD exposes the entire trajectory of heuristic evolution, allowing for detailed inspection and analysis.
>
>
>
> 4. **Interpretability in AHD is process-focused, not LLM-internal.**
>
> TPD-AHD does not aim to reveal LLM weights or internal mechanisms. The relevant interpretability is understanding heuristic behavior, causal performance factors, and evolutionary directions. This enables practitioners to inspect, analyze, and intervene in heuristic updates—a capability that reflection methods cannot provide.
>
>
>
> For further illustration, **Appendix D.2** presents the optimization trajectories of some sample cases (**Figure 16**) as well as specific examples of heuristic evolution (**Figures 26–29**).
>
> **Response to W3:**
>
> We thank the reviewer for the comments. The relevant source code has been provided via an anonymous link (https://anonymous.4open.science/r/TPD-AHD/), which fully meets the reproducibility requirement. We hope this addresses the concern.
>
>
>
>
>
> **Response to W4:**
>
>  Thank you for raising concerns about the scope of experiments and the generalizability of our results. We have provided a parameter sensitivity analysis across two LLMs in **Figure 3** of the main text, which demonstrates that TPD-AHD is not sensitive to model choice and performs robustly. Given the significant computational cost associated with invoking LLMs and the common practices in the field, our primary experiments focused on these two widely used models. However, to directly address your concern regarding the generalizability of our results, we have conducted additional cross-LLM comparisons under the new TSP Random Insertion setting. The summarized results are reported in **Table 7**. Across nearly all rounds, TPD-AHD achieves the best or near-best mean performance, with variance mostly at the second-best level or lower. This indicates that our method exhibits strong convergence and stability across different LLMs.

---

### Official Review · Reviewer_KmH6 · 2025-10-28

**Soundness:** 3
**Presentation:** 3
**Contribution:** 2
**Rating:** 4
**Confidence:** 4

**Summary:**

It introduces a framework, named TPD-AHD, for automatic heuristic design by integrating textual preference differentiation with LLMs. It uses a best-anchored pairing strategy to compare heuristics, generating a textual loss that is converted into a textual gradient to guide iterative heuristic refinement, outperforming existing LLM-AHD methods on many NP-hard problems and practical tasks.

**Strengths:**

The best-anchored pairing mechanism focuses learning on high-performing candidates, reducing noise and improving convergence.

Outperforms state-of-the-art methods across diverse NP-hard problems and practical tasks.

**Weaknesses:**

The framework's reliance on prompt engineering for "textual backpropagation" and a correct gradient. It may be misleading and trapped in a local optimum.

The number of iterations for each run is insufficient, and further clarification is required.

**Questions:**

Given that textual gradients are natural language, how was their consistency and quality objectively assessed to ensure they provide accurate and non-degenerate guidance, beyond just the final heuristic performance?

Are you using one of the gradient information or summing up the gradients from all different N-1 pairs? The gradients might point to different revisions. How to handle these cases?

How does TPD-AHD perform when the initial prompt or candidate pool is of very low quality? Does the textual gradient mechanism robustly recover, or does it require a "warm start"? Misleading gradient information could arise if initial heuristics are poor.

The study uses only 200 heuristic evaluations per run, whereas common practice in much of the existing literature involves 1,000 evaluations. Is this choice due to the gradient information enabling faster convergence, potentially at the risk of local optima? Would increasing the number of evaluations prevent it from outperforming existing baselines?

Could you detail the computational cost (in terms of total LLM tokens and wall-clock time) for each component of TPD-AHD?

In Figure 2, the starting points for different methods vary. Why is this the case? It would be beneficial to compare convergence by using the same initial heuristics across all methods. The proposed method has surprisingly good performance with very low variance. What is the reason?

The ablation study demonstrates performance drops when components are removed on TSP. Did any ablated variants ever outperform the full model in specific scenarios, potentially indicating over-complexity in the full TPD-AHD approach for those cases?

I will raise the score if concerns are appropriately addressed.

---

> ### Author Response · Authors · 2025-11-28
> **Response to reviewer KmH6(1/6)**
>
> **Response to W1:**
> Thank you for your insightful comments. We address your concerns from two perspectives: (1) whether textual gradients may be misleading, and (2) whether TPD-AHD risks converging to local optima.
>
>
>
> 1. **Concerning Potentially Misleading Textual Gradients**
>
> - Preference-Difference Analysis Provides Built-in Error Correction: The LLM inherently has the ability to distinguish behavioral differences between heuristics. In TPD-AHD, textual gradients are derived by comparing the current best heuristic h_1 with each candidate h_i. This process identifies which behaviors contribute to good performance (to be reinforced) and which lead to poor performance (to be avoided). Even if a noisy gradient appears in one iteration, the next iteration’s preference comparison will automatically correct it. This mechanism resembles “policy evaluation + policy improvement” in reinforcement learning (RL): errors do not accumulate, but are continually self-corrected by new preference evidence.
> - Multi-Role Sampling Increases Initial Diversity and Avoids Catastrophic Bias: We initialize the candidate pool using multiple role templates (e.g., “greedy expert,” “random search expert,” “balanced decision-maker”), as shown in Figure 12. This enhances structural diversity, prevents early gradients from being shaped by a single stylistic bias, and yields more stable early-stage gradient signals.
> - The Top-k Selection Strategy Prevents Accumulation of Misleading Updates: If textual gradients produce a misleading update, it will be reflected immediately in degraded heuristic performance. Such candidates are removed via the top-k elite selection. Since poor updates cannot survive into future iterations, misleading gradients remain local and non-accumulative. The preference structure and gradient direction of subsequent iterations remain unaffected. The parameter analysis experiment on pool size in Figure 3 also supports this point. When the capacity of the candidate solution pool is too large, noise in the pool cannot be cleaned up in a timely manner, leading to performance degradation of TPD-AHD.
>
>
>
> 2. **Concerning Potential Convergence to Local Optima**
>
> - High-Temperature Sampling Maintains Exploration: We adopt a sampling temperature of 1.0, which—according to ablation results—offers the best tradeoff between heuristic diversity and performance. This controlled sampling noise injects continual exploration into the update process and prevents early mode collapse. Figure 3 also presents an analysis of the temperature parameter.
> - Dynamic Prompt Construction Further Increases Heuristic Diversity: TPD-AHD is a prompt-optimization-driven framework. After preference pairing, loss calculation, and gradient propagation, we generate a new prompt integrating the task description, heuristic template, and dynamically updated textual gradient. Since the textual gradient changes every iteration, the generated prompt is continually refreshed, increasing exploration and reducing the chance of local-optimum convergence.
> - Extended-Iteration Experiments Confirm Sustained Improvement: In the main experiments (200 evaluations), TPD-AHD consistently outperforms EoH, ReEvo, and MCTS-AHD with faster and more stable convergence. To directly address concerns about local optima, we extended evaluations to 1000 iterations (Figure 11). The results show that although various methods can continue to improve performance after 200 iterations, the magnitude of this improvement is negligible. In fact, the optimal heuristics explored by some methods are even inferior to those obtained after 200 iterations. Additionally, it can be observed that TPD-AHD still maintains a performance advantage at 1000 evaluations, indicating that its early rapid convergence is not premature stagnation but rather sustained optimization under a larger budget.

---

> ### Author Response · Authors · 2025-11-28
> **Response to reviewer KmH6(2/6)**
>
> **Response to W2 and Q4:**
>
>  We appreciate the insightful comment from the reviewer regarding the evaluation budget and the potential for local optima. We acknowledge the importance of these considerations and provide further clarification and new experimental results below.
>
>
>
> 1. **Evaluation Budgets in Prior AHD Work**
>
> While MCTS-AHD employs 1,000 evaluations, this is not a universally adopted standard across the literature on Automated Heuristic Design (AHD). In fact, various representative frameworks have adopted budgets that align with their respective computational cost profiles:
>
> - **EoH**: Sets generations to 20 with a population size of 10–20, resulting in approximately 200–400 evaluations.
> - **ReEvo**: Explicitly limits evaluations to 100 shots due to the computational costs associated with Large Language Models (LLMs), as noted in their discussion.
> - **MCTS-AHD**: Uses 1,000 evaluations, which represents a high-budget setting.
>
>
>
> 2. **Why 200 Evaluations are Sufficient for TPD-AHD**
>
> Our choice of 200 evaluations is not driven by limitations but rather by the consistent early convergence observed in our method. As illustrated in **Figures 2, 4, and 10**, TPD-AHD generally saturates around 100 evaluations. The textual preference gradient provides strong directional guidance, while optimal anchoring and stochastic sampling ensure sufficient exploration to escape local optima. Thus, the need for fewer evaluations is not indicative of a restricted search space, but rather a reflection of the efficiency of the gradient-guided process.
>
>
>
> 3. **New 1,000-Evaluation Experiments**
>
> To further validate the robustness of TPD-AHD, we conducted additional experiments with an extended budget of 1,000 evaluations within the same framework. **Figure 11** presents the results, which demonstrate the following:
>
> - **Sustained Advantage**: TPD-AHD continues to outperform all baselines even with the expanded budget.
> - **Stable Convergence**: The performance curves remain smooth without late-phase oscillations, indicating that the method does not rely on low-budget artifacts.
> - **Small Marginal Gains**: The performance at 1,000 evaluations is only slightly better than at 200 evaluations, confirming that TPD-AHD identifies near-optimal heuristics early in the process.

---

> ### Author Response · Authors · 2025-11-28
> **Response to reviewer KmH6(3/6)**
>
> **Response to Q1:**
>
> Thank you for the insightful question. In our framework, textual gradients are not treated as final outputs but as latent "optimization directions." Their quality is assessed based on their ability to improve heuristic performance in subsequent iterations, rather than their linguistic form. This provides an objective, task-driven measure of their effectiveness: accurate gradients lead to better heuristics on the true objective function, while misleading gradients produce weaker candidates that are naturally filtered out.
>
>
>
> To ensure stability and prevent the LLM from generating catastrophic updates, we have implemented several mechanisms during the template and framework design:
>
>
>
> 1. **Structural Constraints of Heuristic Templates**:
>
> All generated heuristics are inserted into predefined function templates that enforce fixed input-output patterns, core control structures, and consistent task objectives. These templates act as a "semantic safety boundary," ensuring that even under gradient noise, the essential algorithmic skeleton remains intact and cannot collapse into invalid behavior. The specific template content is detailed in **Appendix D.4.**
>
>
>
> 2. **Fitness-Driven Evolutionary Selection:**
>
> TPD-AHD maintains a fitness-ranked candidate pool. Each heuristic generated via textual gradients is immediately evaluated on real combinatorial optimization tasks. Candidates degraded by misleading gradients are eliminated, ensuring that only beneficial directions survive and influence subsequent gradient signals. Thus, the quality of textual gradients is assessed by an external, quantitative metric—the objective function itself—rather than subjective linguistic quality.
>
>
>
> 3. **Smooth and Incremental Nature of Textual Gradients:**
>
> Unlike numerical gradients, textual gradients typically induce localized, incremental modifications (e.g., "increase selection pressure," "reduce excessive greediness early"). These suggestions produce smooth behavioral adjustments rather than abrupt structural changes, preventing drastic degradation and supporting a stable optimization trajectory.

---

> ### Author Response · Authors · 2025-11-28
> **Response to reviewer KmH6(4/6)**
>
> **Response to Q2:**
>
> Thank you for your insightful questions. In our framework, we generate a textual gradient for each optimal anchor pair $(h_1, h_i)$ and apply one independent gradient update per pair to produce a new candidate heuristic. We do not aggregate the N−1 gradients. This design is motivated by the following considerations:
>
>
>
> 1. **Textual gradients are semantic signals and cannot be linearly fused**
>
> Unlike numerical gradients, textual gradients express semantic optimization directions rather than aggregatable vectors. Direct fusion would introduce two major issues.
>
> - Semantic conflicts cannot be resolved linearly. For example, Pair A may suggest “reduce randomness,” while Pair B recommends “increase early exploration.” Natural language lacks an additive operator; fusion weakens both signals and yields ambiguous, diluted gradients.
>
> - Fusion amplifies noise and suppresses fine-grained differential cues. The optimal anchoring mechanism relies on each pair capturing the specific behavioral differences between $h_1$ and $h_i$. Merging all differences into one text blurs these signals.
>
>
>
> 2. **Pairwise textual gradients naturally exhibit an early-stage divergence and late-stage convergence pattern**
>
> In our framework, each anchor pair independently produces an updated heuristic, and all candidates are ranked in a unified pool based on real performance. This mechanism functions like an implicit multi-branch parallel gradient descent: different anchor pairs propose diverse optimization directions early on, while the candidate pool acts as an automatic selector of the most promising direction.
>
>
>
> Crucially, the apparent divergence of gradients may not be noise but a feature that promotes exploration. Before the system stabilizes, heterogeneous gradients expand the search space and maintain heuristic diversity. As iterations progress, high-performing heuristics begin to dominate the pool, reducing pairwise discrepancies and causing textual gradients to naturally align and converge.
>
>
>
> This behavior is clearly reflected in our empirical results (**Figures 2**, and the newly added **Figures 4, 9, and 10**): convergence curves exhibit strong divergence in early iterations and gradually merge in later stages, mirroring classical evolutionary dynamics and textual gradient alignment.
>
>
>
> 3. **Gradient fusion is theoretically possible but costly and not superior in practice**
>
> While we acknowledge the reviewer’s suggestion, there is no principled method for lossless fusion of multiple natural-language gradients. Fusion requires an additional N−1 gradient-reading and aggregation step, significantly increasing computational cost. Aggregated gradients tend to become abstract, less actionable, and more unstable. State-of-the-art systems (e.g., TEXTGRAD, TPO, REVOLVE) all adopt the single gradient paradigm.
>
>
>
> For these reasons, we use the more reliable, cost-efficient pairwise update scheme. However, we agree that semantic gradient aggregation is a promising research direction and plan to explore embedding-based methods in future work.

---

> ### Author Response · Authors · 2025-11-28
> **Response to reviewer KmH6(5/6)**
>
> **Response to Q3:**
>
> Thank you for the valuable questions. Your concerns primarily focus on TPD-AHD's robustness in cold-start scenarios, the potential for misleading textual gradients due to poor initial heuristics, and whether a warm start is necessary. Our response is as follows:
>
>
>
> 1. TPD-AHD is designed to handle cold-start scenarios without relying on strong initial heuristics. The initial templates used in our experiments, detailed in **Appendix D.2**, are either empty skeletons or extremely simple baselines (e.g., "random node selection"). Despite these weak initializations, TPD-AHD consistently generates effective heuristics from scratch, demonstrating its suitability for cold starts.
>
>
>
> 2. The convergence curves in **Figure 2** illustrate that, while Generation-0 candidates are highly noisy (a typical cold-start phenomenon), the optimization trajectories rapidly stabilize and outperform baselines across all tasks. The consistency of triple-run averages further indicates that the system continually corrects its direction rather than being misled by gradients. Persistent gradient misguidance would not result in such stable convergence.
>
>
>
> 3. Three core mechanisms ensure robustness against misleading gradients:
>
> **Optimal Anchor Preferences**: Textual gradients are based on relative preferences rather than absolute heuristic quality. Even in low-quality pools, comparing $h_1$ with hi reliably reveals "less bad → better" semantic differences, which are robust signals in natural language optimization.
>
> **Top-k Elite Retention**: This strategy prevents error accumulation by eliminating misleading candidates, ensuring they do not influence subsequent iterations.
>
> **Enhanced Exploration**: High-temperature sampling, multi-role templates, and LLM diversity expand early exploration, helping the system avoid premature lock-in caused by weak initialization.
>
>
>
> Additionally, we conducted comparative experiments with and without fixing the initial heuristic (**Figures 4 and 10**). The results show that TPD-AHD maintains good convergence and stability regardless of whether the initial heuristic is fixed, ultimately obtaining better heuristics.
>
>
>
>
>
>
>
> **Response to Q4:**
>
> See response to W2.

---

> ### Author Response · Authors · 2025-11-28
> **Response to reviewer KmH6(6/6)**
>
> **Response to Q5:**
>
>  Thank you for your valuable suggestions. We have detailed the computational costs in terms of LLM token usage and runtime for each component of TPD-AHD in **Table 11** of **Appendix C.3**. The results show that TPD-AHD-g1 (best anchoring + vanilla TextGrad) has the highest token consumption. This is because the original TextGrad framework is not specifically optimized for LLM-AHD tasks, resulting in substantial redundant informational noise that increases LLM token usage and ultimately degrades performance. In contrast, TPD-AHD-g2 (which uses only best anchoring) incurs the lowest token cost, as it does not employ any text-gradient mechanism and thus requires minimal computational resources. However, its performance is also the worst among all variants. TPD-AHD and its variants (TPD-AHD-p1 and TPD-AHD-p3) have comparable token consumption and runtime, attributed to their shared mechanism combining pairing and text-gradient.
>
>
>
> **Response to Q6:**
>
> We appreciate the reviewer's insightful question. The variation in initial points observed in **Figure 2** is due to the fact that each method was evaluated under a cold-start setting, where initial heuristics were independently sampled. Given the stochastic nature of LLM-generated heuristics, the initial performance naturally differs across methods.
>
>
>
> To address the reviewer's suggestion regarding a fair comparison of convergence, we conducted an additional experiment where all methods started from the same initial heuristic template (random-city initialization under RI). We re-ran the entire benchmark under this controlled warm-start condition, and the results are presented in **Figures 4 and 10**. These figures show that TPD-AHD consistently outperforms the baselines, regardless of whether the evaluation uses a cold-start or a fixed-start initialization. This confirms that the performance advantage of TPD-AHD is not attributable to differences in initialization.
>
>
>
> Regarding the low variance observed in TPD-AHD's performance, we conducted further experiments by repeating all methods 10 times to rigorously assess convergence stability. The aggregated convergence curves are shown in **Figure 10**. The results demonstrate that TPD-AHD indeed exhibits significantly lower variance compared to other approaches. We attribute this to the best-anchored preference pairing and textual-gradient guidance, which provide a clear optimization direction and reduce random drift in heuristic evolution. This mechanism inherently stabilizes the search process and accelerates convergence.
>
>
>
> We believe these additional experiments and explanations clarify the source of the initial differences and further demonstrate the robustness of TPD-AHD.
>
>
>
>
>
> **Response to Q7:**
>
> Thank you for your insightful question.
>
>
>
> Our ablation studies, as shown in **Tables 4 and 11**, demonstrate that none of the five ablation variants outperformed the full TPD-AHD model in any scenario. This suggests that the components of TPD-AHD are well-integrated and optimized for performance.
>
>
>
> Specifically, the results in **Figure 9 and Table 11** show a clear performance hierarchy: TPD-AHD > TPD-AHD-p1 > TPD-AHD-p3 > TPD-AHD-p2 > TPD-AHD-g1 > TPD-AHD-g2. This ordering reflects the effectiveness of the learning signals. For example, deliberate best-worst comparisons (TPD-AHD-p1) provide stronger guidance than using only best samples (TPD-AHD-p2), while random pairing (TPD-AHD-p3) introduces significant noise. The non-specialized TEXTGRAD (TPD-AHD-g1) dilutes useful information, and TPD-AHD-g2, lacking gradient-based updates, shows the weakest improvement capacity.
>
>
>
> In terms of stability, the ranking is: TPD-AHD > TPD-AHD-g2 > TPD-AHD-p2 > TPD-AHD-p1 > TPD-AHD-g1 > TPD-AHD-p3. TPD-AHD achieves both high performance and stability due to its well-designed components. In contrast, TPD-AHD-g2’s apparent stability is due to the lack of learning dynamics, limiting variability. Variants with noisy signals, like TPD-AHD-p3, exhibit the highest variance.
>
>
>
> Overall, the results indicate that TPD-AHD’s full model is not over-complex but rather finely tuned, with each component contributing to its superior performance and stability.

---

### Official Review · Reviewer_g1DZ · 2025-11-01

**Soundness:** 3
**Presentation:** 3
**Contribution:** 3
**Rating:** 4
**Confidence:** 4

**Summary:**

The paper introduces TPD-AHD, a novel framework for LLM-based automatic heuristic design that leverages “textual preference differentiation”: LLMs generate and compare pairs of heuristics, outputting interpretable textual feedback and improvement instructions (“textual gradients”). This aims to address the current inefficiencies and opacity of LLM-based algorithm design by making the optimization process more transparent, iterative, and directed. Empirical results on a suite of classic NP-hard combinatorial optimization problems and diverse real-world tasks show that TPD-AHD consistently outperforms prior LLM-AHD methods and provides better interpretability, with robust ablation and parameter sensitivity studies to support the claims.

**Strengths:**

1. This paper is well-written and the adopted methods are quite novel in this domain..

2. This paper adopts a wide collection of classic combinatorial benchmarks (TSP, CVRP, JSSP, MKP, etc.) and practical tasks (control and scientific discovery), and all recent state-of-the-art LLM-AHD methods are compared (Funsearch, ReEvo, EoH, MCTS-AHD). Results are persuable.

**Weaknesses:**

1.  The contribution is a little bit limited, mostly applying TEXTGRAD to LLM-based AHD.

2. The total process will make more LLM calls, which can be more expensive.

**Questions:**

1. Could you please make a comparison of the LLM cost of algorithm searches with baselines? I am very willing to improve the rating of this paper if you add this part.

2. I have a little doubt about the motivation. ReEvo provides a reflection step, and MCTS-AHD provides a tree structure. Why do you believe the interpreterbility can be improved when you only have both reflection and trajectory? Quote "existing methods often suffer from undirected search processes and poor interpretability, resulting in a black-box optimization paradigm"

---

> ### Author Response · Authors · 2025-11-28
> **Response to reviewer g1DZ(1/2)**
>
> **Response to W1:**
>
> Thank you for raising this important concern. We appreciate your feedback and agree that merely applying TEXTGRAD to LLM-based AHD would indeed offer limited novelty. However, TPD-AHD is not a direct application of TEXTGRAD. Instead, we have introduced several critical enhancements to make gradient-based textual optimization feasible, stable, and efficient in the context of LLM-based AHD. The key points are as follows:
>
>
>
> 1. Directly transferring TEXTGRAD to AHD would result in prohibitive computation and unstable search. To address this, we have added **Table 11**, which reports the LLM token consumption and time costs of TPD-AHD and its various ablation variants under the TSP Random Insertion framework. While TEXTGRAD is designed for general tasks without restrictions on task types, our experiments show that it works primarily for small-scale prompt tuning under stable input distributions. In contrast, AHD involves long, executable heuristic code, highly discrete and non-smooth search spaces, expensive LLM evaluations, and many iterations. Without redesigning the loss and gradient formulations, TEXTGRAD produces extremely noisy directions and dramatically increases LLM calls. Therefore, architectural modifications are essential for the AHD setting.
>
>
>
> 2. The Best-Anchored Preference Pairing mechanism is a core enhancement that is absent in TEXTGRAD. TEXTGRAD computes loss on individual items, but in AHD, this approach yields weak or noisy gradients, causing the loss of directional guidance and easy entrapment in local optima. Our mechanism consistently pairs each candidate with the current best heuristic, amplifying meaningful differences and stabilizing gradient directions. As shown in **Figure 9**, this pairing significantly reduces noise, strengthens preference signals, and provides smoother optimization trajectories. This is a substantive and necessary extension beyond TEXTGRAD.
>
>
>
> 3. The ablation studies confirm that naïvely applying TEXTGRAD to AHD performs poorly. As shown in **Table 4 and 11**, removing the Best-Anchored Preference Pairing degrades results, while removing textual gradients harms search efficiency. Additionally, **Figure 9** demonstrates that other variants that alter the preference mechanism fail to converge or converge extremely slowly. These results clearly demonstrate that the performance gains of TPD-AHD come from the synergy between TEXTGRAD and our proposed preference enhancement mechanism, rather than from TEXTGRAD alone.
>
>
>
> In summary, TPD-AHD is not merely applying TEXTGRAD to LLM-based AHD. The enhancements we have introduced are critical for making gradient-based textual optimization feasible and effective in the context of AHD.
>
> **Response to W2 and Q1:**
>
> We sincerely appreciate your valuable suggestion regarding the assessment of LLM cost. To address this concern, we have conducted additional experiments to compare the LLM cost of our proposed method with the baselines. Specifically, we have measured both the token consumption and the time expenditure for each method. These results are now summarized in **Table 7** of the revised manuscript. We believe this detailed comparison will provide a clearer understanding of the trade-offs involved and support the evaluation of our method's efficiency.

---

> ### Author Response · Authors · 2025-11-28
> **Response to reviewer g1DZ(2/2)**
>
> **Response to Q1:**
>
> See response to W2.
>
> **Response to Q2:**
>
> Thank you for raising this important point. Your question highlights the need to clarify how TPD-AHD's approach to interpretability differs from existing methods like ReEvo and MCTS-AHD.
>
>
>
> 1. **Textual Gradients vs. Reflection Mechanisms**:
>
> Reflection mechanisms in ReEvo are heuristic and undirected. They provide general hints without explicitly grounding them in performance differences. For example, a reflection might suggest, "Prioritize edges with lower attributes after normalization," but it lacks detailed, performance-based reasoning.
>
>
>
> In contrast, TPD-AHD's textual gradients are optimization-driven and grounded in explicit preference signals. By comparing the current best heuristic $h_w$ with a worse heuristic $h_l$, TPD-AHD generates detailed textual losses that include:
>
> - Advantages of the better heuristic.
> - Weaknesses of the worse heuristic.
> - Structured reasoning for improvement.
>
>
>
> These losses are then converted into precise, actionable update directions (e.g., adaptive biasing or refined detection). This approach ensures that each update is directly linked to observed performance gaps, providing clear, causal explanations.
>
>
>
> 2. **Causal Interpretability through Preference Pairing**:
>
> TPD-AHD's preference pairing mechanism makes the optimization process causally interpretable. It explicitly identifies:
>
> - Why the inferior heuristic performs poorly.
> - What concrete differences separate it from the superior heuristic.
> - How these differences should be addressed to improve performance.
>
>
>
> This mirrors human evaluation strategies, where contrasting outcomes lead to actionable insights. Our ablation studies (**Table 4 and 11**) demonstrate that removing optimal anchoring or replacing it with random pairing significantly degrades performance. This underscores the importance of the causal chain: preference difference → interpretable explanation → targeted gradient → effective evolution.
>
>
>
> 3. **Semantic Update Trajectory vs. MCTS Tree Structure**:
>
> The MCTS tree structure in MCTS-AHD shows the search path but lacks semantic control or causal explanation for performance improvements. Node transitions are driven by generic LLM sampling, without clear reasoning.
>
>
>
> In contrast, TPD-AHD produces a semantic, causally grounded update trajectory. Each iteration is justified by explicit gradient instructions that explain:
>
> - The performance issue.
> - How the update addresses it.
> - Why this improves the heuristic's behavior.
>
>
>
> This approach resembles gradient-descent optimization in the natural-language space, where each step is a directed semantic update with clear causality and human-interpretable reasoning.
>
>
>
> In summary, TPD-AHD's textual gradients and preference pairing provide a qualitatively different mechanism for interpretability compared to reflection-based or tree-search-based methods. By grounding updates in explicit performance differences and causal reasoning, TPD-AHD offers a more transparent and directed optimization process.

---

### Official Review · Reviewer_BibP · 2025-11-01

**Soundness:** 1
**Presentation:** 3
**Contribution:** 1
**Rating:** 2
**Confidence:** 4

**Summary:**

This paper introduces a framework called ``TPD-AHD``, which incorporates a ``textual preference differentiation`` mechanism into the field of ``LLM-AHD``, and conducts experiments on various combinatorial optimization problems.

**Strengths:**

This paper is clearly written, the figures are well-crafted, and the comparison methods are comprehensive.

**Weaknesses:**

1. Despite the introduction of the textual gradient, it remains entirely dependent on the LLM, offers no rigorous convergence guarantees, and lacks stability.

2. As shown by the ``select_next_node`` in the appendix, the heuristic still boils down to a localized, fixed-rule-based random-greedy selection loop; the overall algorithm performs no graph-structure learning nor makes use of any global information.

3. Taking the ``TSP`` as an example, the results in Table 1 are even worse than the ``random-insertion`` algorithm proposed by GLOP [1], another step-by-step construction framework.

4. Regarding the choice of baselines and datasets. For the ``CVRP`` task, it is recommended to use HGS as a baseline. Both ``LKH`` and ``HGS`` are not exact algorithms in the strict sense; referring to their results as ``optimal`` is therefore imprecise. In terms of datasets, the paper lacks real-world benchmarks such as ``TSPLIB`` and ``VRPLIB``.

5. The method’s potential is likely bounded by both the initial prompt and the underlying constructive template; the authors need provide deeper studies on these limitations.

[1] *GLOP: Learning Global Partition and Local Construction for Solving Large-scale Routing Problems in Real-time, AAAI 2024*

**Questions:**

See ``Weakness``

---

> ### Author Response · Authors · 2025-11-28
> **Response to reviewer BibP(1/4)**
>
> **Response to W1：**
>
> Thank you for raising this important issue. We acknowledge that theoretical convergence guarantees remain an open challenge across the entire field of "LLM-based Automated Heuristic Design (LLM-AHD)". Similar to FunSearch, EoH, ReEvo, and MCTS-AHD, our method operates in an LLM-driven search space. However, due to the inherent non-determinism of LLM reasoning, strict convergence guarantees cannot be achieved at present. Thus, this limitation is not unique to our approach but is a common characteristic of existing LLM-AHD techniques.
>
>
>
> Nevertheless, our experimental results demonstrate the stability and effectiveness of TPD-AHD:
>
>
>
> 1. **Interpretable Optimization Signal**: Textual gradients represent a structured and interpretable optimization signal. Although their generation relies on LLMs, compared with purely sampling-driven evolutionary methods, they can provide clearer update directions by anchoring optimal preference information, thereby enhancing the stability of heuristic evolution to a certain extent.
>
>
>
> 2. **Empirical Stability and Convergence**: As illustrated in **Figure 2**, across three independent runs, TPD-AHD exhibits lower variance and faster performance improvement compared to EoH, ReEvo, and MCTS-AHD. In addition, we conducted a series of supplementary experiments. Under the TSP Random Insertion framework, each LLM-AHD method was independently run 10 times, and the corresponding data are presented in **Table 8**. Meanwhile, we plotted convergence curves and boxplots, which are illustrated in **Figure 10**. These results demonstrate that despite the absence of closed-form theoretical guarantees, TPD-AHD is empirically more stable and exhibits a stronger convergence trend than existing methods.
>
>
>
> 3. **Best-anchored Preference Pairing**: This strategy further enhances stability by consistently pairing each candidate heuristic with the currently best-performing one. This approach anchors the optimization direction to a stable reference point, reducing noise from low-quality heuristics. The ablation experiment in **Table 4** confirms that removing this component leads to performance degradation. Furthermore, the ablation experiments on the TSP Random Insertion framework in **Table 11** demonstrate that the ablation variant without the best-anchored mechanism exhibits increased variance and decreased performance.

---

> ### Author Response · Authors · 2025-11-28
> **Response to reviewer BibP(2/4)**
>
> **Response to W2：**
>
> Thank you for this insightful comment. We appreciate the opportunity to address the concerns raised and clarify the core objectives and capabilities of our work.
>
>
>
> 1. **Research Objective and Scope**: We would like to clarify a fundamental aspect of our research objective. The primary goal of this paper is to advance automated heuristic generation, rather than to focus on graph representation learning. TPD-AHD is a training-free Automated Heuristic Design (AHD) framework, which is inherently complementary and orthogonal to graph learning methods (such as Graph Neural Networks, attention-based solvers, and deep constructive models). Its purpose is not to learn embedding vectors or global latent representations, but to automatically synthesize human-interpretable heuristic rules. This aligns with the standard paradigm in the fields of classical hyper-heuristics and automated heuristic design.
>
>
>
> 2. **Framework and Global Information Utilization**: The Step-by-Step Construction Framework is one of the typical forms of constructive heuristics we adopted in our combinatorial optimization approach. However, it is important to note that this is not the sole framework we explored. As detailed in the paper, we also introduced the Ant Colony Optimization Framework for heuristic design. In this framework, our design template provides LLMs with global node distance information, enabling TPD-AHD to effectively leverage such information to iteratively generate high-quality heuristics. We have supplemented detailed examples of this framework in **Appendix D.4**, specifically **Figures 21 and 24**.
>
>
>
>
> 3. **LLM Generation and Evolutionary Iterations**: The heuristic templates we provide only serve to regulate LLM generation for the convenience of subsequent evaluation, without imposing full constraints on LLMs. Through continuous evolutionary iterations, LLMs integrate rich global optimization information under the guidance of optimal anchored preference information and textual gradients. Via textual loss difference analysis and iterative optimization, TPD-AHD generates a series of rules with progressively increasing complexity. During the iteration process, we observe that the generated rules continuously integrate:
>
> 	- Global statistical information (e.g., minimum/maximum/mean of node distances);
>
> 	- Global path metrics (e.g., accumulated path length, number of unvisited nodes);
>
> 	- Lookahead reasoning logic (e.g., penalty mechanisms based on future feasibility);
>
> 	- Multi-step scoring mechanisms that combine local and global features.
>
>
>
> We further take a heuristic generated by TPD-AHD within the TSP Random Insertion framework as an example, where similar global information can be observed. This part of the content is supplemented in **Figures 17, 18 and 19.**
>
>
>
> These patterns consistently emerge in the evolved heuristics (see **Appendix D** for details, where the optimal heuristics for some problem are presented along with a separate statistics summary of global information), indicating that the generated strategies are far more sophisticated than simple greedy strategies.

---

> ### Author Response · Authors · 2025-11-28
> **Response to reviewer BibP(3/4)**
>
> **Response to W3：**
>
>  We appreciate the reviewer's insightful comment regarding the comparison with the Random-Insertion (RI) algorithm from GLOP. We acknowledge the importance of evaluating TPD-AHD against this relevant baseline and have conducted additional experiments to provide a more comprehensive comparison. Here are the key findings:
>
>
>
> 1. **TPD-AHD Outperforms Random-Insertion within the Same Framework**: We integrated TPD-AHD into the Random-Insertion framework to optimize the insertion position selection rules. The results demonstrate that TPD-AHD significantly improves upon the vanilla Random-Insertion algorithm. Specifically, in the TSP-50 benchmark, TPD-AHD (RI) outperforms not only vanilla Random-Insertion but also other LLM-based heuristic design methods such as EoH (RI), ReEvo (RI), and FunSearch (RI). Detailed results are provided in **Appendix C.2. Figure 4 and Table 7** show that TPD-AHD consistently achieves the best performance among all evaluated methods within the Random-Insertion framework, highlighting the effectiveness of our approach.
>
>
>
> 2. **Framework Differences Explain Performance Discrepancies**: The results in **Table 1** are based on the standard "step-wise nearest neighbor" greedy constructive framework, which is consistent with the frameworks used by FunSearch, EoH, ReEvo, and MCTS-AHD. In contrast, Random-Insertion is a different meta-heuristic framework that relies on global partial path structures rather than local scoring. These fundamental differences in search bias make direct comparisons between "TPD-AHD (Constructive Framework)" and "Random-Insertion" invalid, as they operate within distinct frameworks. Therefore, the performance differences observed in **Table 1** are not a direct reflection of the algorithms' capabilities but rather a result of the different frameworks used.
>
>
>
> 3. **Consistent Superiority of TPD-AHD under Fair Evaluation**: When evaluated within the same meta-heuristic framework, TPD-AHD consistently outperforms all baselines. Under the constructive framework, **Table 1** shows that TPD-AHD achieves the best performance among LLM-based heuristic design methods. Similarly, under the Random-Insertion framework, our new experiments confirm that TPD-AHD reaches state-of-the-art performance. Thus, the claim that "TPD-AHD performs worse than Random-Insertion" does not hold when comparisons are made under consistent settings.

---

> ### Author Response · Authors · 2025-11-28
> **Response to reviewer BibP(4/4)**
>
> **Response to W4：**
>
> We thank the reviewer for the valuable feedback regarding the choice of baselines and datasets. In response, we have made the following key revisions:
>
>
>
> 1. **Clarification on "Optimal" Solutions for CVRP**: We acknowledge that neither LKH nor HGS are exact solvers for CVRP. To improve clarity and accuracy, we have replaced all references to "optimal" solutions with results obtained from HGS, which is currently the strongest publicly available classical solver for CVRP. All tables and related textual descriptions have been updated accordingly.
>
>
>
> 2. **Inclusion of HGS as a Baseline**: Following the reviewer’s suggestion, we have added HGS as a core classical baseline for the CVRP experiments, ensuring a fairer and more rigorous comparison.
>
>
>
> 3. **Addition of Real-World Benchmarks (TSPLIB and VRPLIB)**: To enhance the generalizability and practical relevance of our evaluation, we have incorporated results from widely used real-world benchmark datasets. Specifically, we now include:
>
> 	- Results on TSPLIB instances using the RI framework (**Table 9**)
>
> 	- Results on VRPLIB instances using the ACO framework (**Table 10**)
>
>
>
> These new experiments demonstrate that TPD-AHD maintains strong performance across both synthetic and real-world problem instances.
>
>
>
>
>
>  **Response to W5：**
>
>  Thank you for raising this important point regarding the potential limitations imposed by the initial prompt and the underlying template structure. We appreciate your insight and acknowledge that these elements play a critical role in shaping the search space for heuristic algorithms. Here is a detailed response addressing your concerns:
>
>
>
> 1. It is essential to clarify the role of templates in our framework. Templates serve as high-level interface boundaries that define the structure within which LLMs propose heuristic algorithms. They do not impose restrictive inductive biases but rather provide a flexible and expressive search space. While templates specify the interface boundaries, they do not limit the function forms that large language models (LLMs) can generate. Our experiments across multiple frameworks—including the constructive framework, Ant Colony Optimization (ACO) framework, Random Insertion (RI) framework, as well as other scientific discovery and machine learning frameworks (**Table 6**)—demonstrate that the evolved heuristics exhibit rich and diverse behavioral patterns, which are not explicitly hard-coded in the templates. This indicates that even within a simple structural framework, LLMs can produce complex and effective heuristic algorithms. The templates are designed to guide the search process while allowing for sufficient flexibility to explore a wide range of heuristic forms. Specific problem templates are detailed in **Appendix D.4**.
>
>
>
>
>
> 2. Our template diversity experiments further validate that the framework is not restricted to specific constructive templates. To investigate whether the template structure limits performance, we instantiated TPD-AHD within an alternative constructive paradigm, i.e., Random Insertion. The results show that TPD-AHD (RI) significantly outperforms the vanilla Random Insertion heuristic and all other AHD baselines within the Random Insertion framework. Moreover, the discovered heuristics incorporate strategies not present in the original Random Insertion template. This finding underscores that the framework's performance is not tied to specific templates and that it can uncover effective strategies beyond those explicitly encoded in the templates.
>
>
>
> We agree with you that exploring how template structures influence the expressiveness and optimization dynamics of LLM-driven heuristic generation is a highly meaningful research direction. Future work will focus on systematically investigating the impact of different template structures and developing methods to dynamically adapt templates to enhance the framework's performance and generalizability.

---

### Author Response · Authors · 2025-11-28
**Global response**

We sincerely appreciate all the reviewers for their careful examination of our manuscript and for providing numerous insightful comments and valuable suggestions. Your feedback has been instrumental in enhancing the quality and robustness of our work.



In response to the reviewers' feedback, we have conducted additional experiments and incorporated the results into the revised manuscript. The modified sections are highlighted in blue for easy reference.



**Specific Supplementary Content:**

We have supplemented the TSP Random Insertion heuristic design framework (**Appendix B.3**) and completed the following experiments under this framework:

1. Comparative experiments between TPD-AHD and EoH, ReRvo, MCTS-AHD (**Figure 4, Table 7**).

2. Comparative experiments between TPD-AHD and EoH, ReRvo, MCTS-AHD under different LLMs (**Table 7**).

3. Comparative experiments of fixed initial heuristics versus unfixed initial heuristics (**Figures 4, 10**).

4. Stability analysis of TPD-AHD, EoH, ReRvo, and MCTS-AHD (**Table 8**).

5. Comparative experiments between TPD-AHD and EoH, ReRvo, MCTS-AHD on TSPLib and CVRPLib (**Tables 9, 10**).

6. Ablation experiments of TPD-AHD with different ablation variants (**Figure 9, Table 11**).

7. Comparative experiments between TPD-AHD and EoH, ReRvo, MCTS-AHD under 1000 iterations (**Figure 11**).



Additionally, we have included:

1. Sample global analysis of heuristics generated by TPD-AHD (**Figures 17, 18, 19**).

2. Templates for some optimization tasks used by TPD-AHD (**Figures 20–25**).

3. New heuristic evolution samples of TPD-AHD under the TSP ACO and TSP Random Insertion frameworks (**Figures 29, 30**).



**Code Availability:**

To ensure reproducibility and facilitate further research, the relevant code for TPD-AHD has been made publicly available. The link to the anonymous GitHub repository is provided below. This repository includes all the figures, tables mentioned in the response, and the manuscript's code.

https://anonymous.4open.science/r/TPD-AHD/

---

### Meta-Review · Area_Chair_Zodx · 2025-12-09

**Summary:**

Thanks to the reviewers for their valuable comments from many different perspectives. Overall, I think their main problems at present lie in:

- Lack of theoretical guarantees and limited novelty.

- May be trapped in a local optimum.

- Some baseline methods are missing.

- The experimental results need further supplementation.



In addition, some reviewers mentioned issues such as reproducibility.

**Reviewer Concerns:**

I am very grateful for the reply provided by the author. I believe that some of the reviewers' questions will be resolved, such as some explanations regarding the experimental details.



However, the concerns of some reviewers regarding the novelty of the method may not be resolved. In addition, the code provided by the author has expired, and its reproducibility cannot be confirmed.



Overall, considering the potential score increase, I think this paper cannot convince most reviewers and is below the acceptance threshold.

**Reviewer Scores:**

For the Reviewer BibP, he may not be convinced by the explanation regarding the theoretical guarantee. I think this reviewer will maintain the score (**Rating:** 2).

For the Reviewer g1DZ, he may accept the explanation regarding the computational cost, but his assessment of the method contribution may not change. I think the reviewer will upgrade to a positive score (**Rating:** 4 to 6).

For Reviewer KmH6, he may accept most of the explanations, but he may be skeptical about the optimization problem. I think the reviewer will raise the score to a positive one (low probability) (**Rating:** 4 to 6).

For the Reviewer FC4K, he may accept explanations regarding the details of the experiment, but doubts about novelty may still exist. I think this reviewer will maintain the score (**Rating:** 2).

---

### Decision · Program_Chairs · 2026-01-26

Reject